# Shared Adversarial Unlearning: Backdoor Mitigation by Unlearning Shared Adversarial Examples

**Shaokui Wei[1]**    **Mingda Zhang[1]**    **Hongyuan Zha[1,2]**    **Baoyuan Wu[1]***

[1]School of Data Science,
The Chinese University of Hong Kong, Shenzhen (CUHK-Shenzhen), China
[2]Shenzhen Institute of Artificial Intelligence and Robotics for Society, China

## Abstract

Backdoor attacks are serious security threats to machine learning models where an adversary can inject poisoned samples into the training set, causing a backdoored model which predicts poisoned samples with particular triggers to particular target classes, while behaving normally on benign samples. In this paper, we explore the task of purifying a backdoored model using a small clean dataset. By establishing the connection between backdoor risk and adversarial risk, we derive a novel upper bound for backdoor risk, which mainly captures the risk on the shared adversarial examples (SAEs) between the backdoored model and the purified model. This upper bound further suggests a novel bi-level optimization problem for mitigating backdoor using adversarial training techniques. To solve it, we propose **S**hared **A**dversarial **U**nlearning (SAU). Specifically, SAU first generates SAEs, and then, unlearns the generated SAEs such that they are either correctly classified by the purified model and/or differently classified by the two models, such that the backdoor effect in the backdoored model will be mitigated in the purified model. Experiments on various benchmark datasets and network architectures show that our proposed method achieves state-of-the-art performance for backdoor defense. The code is available at https://github.com/SCLBD/BackdoorBench (PyTorch) and https://github.com/shawkui/MindTrojan (MindSpore).

## 1   Introduction

Recent decades have witnessed a growing application of deep neural network (DNN) techniques in various domains, such as face recognition, autonomous driving, and medical image processing [1, 17, 32, 44]. However, DNNs are known to be vulnerable to malicious attacks that can compromise their security and reliability. One of the emerging threats is the backdoor attack, where an attacker attempts to inject stealthy backdoor into a DNN model by manipulating a small portion of the training data such that the poisoned model will behave normally for clean inputs while misclassifying any input with the trigger pattern to a target label. Since the poisoned models can have real-world consequences, such as allowing an attacker to gain unauthorized access to a system or to cause physical harm, it is crucial to develop effective methods for backdoor defense in developing and deploying DNNs.

Adversarial training (AT) is one of the most effective methods for improving the robustness of DNNs [33]. AT is typically formulated as a min-max optimization problem, where the inner maximization aims to find adversarial perturbations that fool the model, while the outer minimization is to reduce the adversarial risk. Recently, several works have explored the use of AT for defending against backdoor attacks. However, these works face some limitations and challenges. Weng et al. [52] and

---

*Corresponds to Baoyuan Wu (wubaoyuan@cuhk.edu.cn).

37th Conference on Neural Information Processing Systems (NeurIPS 2023).

Gao et al. [13] found that AT can only be effective for mitigating backdoor attacks with certain trigger patterns and may even increase the vulnerability to backdoor attacks when training from scratch on poisoned datasets. Zeng et al. [59] proposed to unlearn the universal adversarial perturbation (UAP) to remove backdoors from poisoned models. Since directly unlearning UAP leads to highly unstable performance for mitigating backdoors, they employ implicit hyper gradient to solve the min-max problem and achieve remarkable performance. However, their method assumes that the same trigger can activate the backdoor regardless of the samples it is planted on, and therefore lacks a guarantee against more advanced attacks that use sample-specific and/or non-additive triggers.

In this paper, we consider the problem of purifying a poisoned model. After investigating the relationship between adversarial examples and poisoned samples, a new upper bound for backdoor risk to the fine-tuned model is proposed. Specifically, by categorizing adversarial examples into three types, we show that the backdoor risk mainly depends on the shared adversarial examples that mislead both the fine-tuned model and the poisoned model to the same class. Shared adversarial examples suggest a novel upper bound for backdoor risk, which combines the shared adversarial risk and vanilla adversarial risk. Besides, the proposed bound can be extended with minor modifications to universal adversarial perturbation and targeted adversarial perturbation. Based on the new bound, we propose a bi-level formulation to fine-tune the poisoned model and mitigate the backdoor. To solve the bi-level problem, we proposed **S**hared **A**dversarial **U**nlearning (SAU). SAU first identifies the adversarial examples shared by the poisoned model and the fine-tuned model. Then, to break the connection between poisoned samples and the target label, the shared adversarial examples are unlearned such that they are either classified correctly by the fine-tuned model or differently by the two models. Moreover, our method can be naturally extended to defend against backdoor attacks with multiple triggers and/or multiple targets. To evaluate our method, we compare it with six state-of-the-art (SOTA) defense methods on seven SOTA backdoor attacks with different model structures and datasets. Experimental results show that our method achieves comparable and even superior performance to all the baselines.

Our contributions are three folds: **1)** We analyze the relationship between adversarial examples and poisoned samples, and derive a novel upper bound for the backdoor risk that can be generalized to various adversarial training-based methods for backdoor defense; **2)** We formulate a bi-level optimization problem for mitigating backdoor attacks in poisoned models based on the derived bound, and propose an efficient method to solve it; **3)** We conduct extensive experiments to evaluate the effectiveness of our method, and compare it with six state-of-the-art defense methods on seven challenging backdoor attacks with different model structures and datasets.

## 2 Related work

**Backdoor attack.** Backdoor attack is one of the major challenges to the security of DNNs. The poisoned model behaves normally on clean inputs but produces the target output when the trigger pattern is present. Based on the types of triggers, backdoor attacks can be classified into two types: fixed-pattern backdoor attacks and sample-specific backdoor attacks. BadNets [15] is the first backdoor attack that uses fixed corner white blocks as triggers. To improve the stealth of the triggers, Blended [7] is proposed to blend the trigger with the image in a weighted way. Since fixed-pattern triggers can be recognized easily, sample-specific backdoor attacks have been proposed. SSBA [30], WaNet [38], LF [58], IRBA [12], VSSC [49] and TAT [8] use different techniques to inject unique triggers for different samples from different angles. Sleeper-agent [42] and Lira [9] optimize the target output to obtain more subtle triggers. Recently, Zhu et al. [67] proposed a learnable poisoning sample selection strategy to further boost the effect of backdoor attacks. To keep the label of the backdoor image matching the image content, LC [40] and SIG [2] use counterfactual and other methods to modify the image to deploy clean label attacks.

**Backdoor defense.** Backdoor defense aims to reduce the impact of backdoor attacks on deep neural networks (DNNs) through training and other means. There are three types of backdoor defense: pre-processing, in-processing, and post-processing. Pre-processing backdoor defense aims to identify the poisoned samples in the training dataset. For instance, AC [5] filters out the poisoned samples by the abnormal activation clustering phenomenon in the target class; Confusion Training [39] identifies the poisoned samples by training a poisoned model that only fits the poisoned samples; VDC [68] incorporates multimodal large language models to detect the poisoned samples. In-processing

backdoor defense reduces the effect of the backdoor during training. For instance, ABL [27] exploits the fact that the learning speed of backdoor samples is faster than that of the clean sample, and splits some poisoned samples to eliminate the backdoor by forgetting these poisoned samples; DBD [19] divides the backdoor training process and directly inhibits the backdoor learning process; D-ST [6] splits the backdoor samples and uses semi-supervised learning by observing that the clean samples are more robust to image transformation than the backdoor samples. Post-processing backdoor defense mitigates the effect of backdoors for a poisoned model by pruning or fine-tuning. For example, FP [31] prunes some potential backdoor neurons and fine-tunes the model to eliminate the backdoor effect; ANP [55] finds the backdoor neurons by adversarial perturbation to model weights; EP [64] and CLP [63] distinguish the characteristics of the backdoor neurons from the clean neurons; NAD [28] uses a mild poisoned model to guide the training of the poisoned model to obtain a cleaner model; I-BAU [59] finds possible backdoor triggers by the universal adversarial attack and unlearn these triggers to purify the model; FT-SAM [65] boosts the performance of fine-tuning for backdoor mitigation by incorporating sharpness-aware minimization; NPD [66] learns a lightweight linear transformation layer by solving a well designed bi-level optimization problem to defend against backdoor attack.

**Adversarial training.**   In adversarial training, models are imposed to learn the adversarial examples in the training stage and therefore, resistant to adversarial attacks in the inference stage. In one of the earliest works [14], the adversarial examples are generated using Fast Gradient Sign Method. In [33], PGD-AT is proposed, which generates adversarial examples by running FGSM multiple steps with projection and has become the most widely used baseline for adversarial training. Some further improvements of PGD-AT include initialization improvement [21, 23], attack strategy improvement [22], and efficiency improvement [53, 62].

## 3  Methodology

In Section 3.1, we first introduce notations, threat model, and defense goal to formulate the problem. By investigating the relationship between adversarial risk and backdoor risk, a new upper bound of backdoor risk is derived in Section 3.2, from which a bi-level formulation is proposed in Section 3.3.

### 3.1  Preliminary

**Notations.**   We consider a $K$-class ($K \geq 2$) classification problem that aims to predict the label $y \in \mathcal{Y}$ of a given sample $\boldsymbol{x} \in \mathcal{X}$, where $\mathcal{Y} = [1, \cdots, K]$ is the set of labels and $\mathcal{X}$ is the space of samples. Let $h_{\boldsymbol{\theta}} : \mathcal{X} \to \mathcal{Y}$ be a DNN classifier with model parameter $\boldsymbol{\theta}$. We use $\mathcal{D}$ to denote the set of data $(\boldsymbol{x}, y)$. For simplicity, we use $\boldsymbol{x} \in \mathcal{D}$ as a abbreviation for $(\boldsymbol{x}, y) \in \mathcal{D}$. Then, for a sample $\boldsymbol{x} \in \mathcal{D}$, its predicted label is

$$h_{\boldsymbol{\theta}}(\boldsymbol{x}) = \arg\max_{k=1,\cdots,K} \boldsymbol{p}_k(\boldsymbol{x}; \boldsymbol{\theta}), \tag{1}$$

where $\boldsymbol{p}_k(\boldsymbol{x}; \boldsymbol{\theta})$ is the (softmax) probability of $\boldsymbol{x}$ belonging to class $k$.

**Threat model.**   Let $\mathcal{V}$ be the set of triggers and define $g : \mathcal{X} \times \mathcal{V} \to \mathcal{X}$ as the generating function for poisoned samples. Then, given a trigger $\Delta \in \mathcal{V}$ and a sample $\boldsymbol{x} \in \mathcal{X}$, one can generate a poisoned sample $g(\boldsymbol{x}; \Delta)$. For simplicity, we only consider all to one case, *i.e.*, there is only one trigger $\Delta$ and one target label $\hat{y}$. The all to all case and multi-trigger case are left in **Appendix** A. We assume that the attacker has access to manipulate the dataset and/or control the training process such that the trained model classifies the samples with pre-defined trigger $\Delta$ to the target labels $\hat{y}$ while classifying clean samples normally. In addition, we define the poisoning ratio as the proportion of poisoned samples in the training dataset.

**Defense goal.**   We consider a scenario where a defender is given a poisoned model with parameter $\theta_{bd}$ and a small set of *clean* data $\mathcal{D}_{cl} = \{(\boldsymbol{x}_i, y_i)\}_{i=1}^{N}$. Since we are mainly interested in samples whose labels are not the target label, we further define the set of non-target samples as $\mathcal{D}_{-\hat{y}} = \{(\boldsymbol{x}, y) | (\boldsymbol{x}, y) \in \mathcal{D}_{cl}, y \neq \hat{y}\}$. Note that the size of $\mathcal{D}_{cl}$ is small and the defender cannot train a new model from scratch using only $\mathcal{D}_{cl}$. The defender's goal is to purify the model so that the clean performance is maintained and the backdoor effect is removed or mitigated. We assume that the defender cannot access the trigger $\Delta$ or the target label $\hat{y}$.

**Problem formulation.** Using the 0-1 loss [50, 60], the classification risk $\mathcal{R}_{cl}$ and the **backdoor risk** $\mathcal{R}_{bd}$ with respect to classifier $h_{\boldsymbol{\theta}}$ on $\mathcal{D}_{cl}$ can be defined as:

$$\mathcal{R}_{cl}(h_{\boldsymbol{\theta}}) = \frac{1}{N} \sum_{i=1}^{N} \mathbb{I}(h_{\boldsymbol{\theta}}(\boldsymbol{x}_i) \neq y_i), \quad \mathcal{R}_{bd}(h_{\boldsymbol{\theta}}) = \frac{\sum_{i=1}^{N} \mathbb{I}(h_{\boldsymbol{\theta}}(g(\boldsymbol{x}_i, \Delta)) = \hat{y}, \boldsymbol{x}_i \in \mathcal{D}_{-\hat{y}})}{|\mathcal{D}_{-\hat{y}}|} \quad (2)$$

where $\mathbb{I}$ is the indicator function and $|\cdot|$ denotes the cardinality of a set.

In (2), the classification risk concerns whether a clean sample is correctly classified, while the backdoor risk measures the risk of classifying a poisoned sample from $\mathcal{D}_{-\hat{y}}$ to target class $\hat{y}$.

In this paper, we consider the following problem for purifying a poisoned model:

$$\min_{\boldsymbol{\theta}} \mathcal{R}_{cl}(h_{\boldsymbol{\theta}}) + \lambda \mathcal{R}_{bd}(h_{\boldsymbol{\theta}}), \quad (3)$$

where $\lambda$ is a hyper-parameter to control the tradeoff between classification risk and backdoor risk.

However, directly solving (3) is impractical due to the lack of the trigger $\Delta$ and target label $\hat{y}$. Therefore, a natural choice is to replace $\mathcal{R}_{bd}$ with a surrogate risk irrelevant to $\Delta$ and $\hat{y}$.

### 3.2 Connection between backdoor risk and adversarial risk

To tackle the above problem, we propose a novel upper bound of backdoor risk which can be relaxed to a surrogate of $\mathcal{R}_{bd}$ that is independent of $\Delta$ and $\hat{y}$. Before that, we first decompose the adversarial risk and review the connection between backdoor risk and adversarial risk.

**Adversarial risk.** Given a sample $\boldsymbol{x}$ and an adversarial perturbation $\boldsymbol{\epsilon} \in \mathcal{S}$, an adversarial example $\tilde{\boldsymbol{x}}_{\boldsymbol{\epsilon}}$ can be generated by $\tilde{\boldsymbol{x}}_{\boldsymbol{\epsilon}} = \boldsymbol{x} + \boldsymbol{\epsilon}$, where $\mathcal{S}$ is the set of perturbations. We define the set of adversarial example-label pair to $h_{\boldsymbol{\theta}}$ generated by perturbation $\boldsymbol{\epsilon}$ and dataset $\mathcal{D}_{cl}$ by $\mathcal{A}_{\boldsymbol{\epsilon},\boldsymbol{\theta}} = \{(\tilde{\boldsymbol{x}}_{\boldsymbol{\epsilon}}, y) | h_{\boldsymbol{\theta}}(\tilde{\boldsymbol{x}}_{\boldsymbol{\epsilon}}) \neq y, (\boldsymbol{x}, y) \in \mathcal{D}_{cl}\}$ and abbreviate $(\tilde{\boldsymbol{x}}_{\boldsymbol{\epsilon}}, y) \in \mathcal{A}_{\boldsymbol{\epsilon},\boldsymbol{\theta}}$ by $\tilde{\boldsymbol{x}}_{\boldsymbol{\epsilon}} \in \mathcal{A}_{\boldsymbol{\epsilon},\boldsymbol{\theta}}$. Then, the vanilla adversarial risk for $h_{\boldsymbol{\theta}}$ on $\mathcal{D}_{-\hat{y}}$ can be defined as

$$\mathcal{R}_{adv}(h_{\boldsymbol{\theta}}) = \frac{\sum_{i=1}^{N} \max_{\boldsymbol{\epsilon}_i \in \mathcal{S}} \mathbb{I}(\tilde{\boldsymbol{x}}_{i,\boldsymbol{\epsilon}_i} \in \mathcal{A}_{\boldsymbol{\epsilon}_i,\boldsymbol{\theta}}, \boldsymbol{x}_i \in \mathcal{D}_{-\hat{y}})}{|\mathcal{D}_{-\hat{y}}|}. \quad (4)$$

To bridge the adversarial examples and poison samples, we provide the following assumption:

**Assumption 1.** *Assume that $g(\boldsymbol{x}; \Delta) - \boldsymbol{x} \in \mathcal{S}$ for $\forall \boldsymbol{x} \in \mathcal{D}_{cl}$.*

Assumption 1 ensures that there exists $\boldsymbol{\epsilon} \in \mathcal{S}$ such that $\boldsymbol{x} + \boldsymbol{\epsilon} = g(\boldsymbol{x}; \Delta)$. Note that **the analysis and the proposed method are independent of any specific types of perturbation**. More discussions about the choice of $\mathcal{S}$ and its influence on the proposed method are discussed in **Appendix B.4**.

Then, we reach the first upper bound of backdoor risk:

**Proposition 1.** *Under Assumption 1, $\mathcal{R}_{adv}$ serves as an upper bound of $\mathcal{R}_{bd}$, i.e., $\mathcal{R}_{bd} \leq \mathcal{R}_{adv}$.*

**Remark.** Although $\mathcal{R}_{adv}$ seems to be a promising surrogate for $\mathcal{R}_{bd}$, it neglects some essential connections between adversarial examples and poisoned samples, resulting in a significant gap between adversarial risk and backdoor risk, as shown in Figure 1. *Such a gap also leads to a fluctuating learning curve for backdoor risk and a significant drop in accuracy in the backdoor mitigation process by adversarial training.* Therefore, we seek to construct a tighter upper bound for $\mathcal{R}_{bd}$. A key insight is that not all adversarial examples contribute to backdoor mitigation.

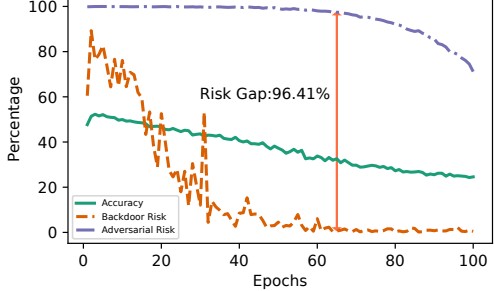

Figure 1: Example of purifying poisoned model using adversarial training on Tiny ImageNet [26]. The curves for Accuracy, Backdoor Risk, and Adversarial Risk are indicated by Green, Orange, and Purple, respectively.

To further identify AEs important for mitigating backdoors, we leverage the information from the poisoned model to categorize the adversarial examples to $h_{\boldsymbol{\theta}}$ into three types, as shown in Table 1.

Table 1: Different types of adversarial examples to $h_{\boldsymbol{\theta}}$.

| Type | Description | Definition |
|------|-------------|------------|
| I | Mislead $h_{\boldsymbol{\theta}}$ and $h_{\boldsymbol{\theta}_{bd}}$ to the same class | $h_{\boldsymbol{\theta}_{bd}}(\tilde{\boldsymbol{x}}_{\boldsymbol{\epsilon}}) = h_{\boldsymbol{\theta}}(\tilde{\boldsymbol{x}}_{\boldsymbol{\epsilon}}) \neq y$ |
| II | Mislead $h_{\boldsymbol{\theta}}$, but not mislead $h_{\boldsymbol{\theta}_{bd}}$ | $h_{\boldsymbol{\theta}_{bd}}(\tilde{\boldsymbol{x}}_{\boldsymbol{\epsilon}}) \neq h_{\boldsymbol{\theta}}(\tilde{\boldsymbol{x}}_{\boldsymbol{\epsilon}}), h_{\boldsymbol{\theta}_{bd}}(\tilde{\boldsymbol{x}}_{\boldsymbol{\epsilon}}) = y$ |
| III | Mislead $h_{\boldsymbol{\theta}}$ and $h_{\boldsymbol{\theta}_{bd}}$ to different classes | $h_{\boldsymbol{\theta}_{bd}}(\tilde{\boldsymbol{x}}_{\boldsymbol{\epsilon}}) \neq h_{\boldsymbol{\theta}}(\tilde{\boldsymbol{x}}_{\boldsymbol{\epsilon}}), h_{\boldsymbol{\theta}_{bd}}(\tilde{\boldsymbol{x}}_{\boldsymbol{\epsilon}}) \neq y, h_{\boldsymbol{\theta}}(\tilde{\boldsymbol{x}}_{\boldsymbol{\epsilon}}) \neq y$ |

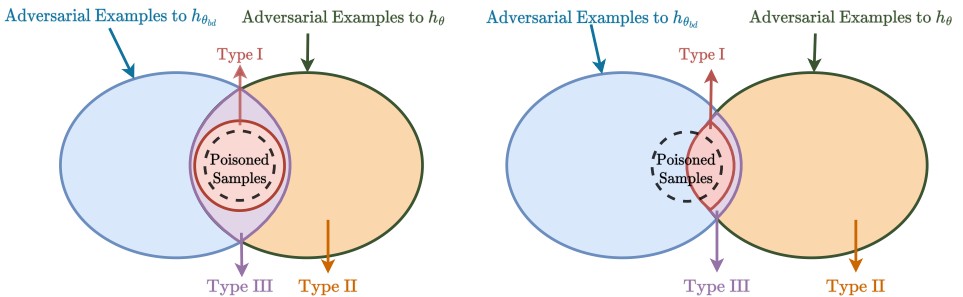

Figure 2: A schematic of the relationship between adversarial examples, shared adversarial examples (SAEs, Type I) and poisoned samples. The adversarial examples for $h_{\boldsymbol{\theta}_{bd}}$ and $h_{\boldsymbol{\theta}}$ are shown in the blue and green solid ellipses, respectively. The poisoned samples are in the black dashed circle. Assume that $h_{\boldsymbol{\theta}_{bd}}$ and $h_{\boldsymbol{\theta}}$ have $100\%$ backdoor attack success rates in the left, such that all poisoned samples are contained in SAEs. Thus, by reducing the shared adversarial examples between $h_{\boldsymbol{\theta}_{bd}}$ and $h_{\boldsymbol{\theta}}$ (from left to right), the backdoor risk can be mitigated in $h_{\boldsymbol{\theta}}$.

Furthermore, we refer adversarial examples of Type I as the shared adversarial examples between $h_{\boldsymbol{\theta}}$ and $h_{\boldsymbol{\theta}_{bd}}$, as defined below:

**Definition 1** (Shared Adversarial Example). *Given two classifiers $h_{\boldsymbol{\theta}_1}$ and $h_{\boldsymbol{\theta}_2}$, an adversarial example $\tilde{\boldsymbol{x}}_{\boldsymbol{\epsilon}}$ is shared between $h_{\boldsymbol{\theta}_1}$ and $h_{\boldsymbol{\theta}_2}$ if and only if $h_{\boldsymbol{\theta}_1}(\tilde{\boldsymbol{x}}_{\boldsymbol{\epsilon}}) = h_{\boldsymbol{\theta}_2}(\tilde{\boldsymbol{x}}_{\boldsymbol{\epsilon}}) \neq y$.*

Let $\mathcal{D}_s$ be the subset of samples in $\mathcal{D}_{-\hat{y}}$ on which planting a trigger can successfully activate the backdoor in $h_{\boldsymbol{\theta}_{bd}}$, as summarized in Table 2. As illustrated in Figure 2, the following proposition reveals a deeper connection between adversarial examples and poisoned samples.

**Proposition 2.** *For $\forall(\boldsymbol{x}, y) \in \mathcal{D}_s$, the poisoned sample $g(\boldsymbol{x}; \Delta)$ can attack $h_{\boldsymbol{\theta}}$ if and only if $h_{\boldsymbol{\theta}}(g(\boldsymbol{x}; \Delta)) = h_{\boldsymbol{\theta}_{bd}}(g(\boldsymbol{x}; \Delta)) \neq y$. Furthermore, under Assumption 1, the poisoned sample $g(\boldsymbol{x}; \Delta)$ is a shared adversarial example between $h_{\boldsymbol{\theta}}$ and $h_{\boldsymbol{\theta}_{bd}}$, if it can attack $h_{\boldsymbol{\theta}}$.*

Table 2: Table of notations.

| Notation | Description/Definition |
|----------|------------------------|
| $h_{\boldsymbol{\theta}_{bd}}$ | Poisoned model |
| $h_{\boldsymbol{\theta}}$ | Purified model |
| $\Delta$ | Trigger |
| $\hat{y}$ | Target label |
| $g(\boldsymbol{x}, \Delta)$ | Poisoned sample generated from $\boldsymbol{x}$ |
| $\mathcal{S}$ | Set of perturbations |
| $\boldsymbol{\epsilon}$ | Perturbation |
| $\tilde{\boldsymbol{x}}_{\boldsymbol{\epsilon}}$ | Adversarial example, $\boldsymbol{x} + \boldsymbol{\epsilon}$ |
| $\mathcal{D}_{cl}$ | $\{(\boldsymbol{x}_i, y_i)\}_{i=1}^N$ |
| $\mathcal{D}_{-\hat{y}}$ | $\{(\boldsymbol{x}, y)\|(\boldsymbol{x}, y) \in \mathcal{D}_{cl}, y \neq \hat{y}\}$ |
| $\mathcal{D}_s$ | $\{(\boldsymbol{x}, y)\|h_{\boldsymbol{\theta}_{bd}}(g(\boldsymbol{x}; \Delta)) = \hat{y}, \boldsymbol{x} \in \mathcal{D}_{-\hat{y}}\}$ |
| $\mathcal{A}_{\boldsymbol{\epsilon}, \boldsymbol{\theta}}$ | $\{(\tilde{\boldsymbol{x}}_{\boldsymbol{\epsilon}}, y)\|h_{\boldsymbol{\theta}}(\tilde{\boldsymbol{x}}_{\boldsymbol{\epsilon}}) \neq y, (\boldsymbol{x}, y) \in \mathcal{D}_{cl}\}$ |

Then, the adversarial risk $\mathcal{R}_{adv}$ can be decomposed to three components as below:

$$\mathcal{R}_{adv}(h_{\boldsymbol{\theta}}) = \frac{1}{|\mathcal{D}_{-\hat{y}}|} \sum_{i=1}^N \max_{\boldsymbol{\epsilon}_i \in \mathcal{S}} \Big\{ \underbrace{\mathbb{I}(\boldsymbol{x}_i \in \mathcal{D}_s)\mathbb{I}(h_{\boldsymbol{\theta}}(\tilde{\boldsymbol{x}}_{i,\boldsymbol{\epsilon}_i}) = h_{\boldsymbol{\theta}_{bd}}(\tilde{\boldsymbol{x}}_{i,\boldsymbol{\epsilon}_i}), \tilde{\boldsymbol{x}}_{i,\boldsymbol{\epsilon}_i} \in \mathcal{A}_{\boldsymbol{\epsilon}_i, \boldsymbol{\theta}})}_{\text{Type I adversarial examples on } \mathcal{D}_s}$$

$$+ \underbrace{\mathbb{I}(\boldsymbol{x}_i \in \mathcal{D}_s)\mathbb{I}(h_{\boldsymbol{\theta}}(\tilde{\boldsymbol{x}}_{i,\boldsymbol{\epsilon}_i}) \neq h_{\boldsymbol{\theta}_{bd}}(\tilde{\boldsymbol{x}}_{i,\boldsymbol{\epsilon}_i}), \tilde{\boldsymbol{x}}_{i,\boldsymbol{\epsilon}_i} \in \mathcal{A}_{\boldsymbol{\epsilon}_i, \boldsymbol{\theta}})}_{\text{Type II or III adversarial examples on } \mathcal{D}_s} + \underbrace{\mathbb{I}(\boldsymbol{x}_i \in \mathcal{D}_{-\hat{y}} \setminus \mathcal{D}_s, \tilde{\boldsymbol{x}}_{i,\boldsymbol{\epsilon}_i} \in \mathcal{A}_{\boldsymbol{\epsilon}_i, \boldsymbol{\theta}})}_{\text{vanilla adversarial examples on } \mathcal{D}_{-\hat{y}} \setminus \mathcal{D}_s} \Big\}.$$

$$(5)$$

Proposition 2 implies that the first component in (5) is essential for backdoor mitigation, as it captures the risk of shared adversarial examples between the poisoned model $h_{\boldsymbol{\theta}_{bd}}$ and the fine-tuned model $h_{\boldsymbol{\theta}}$, and the poisoned samples effective for $h_{\boldsymbol{\theta}}$ belong to SAEs. The second component is irrelevant

for backdoor mitigation, since adversarial examples of Type II and III are either not poisoned samples or ineffective to backdoor attack against $h_{\boldsymbol{\theta}}$. The third component protects the samples that are originally resistant to triggers from being backdoor attacked in the mitigation process.

Thus, by removing the second component in (5), we propose the following sub-adversarial risk

$$
\begin{aligned}
\mathcal{R}_{sub}(h_{\boldsymbol{\theta}}) =& \frac{1}{|\mathcal{D}_{-\hat{y}}|} \sum_{i=1}^{N} \max_{\boldsymbol{\epsilon}_i \in \mathcal{S}} \Big\{ \mathbb{I}(\boldsymbol{x}_i \in \mathcal{D}_s) \mathbb{I}(h_{\boldsymbol{\theta}}(\tilde{\boldsymbol{x}}_{i,\boldsymbol{\epsilon}_i}) = h_{\boldsymbol{\theta}_{bd}}(\tilde{\boldsymbol{x}}_{i,\boldsymbol{\epsilon}_i}), \tilde{\boldsymbol{x}}_{i,\boldsymbol{\epsilon}_i} \in \mathcal{A}_{\boldsymbol{\epsilon}_i,\boldsymbol{\theta}}) \\
&+ \mathbb{I}(\boldsymbol{x}_i \in \mathcal{D}_{-\hat{y}} \setminus \mathcal{D}_s, \tilde{\boldsymbol{x}}_{i,\boldsymbol{\epsilon}_i} \in \mathcal{A}_{\boldsymbol{\epsilon}_i,\boldsymbol{\theta}}) \Big\}.
\end{aligned}
\tag{6}
$$

Compared to vanilla adversarial risk $\mathcal{R}_{adv}$, the proposed sub-adversarial risk $\mathcal{R}_{sub}$ focuses on the shared adversarial examples for samples in $\mathcal{D}_s$ while considering vanilla adversarial examples for samples in $\mathcal{D}_{-\hat{y}} \setminus \mathcal{D}_s$. When $\mathcal{D}_s = \mathcal{D}_{-\hat{y}}$, i.e., $h_{\boldsymbol{\theta}_{bd}}$ has backdoor risk 100%, the sub-adversarial risk measures the shared adversarial risk on $\mathcal{D}_{-\hat{y}}$.

Then, we provide the following proposition to establish the relation between backdoor risk and (sub) adversarial risk:

**Proposition 3.** *Under Assumption 1, for a classifier $h_{\boldsymbol{\theta}}$, the following inequalities hold*

$$
\mathcal{R}_{bd}(h_{\boldsymbol{\theta}}) \leq \mathcal{R}_{sub}(h_{\boldsymbol{\theta}}) \leq \mathcal{R}_{adv}(h_{\boldsymbol{\theta}}).
$$

Therefore, $\mathcal{R}_{sub}$ is a tighter upper bound for $\mathcal{R}_{bd}$ compared with $\mathcal{R}_{adv}$. After replacing $\mathcal{R}_{bd}$ with $\mathcal{R}_{sub}$ in (3), we reach the following optimization problem:

$$
\min_{\boldsymbol{\theta}} \mathcal{R}_{cl}(h_{\boldsymbol{\theta}}) + \lambda \mathcal{R}_{sub}(h_{\boldsymbol{\theta}}).
\tag{7}
$$

Due to the space limit, the proofs of the above propositions, as well as some detailed discussions and extensions will be provided in **Appendix A**.

### 3.3 Proposed method

Since the target label $\hat{y}$ is unavailable and 0-1 loss is non-differentiable, we first discuss the relaxation of $\mathcal{R}_{sub}$ and then replace 0-1 loss with suitable surrogate loss functions in this section.

**Relaxation.** One limitation of $\mathcal{R}_{sub}$ is the inaccessibility of $\mathcal{D}_s$, i.e., the subset of samples in $\mathcal{D}_{-\hat{y}}$ on which planting a trigger can successfully activate the backdoor in $h_{\boldsymbol{\theta}_{bd}}$. Since a poisoned model usually has a high attack success rate (ASR), the first relaxation is replacing $\mathcal{D}_s$ with $\mathcal{D}_{-\hat{y}}$, i.e.,

$$
\mathcal{R}_{sub}(h_{\boldsymbol{\theta}}) \approx \frac{1}{|\mathcal{D}_{-\hat{y}}|} \sum_{i=1}^{N} \max_{\boldsymbol{\epsilon}_i \in \mathcal{S}} \Big\{ \mathbb{I}(\boldsymbol{x}_i \in \mathcal{D}_{-\hat{y}}) \mathbb{I}(h_{\boldsymbol{\theta}}(\tilde{\boldsymbol{x}}_{i,\boldsymbol{\epsilon}_i}) = h_{\boldsymbol{\theta}_{bd}}(\tilde{\boldsymbol{x}}_{i,\boldsymbol{\epsilon}_i}), \tilde{\boldsymbol{x}}_{i,\boldsymbol{\epsilon}_i} \in \mathcal{A}_{\boldsymbol{\epsilon}_i,\boldsymbol{\theta}}) \Big\}.
$$

Since $\hat{y}$ is unavailable, we also replace $\mathcal{D}_{-\hat{y}}$ by $\mathcal{D}_{cl}$ in the sub-adversarial risk. As $g(\boldsymbol{x}; \Delta)$ is not malicious for classification if the ground truth label of $\boldsymbol{x}$ is $\hat{y}$, this relaxation has negligible negative influence for mitigating backdoor risk. After the above two relaxations, we reach the following shared adversarial risk on $\mathcal{D}_{cl}$:

$$
\mathcal{R}_{share}(h_{\boldsymbol{\theta}}) = \frac{1}{N} \sum_{i=1}^{N} \max_{\boldsymbol{\epsilon}_i \in \mathcal{S}} \Big\{ \mathbb{I}(h_{\boldsymbol{\theta}}(\tilde{\boldsymbol{x}}_{i,\boldsymbol{\epsilon}_i}) = h_{\boldsymbol{\theta}_{bd}}(\tilde{\boldsymbol{x}}_{i,\boldsymbol{\epsilon}_i}), \tilde{\boldsymbol{x}}_{i,\boldsymbol{\epsilon}_i} \in \mathcal{A}_{\boldsymbol{\epsilon}_i,\boldsymbol{\theta}}) \Big\}.
$$

Due to the space limit, we postpone the detailed discussion of the above two relaxations (e.g., the relaxation gaps) in **Appendix A.7**.

By replacing $\mathcal{R}_{sub}$ with $\mathcal{R}_{share}$ in (7), we reach the following problem:

$$
\min_{\boldsymbol{\theta}} \mathcal{R}_{cl}(h_{\boldsymbol{\theta}}) + \lambda \mathcal{R}_{share}(h_{\boldsymbol{\theta}}).
\tag{8}
$$

By solving (8), we seek $h_{\boldsymbol{\theta}}$ that either classifies the adversarial examples correctly (i.e., $h_{\boldsymbol{\theta}}(\tilde{\boldsymbol{x}}_{\boldsymbol{\epsilon}}) = y$), or differently with $h_{\boldsymbol{\theta}_{bd}}$ (i.e., $h_{\boldsymbol{\theta}}(\tilde{\boldsymbol{x}}_{\boldsymbol{\epsilon}}) \neq h_{\boldsymbol{\theta}_{bd}}(\tilde{\boldsymbol{x}}_{\boldsymbol{\epsilon}})$), while maintaining a high accuracy on $\mathcal{D}_{cl}$.

**Approximation.** We firstly need to approximate two indicators in (8), including: 1) $\mathbb{I}(h_{\boldsymbol{\theta}}(\boldsymbol{x}) \neq y)$; 2) $\mathbb{I}(\tilde{\boldsymbol{x}}_{\boldsymbol{\epsilon}} \in \mathcal{A}_{\boldsymbol{\epsilon},\boldsymbol{\theta}}, h_{\boldsymbol{\theta}}(\tilde{\boldsymbol{x}}_{\boldsymbol{\epsilon}}) = h_{\boldsymbol{\theta}_{bd}}(\tilde{\boldsymbol{x}}_{\boldsymbol{\epsilon}}))$. The former indicator is for the clean accuracy, so we use the widely used cross-entropy (CE) loss as the surrogate loss, $i.e.$, $L_{cl}(\boldsymbol{x}, y; \boldsymbol{\theta}) = \text{CE}(\boldsymbol{p}(\boldsymbol{x}; \boldsymbol{\theta}), y)$. In terms of the latter indicator that measures the shared adversarial risk, different surrogate losses could be employed for different usages:

- To generate shared adversarial examples, we first decompose the second indicator as below:

$$\mathbb{I}(\tilde{\boldsymbol{x}}_{\boldsymbol{\epsilon}} \in \mathcal{A}_{\boldsymbol{\epsilon},\boldsymbol{\theta}} \cap \mathcal{A}_{\boldsymbol{\epsilon},\boldsymbol{\theta}_{bd}})\mathbb{I}(h_{\boldsymbol{\theta}}(\tilde{\boldsymbol{x}}_{\boldsymbol{\epsilon}}) = h_{\boldsymbol{\theta}_{bd}}(\tilde{\boldsymbol{x}}_{\boldsymbol{\epsilon}})), \tag{9}$$

which covers two components standing for fooling $h_{\boldsymbol{\theta}_{bd}}$ and $h_{\boldsymbol{\theta}}$ simultaneously, and keeping the same predicted label for $h_{\boldsymbol{\theta}_{bd}}$ and $h_{\boldsymbol{\theta}}$, respectively.

For the first component, we use CE on both $h_{\boldsymbol{\theta}}$ and $h_{\boldsymbol{\theta}_{bd}}$ as a surrogate loss, $i.e.$,

$$L_{adv}(\tilde{\boldsymbol{x}}_{\boldsymbol{\epsilon}}, y; \boldsymbol{\theta}) = \frac{1}{2}\left(\text{CE}(\boldsymbol{p}(\tilde{\boldsymbol{x}}_{\boldsymbol{\epsilon}}; \boldsymbol{\theta}), y) + \text{CE}(\boldsymbol{p}(\tilde{\boldsymbol{x}}_{\boldsymbol{\epsilon}}; \boldsymbol{\theta}_{bd}), y)\right). \tag{10}$$

For the second component, we measure the distance between $\boldsymbol{p}(\tilde{\boldsymbol{x}}_{i,\epsilon_i}; \boldsymbol{\theta})$ and $\boldsymbol{p}(\tilde{\boldsymbol{x}}_{\boldsymbol{\epsilon}}; \boldsymbol{\theta}_{bd})$ by Jensen-Shannon divergence [11] and adopt the following surrogate loss:

$$L_{share}(\tilde{\boldsymbol{x}}_{\boldsymbol{\epsilon}}, y; \boldsymbol{\theta}) = -\text{JS}(\boldsymbol{p}(\tilde{\boldsymbol{x}}_{\boldsymbol{\epsilon}}; \boldsymbol{\theta}), \boldsymbol{p}(\tilde{\boldsymbol{x}}_{\boldsymbol{\epsilon}}; \boldsymbol{\theta}_{bd})). \tag{11}$$

- To unlearn the shared adversarial examples, we adopt the following surrogate loss

$$L_{sar}(\tilde{\boldsymbol{x}}_{\boldsymbol{\epsilon}}, y; \boldsymbol{\theta}) = -\mathbb{I}(\tilde{y} \neq y) \log\left(1 - \boldsymbol{p}_{\tilde{y}}(\tilde{\boldsymbol{x}}_{\boldsymbol{\epsilon}}; \boldsymbol{\theta})\right), \tag{12}$$

where $\tilde{y} = h_{\boldsymbol{\theta}_{bd}}(\tilde{\boldsymbol{x}}_{\boldsymbol{\epsilon}})$. By reducing $L_{sar}$, the prediction of $\tilde{\boldsymbol{x}}_{\boldsymbol{\epsilon}}$ by $h_{\boldsymbol{\theta}}$, $i.e.$, $h_{\boldsymbol{\theta}}(\tilde{\boldsymbol{x}}_{\boldsymbol{\epsilon}})$ is forced to differ from $h_{\boldsymbol{\theta}_{bd}}(\tilde{\boldsymbol{x}}_{\boldsymbol{\epsilon}})$ if $h_{\boldsymbol{\theta}_{bd}}(\tilde{\boldsymbol{x}}_{\boldsymbol{\epsilon}}) \neq y$, therefore, reducing the shared adversarial risk (SAR).

**Overall objective.** By combining all the above surrogate losses with the linear weighted summation mode, we propose the following bi-level objective:

$$
\begin{aligned}
\min_{\boldsymbol{\theta}} \quad & \frac{1}{N}\sum_{i=1}^{N}\left\{\lambda_1 L_{cl}(\boldsymbol{x}_i, y_i; \boldsymbol{\theta}) + \lambda_2 L_{sar}(\tilde{\boldsymbol{x}}_{i,\epsilon_i^*}, y_i; \boldsymbol{\theta})\right\} \\
\text{s.t.} \quad & \boldsymbol{\epsilon}_i^* = \arg\max_{\boldsymbol{\epsilon}_i \in \mathcal{S}} \lambda_3 L_{adv}(\tilde{\boldsymbol{x}}_{i,\epsilon_i}, y_i; \boldsymbol{\theta}) + \lambda_4 L_{share}(\tilde{\boldsymbol{x}}_{i,\epsilon_i}, y_i; \boldsymbol{\theta}), \quad i = 1, \ldots, N,
\end{aligned}
\tag{13}
$$

where $\lambda_1, \lambda_2, \lambda_3, \lambda_4 \geq 0$ indicate trade-off weights.

**Optimization algorithm.** We propose an optimization algorithm called **Shared Adversarial Unlearning** (SAU, Algorithm 1), which solves (13) by alternatively updating $\boldsymbol{\theta}$, and $\boldsymbol{\epsilon}^*$. Specifically, the model parameter $\boldsymbol{\theta}$ is updated using stochastic gradient descent [3], and the perturbation $\boldsymbol{\epsilon}^*$ is generated by projected gradient descent [33].

---

**Algorithm 1** Shared Adversarial Unlearning

---

**Input:** Training set $\mathcal{D}_{cl}$, poisoned model $h_{\boldsymbol{\theta}_{bd}}$, PGD step size $\eta > 0$, number of PGD steps $T_{adv}$, perturbation set $\mathcal{S}$, max iteration number $T$.
Initialize $h_{\boldsymbol{\theta}} = h_{\boldsymbol{\theta}_{bd}}$.
**for** $t = 0, ..., T - 1$ **do**
    **for** Each mini-batch in $\mathcal{D}_{cl}$ **do**
        Initialize perturbation $\boldsymbol{\epsilon}$.
        **for** $t_{adv} = 0, ..., T_{adv} - 1$ **do**
            Compute gradient $\boldsymbol{g}$ of $\boldsymbol{\epsilon}$ *w.r.t.* the inner maximization objective in (13).
            Update $\boldsymbol{\epsilon} = \prod_{\mathcal{S}}(\boldsymbol{\epsilon} + \eta\,\text{sign}(\boldsymbol{g}))$ where $\prod$ is the projection operation.
        **end for**
        Update $\boldsymbol{\theta}$ *w.r.t.* the outer minimization objective in (13).
    **end for**
**end for**

---

# 4 Experiment

## 4.1 Experiment settings

**Backdoor attack.** We compare SAU with 7 popular state-of-the-art (SOTA) backdoor attacks, including BadNets [15], Blended backdoor attack (Blended) [7], input-aware dynamic backdoor attack (Input-Aware)[37], low frequency attack (LF) [58], sinusoidal signal backdoor attack (SIG) [2], sample-specific backdoor attack (SSBA) [30], and warping-based poisoned networks (WaNet) [38]. To make a fair and trustworthy comparison, we use the implementation and configuration from BackdoorBench [54], a comprehensive benchmark for backdoor evaluation. By default, the poisoning ratio is set to 10% in all attacks, and the $0^{th}$ label is set to be the target label. We evaluate all the attacks on 3 benchmark datasets, CIFAR-10 [24], Tiny ImageNet [26], and GTSRB [43] using two networks: PreAct-ResNet18 [18] and VGG19 [41]. Due to space constraints, the results for GTSRB and VGG19 are postponed to **Appendix D**. Note that for clean label attack SIG, the 10% poisoning ratio can only be implemented for CIFAR-10. More attack details are left in **Appendix C**.

**Backdoor defense.** We compare our method with 6 SOTA backdoor defense methods: ANP [55], Fine-pruning (FP) [31], NAD [28], NC [47], EP [64] and i-BAU [59]. By default, all the defense methods can access 5% benign training data. We follow the recommended configurations for SOTA defenses as in BackdoorBench [54]. For our method, we choose to generate the shared adversarial example with Projected Gradient Descent (PGD) [33] with $L_\infty$ norm. For all experiments, we run PGD 5 steps with norm bound 0.2 and we set $\lambda_1 = \lambda_2 = \lambda_4 = 1$ and $\lambda_3 = 0.01$. More details about defense settings and additional experiments can be found in **Appendix C and D**.

**Evaluation metric.** We use four metrics to evaluate the performance of each defense method: Accuracy on benign data (**ACC**), Attack Success Rate (**ASR**), Robust Accuracy (**R-ACC**) and Defense Effectiveness Rating (**DER**). R-ACC measures the proportion of the poisoned samples classified to their true label, and ASR measures the proportion of poisoned samples misclassified to the target label. Larger R-ACC and lower ASR indicate that the backdoor is effectively mitigated. Note that the samples for the target class are excluded when computing the ASR and R-ACC as done in BackdoorBench. DER $\in [0, 1]$ was proposed in [65] to evaluate the cost of ACC for reducing ASR. It is defined as follows:

$$\text{DER} = [\max(0, \Delta\text{ASR}) - \max(0, \Delta\text{ACC}) + 1]/2, \tag{14}$$

where $\Delta$ASR denotes the drop in ASR after applying defense, and $\Delta$ACC represents the drop in ACC after applying defense.

**Note**: Higher ACC, lower ASR, higher R-ACC and higher DER represent better defense performance. We use **boldface** and underline to indicate the best and the second-best results among all defense methods, respectively, in later experimental results.

## 4.2 Main results

**Effectiveness of SAU.** To verify the effectiveness of SAU, we first summarize the experimental results on CIFAR-10 and Tiny ImageNet in Tables 3 and 4, respectively. As shown in Tables 3-4, SAU can mitigate backdoor for almost all attacks with a significantly lower average ASR. For the experiments on CIFAR-10, SAU achieves the top-2 lowest ASR in 4 of 7 attacks and very low ASR for the other 3 attacks. Similarly, SAU performs the lowest ASR in 3 attacks for Tiny ImageNet and negligible ASR in two of the other attacks. Notably, SAU fails to mitigate WaNet in Tiny ImageNet, although it can defend against WaNet on CIFAR-10. As a transformation-based attack that applies image transforms to construct poisoned samples, WaNet can generate triggers with a large $L_\infty$ norm. Specifically, WaNet has average $L_\infty$ trigger norm 0.348 on TinyImageNet and 0.172 on CIFAR-10. Note that for all experiments, we generate adversarial perturbation with $L_\infty$ norm less than 0.2. The large trigger norm of WaNet on Tiny ImageNet poses a challenge for AT-based methods such as i-BAU and the proposed one, which reveals an important weakness of such methods, *i.e.*, their performance may be degraded if the trigger is beyond the perturbation set. A promising approach for this challenge is to consider adversarial examples for the union of multiple perturbation types [34, 45] and we postpone more discussion to **Appendix B**.

As maintaining clean accuracy is also important for an effective backdoor defense method, we also compare SAU with other baselines in terms of ACC, R-ACC and DER in Table 3 and 4. As SAU

Table 3: Results on CIFAR-10 with PreAct-ResNet18 and poisoning ratio 10%.

| Defense | No Defense | | | | ANP [55] | | | | FP [31] | | | | NC [47] | | | |
|---|---|---|---|---|---|---|---|---|---|---|---|---|---|---|---|---|
| Attack | ACC | ASR | R-ACC | DER | ACC | ASR | R-ACC | DER | ACC | ASR | R-ACC | DER | ACC | ASR | R-ACC | DER |
| BadNets [15] | 91.32 | 95.03 | 4.67 | N/A | 90.88 | 4.88 | 87.22 | 94.86 | 91.31 | 57.13 | 41.62 | 68.95 | 89.05 | 1.27 | 89.16 | 95.75 |
| Blended [7] | 93.47 | 99.92 | 0.08 | N/A | 92.97 | 84.88 | 13.36 | 57.27 | 93.17 | 99.26 | 0.73 | 50.18 | 93.47 | 99.92 | 0.08 | 50.00 |
| Input-Aware [37] | 90.67 | 98.26 | 1.66 | N/A | 91.04 | 1.32 | 86.71 | 98.47 | 91.74 | 0.04 | 44.54 | 99.11 | 92.61 | 0.76 | 90.87 | 98.75 |
| LF [58] | 93.19 | 99.28 | 0.71 | N/A | 92.64 | 39.99 | 43.03 | 79.37 | 92.90 | 98.97 | 1.02 | 50.01 | 91.62 | 1.41 | 87.48 | 98.15 |
| SIG [2] | 84.48 | 98.27 | 1.72 | N/A | 83.36 | 36.42 | 43.67 | 80.36 | 89.10 | 26.20 | 20.61 | 86.03 | 84.48 | 98.27 | 1.72 | 50.00 |
| SSBA [30] | 92.88 | 97.86 | 1.99 | N/A | 92.62 | 60.17 | 36.69 | 68.71 | 92.54 | 83.50 | 15.36 | 57.01 | 90.99 | 0.58 | 87.04 | 97.69 |
| WaNet [38] | 91.25 | 89.73 | 9.76 | N/A | 91.33 | 2.22 | 88.54 | 93.76 | 91.46 | 1.09 | 69.73 | 94.32 | 91.80 | 7.53 | 85.09 | 91.10 |
| Average | 91.04 | 96.91 | 2.94 | N/A | 90.69 | 32.84 | 58.75 | 81.83 | 91.75 | 52.31 | 27.66 | 72.93 | 90.57 | 29.96 | 63.06 | 83.06 |

| Defense | NAD [28] | | | | EP [64] | | | | i-BAU [59] | | | | SAU (Ours) | | | |
|---|---|---|---|---|---|---|---|---|---|---|---|---|---|---|---|---|
| Attack | ACC | ASR | R-ACC | DER | ACC | ASR | R-ACC | DER | ACC | ASR | R-ACC | DER | ACC | ASR | R-ACC | DER |
| BadNets [15] | 89.87 | 2.14 | 88.71 | 95.72 | 89.66 | 1.88 | 89.51 | 95.75 | 89.15 | 1.21 | 88.88 | 95.83 | 89.31 | 1.53 | 88.81 | 95.74 |
| Blended [7] | 92.17 | 97.69 | 2.14 | 50.47 | 92.43 | 52.13 | 37.52 | 73.37 | 88.66 | 13.99 | 53.23 | 90.56 | 90.96 | 6.14 | 64.89 | 95.63 |
| Input-Aware [37] | 93.18 | 1.68 | 91.12 | 98.29 | 89.86 | 2.23 | 85.20 | 97.61 | 90.29 | 63.36 | 32.70 | 67.26 | 91.59 | 1.27 | 88.54 | 98.49 |
| LF [58] | 92.37 | 47.83 | 47.49 | 75.31 | 91.82 | 85.98 | 12.77 | 55.97 | 89.09 | 21.83 | 64.37 | 86.67 | 90.32 | 4.18 | 81.54 | 96.12 |
| SIG [2] | 90.02 | 10.66 | 64.20 | 93.81 | 83.1 | 0.26 | 56.68 | 98.32 | 85.85 | 1.28 | 55.19 | 98.49 | 90.56 | 1.67 | 57.96 | 98.30 |
| SSBA [30] | 91.91 | 77.4 | 20.86 | 59.74 | 92.33 | 10.67 | 78.60 | 93.32 | 88.15 | 2.17 | 77.28 | 95.48 | 90.84 | 1.79 | 85.83 | 97.01 |
| WaNet [38] | 93.17 | 22.98 | 72.69 | 83.38 | 90.09 | 86.64 | 12.54 | 50.96 | 90.91 | 3.37 | 89.10 | 93.01 | 91.26 | 1.02 | 90.28 | 94.36 |
| Average | 91.81 | 37.2 | 55.32 | 79.53 | 89.9 | 34.26 | 53.26 | 80.76 | 88.87 | 15.32 | 65.82 | 89.61 | 90.41 | 2.51 | 79.69 | 96.52 |

Table 4: Results on Tiny ImageNet with PreAct-ResNet18 and poisoning ratio 10%.

| Defense | No Defense | | | | ANP [55] | | | | FP [31] | | | | NC [47] | | | |
|---|---|---|---|---|---|---|---|---|---|---|---|---|---|---|---|---|
| Attack | ACC | ASR | R-ACC | DER | ACC | ASR | R-ACC | DER | ACC | ASR | R-ACC | DER | ACC | ASR | R-ACC | DER |
| BadNets [15] | 56.23 | 100.0 | 0.0 | N/A | 43.45 | 0.00 | 43.13 | 93.61 | 51.73 | 99.99 | 0.01 | 47.76 | 51.52 | 0.10 | 50.82 | 97.59 |
| Blended [7] | 56.03 | 99.71 | 0.22 | N/A | 43.93 | 6.11 | 17.28 | 90.75 | 51.89 | 95.94 | 2.1 | 49.81 | 52.55 | 93.21 | 3.96 | 51.51 |
| Input-Aware [37] | 57.45 | 98.85 | 1.06 | N/A | 35.39 | 0.00 | 11.82 | 88.40 | 57.25 | 62.92 | 24.7 | 66.88 | 56.20 | 0.09 | 52.19 | 98.76 |
| LF [58] | 55.97 | 98.57 | 0.97 | N/A | 45.69 | 62.30 | 14.49 | 63.00 | 51.44 | 95.25 | 2.42 | 49.4 | 52.99 | 85.56 | 8.07 | 55.02 |
| SSBA [30] | 55.22 | 97.71 | 1.68 | N/A | 43.36 | 56.53 | 17.24 | 64.66 | 50.47 | 88.87 | 6.26 | 52.04 | 52.47 | 53.47 | 23.17 | 70.75 |
| WaNet [38] | 56.78 | 99.49 | 0.36 | N/A | 36.16 | 0.07 | 31.97 | 89.40 | 53.84 | 3.94 | 2.38 | 96.3 | 53.33 | 0.23 | 51.63 | 97.90 |
| Average | 56.28 | 99.06 | 0.72 | N/A | 41.33 | 20.83 | 22.66 | 77.12 | 52.44 | 74.48 | 6.31 | 58.88 | 53.18 | 38.78 | 31.64 | 74.50 |

| Defense | NAD [28] | | | | EP [64] | | | | i-BAU [59] | | | | SAU (Ours) | | | |
|---|---|---|---|---|---|---|---|---|---|---|---|---|---|---|---|---|
| Attack | ACC | ASR | R-ACC | DER | ACC | ASR | R-ACC | DER | ACC | ASR | R-ACC | DER | ACC | ASR | R-ACC | DER |
| BadNets [15] | 46.37 | 0.27 | 45.61 | 94.93 | 52.30 | 0.03 | 52.05 | 98.02 | 52.67 | 98.31 | 1.59 | 49.06 | 51.52 | 0.53 | 51.15 | 97.38 |
| Blended [7] | 46.89 | 94.99 | 2.63 | 47.79 | 51.86 | 56.03 | 12.31 | 69.75 | 52.73 | 92.96 | 4.15 | 51.72 | 50.30 | 0.06 | 25.27 | 96.96 |
| Input-Aware [37] | 47.91 | 1.86 | 43.13 | 93.73 | 57.23 | 0.24 | 24.74 | 99.20 | 50.63 | 56.12 | 24.86 | 67.96 | 54.11 | 0.33 | 42.29 | 97.59 |
| LF [58] | 45.45 | 50.49 | 20.21 | 68.78 | 53.33 | 75.41 | 13.06 | 60.26 | 52.77 | 88.04 | 6.82 | 53.67 | 52.65 | 0.97 | 35.87 | 97.14 |
| SSBA [30] | 45.32 | 57.32 | 19.44 | 65.25 | 48.13 | 44.07 | 21.98 | 73.27 | 52.52 | 74.77 | 14.31 | 60.12 | 51.85 | 0.11 | 36.36 | 97.11 |
| WaNet [38] | 46.98 | 0.43 | 43.99 | 94.63 | 56.21 | 0.23 | 55.13 | 99.34 | 54.40 | 94.86 | 3.92 | 51.12 | 54.65 | 85.75 | 10.35 | 55.80 |
| Average | 46.49 | 34.23 | 29.17 | 73.59 | 53.18 | 29.33 | 29.88 | 78.55 | 52.62 | 84.18 | 9.28 | 54.81 | 52.51 | 14.62 | 33.55 | 84.57 |

adopts adversarial examples to mitigate backdoors, it has negative effects on clean accuracy, resulting in a slightly lower clean accuracy compared to the best one. However, SAU achieves the best average R-ACC, which demonstrates its effectiveness in recovering the prediction of poisoned samples, *i.e.*, classifying the poisoned samples correctly. Besides, the best average DER indicates that SAU achieves a significantly better tradeoff between clean accuracy and backdoor risk.

**Influence of poisoning ratio.** To study the influence of the poisoning ratio on the effectiveness of SAU, we test SAU with varying poisoning ratios from 10% to 50%. As shown in Table 5, SAU is still able to achieve remarkable performance even if half of the data is poisoned.

Table 5: Results on CIFAR-10 with PreAct-ResNet18 and different poisoning ratios.

| Poisoning Ratio | | 10% | | 20% | | 30% | | 40% | | 50% | |
|---|---|---|---|---|---|---|---|---|---|---|---|
| Attack | Metric | No Defence | Ours | No Defence | Ours | No Defence | Ours | No Defence | Ours | No Defence | Ours |
| BadNets [15] | ACC | 91.82 | 89.05 | 90.17 | 89.27 | 88.32 | 88.49 | 86.16 | 87.30 | 83.61 | 86.93 |
| | ASR | 93.79 | 1.53 | 96.12 | 1.17 | 97.33 | 1.72 | 97.78 | 1.72 | 98.39 | 2.31 |
| Blended [7] | ACC | 93.69 | 90.96 | 93.00 | 91.02 | 92.78 | 89.49 | 91.64 | 89.64 | 91.26 | 88.92 |
| | ASR | 99.76 | 6.14 | 99.92 | 7.99 | 99.98 | 11.58 | 99.96 | 5.36 | 100.00 | 4.63 |
| LF [58] | ACC | 93.01 | 90.32 | 92.60 | 91.33 | 92.01 | 89.65 | 92.03 | 89.09 | 90.91 | 89.02 |
| | ASR | 99.06 | 4.18 | 99.54 | 1.90 | 99.76 | 2.02 | 99.84 | 1.94 | 99.89 | 1.77 |

### 4.3 Ablation Study

In this section, we conduct ablation study to investigate each component of SAU. Specifically, SAU is composed of two parts: generating shared adversarial examples according to the maximization objective in (13) denoted by **A**, and reducing shared adversarial risk according to the minimization objective in (13), denote by **B**. To study the effect of these parts, we replace them with the corresponding part in vanilla adversarial training, denoted by **C** and **D**, respectively. Then, we run each variant 20 epochs on Tiny ImageNet with PreAct-ResNet18 and a poisoning ratio 10%. Each experiment is repeated five times, and the error bar is only shown in Figure 3 for simplicity.

As shown in Table 6, the maximization objective for generating shared adversarial examples (component **A**) is the key to reducing ASR, and component **B** can help to alleviate the hurt to clean accuracy. When replacing the component **B** with the minimization step in vanilla AT (**D**), it suffers from a drop of clean accuracy, although it can also work for mitigating backdoor effectively. Compared to vanilla AT, SAU achieves significantly lower ASR and higher ACC, with a much more stable learning

Table 6: Results on Tiny ImageNet with different variants of SAU and Vanilla AT.

| Attack | BadNets | | Blended | | LF | |
|---|---|---|---|---|---|---|
| Defense | ACC | ASR | ACC | ASR | ACC | ASR |
| **A+B** (Ours) | 50.60 | 0.27 | 51.47 | 1.02 | 51.31 | 0.72 |
| **A+D** | 37.21 | 0.30 | 38.35 | 1.77 | 36.68 | 0.17 |
| **C+B** | 53.05 | 28.57 | 53.16 | 69.22 | 52.57 | 68.65 |
| **C+D** (Vanilla AT) | 46.66 | 2.48 | 46.39 | 38.03 | 46.69 | 13.68 |

curve, as shown in Figure 3. We remark that the decrease of clean accuracy for **A+D** and vanilla AT demonstrates that imposing a model to learn overwhelming adversarial examples may severely hurt the clean accuracy, a phenomenon widely studied in the area of adversarial training [46, 60].

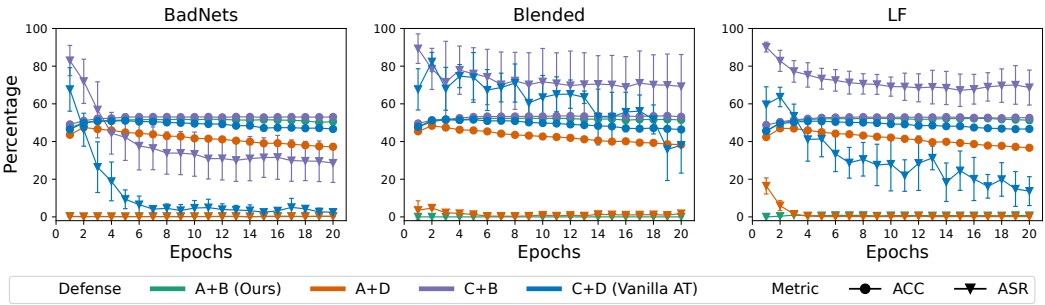

Figure 3: Learn curves for SAU, Vanilla AT and their variants, averaged over five runs.

## 5 Conclusion

In conclusion, this paper proposes Shared Adversarial Unlearning, a method to defend against backdoor attacks in deep neural networks through adversarial training techniques. By developing a better understanding of the connection between adversarial examples and poisoned samples, we propose a novel upper bound for backdoor risk and a bi-level formulation for mitigating backdoor attacks in poisoned models. Our approach identifies shared adversarial examples and unlearns them to break the connection between the poisoned sample and the target label. We demonstrate the effectiveness of our proposed method through extensive experiments, showing that it achieves comparable to, and even superior, performance than six different state-of-the-art defense methods on seven SOTA backdoor attacks with different model structures and datasets. Our work provides a valuable contribution to the field of backdoor defense in deep neural networks, with potential applications in various domains.

**Limitation and future work.** One important direction for future work, and a current challenge, is the accessibility of clean samples. A valuable approach to this challenge is to consider the samples from other domains and the generated samples.

**Structure of Appendix.** The detailed proof, analysis, and extension of propositions are given in **Appendix** A. Some important discussions such as the computation cost, the scalability, and the generalization ability of the proposed method are provided in **Appendix** B. The details of the experiments are provided in **Appendix** C. Additional experiments on different settings such as model structures, poisoning ratios, and sizes of clean datasets are given in **Appendix** D.

## Acknowledgments and Disclosure of Funding

This work is supported by the National Natural Science Foundation of China under grant No. 62076213, Shenzhen Science and Technology Program under grant No. RCYX20210609103057050, No. GXWD20201231105722002-20200901175001001, and No. ZDSYS20211021111415025, and No. JCYJ20210324120011032 and the CAAI-Huawei MindSpore Open Fund, and the Guangdong Provincial Key Laboratory of Big Data Computing, the Chinese University of Hong Kong, Shenzhen.

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

# A Proofs and extension

In this section, we provide detailed proof and analysis of the propositions, methods, and extensions.

## A.1 Proof of Proposition 1

*Proof.* Under Assumption 1, for any sample $\boldsymbol{x} \in \mathcal{D}_{-\hat{y}}$, we have

$$\mathbb{I}(h_{\boldsymbol{\theta}}(g(\boldsymbol{x}; \Delta)) = \hat{y}) \leq \mathbb{I}(g(\boldsymbol{x}; \Delta) \in \mathcal{A}_{g(\boldsymbol{x}; \Delta) - \boldsymbol{x}, \boldsymbol{\theta}}) \leq \max_{\boldsymbol{\epsilon} \in \mathcal{S}} \mathbb{I}(\tilde{\boldsymbol{x}}_{\boldsymbol{\epsilon}} \in \mathcal{A}_{\boldsymbol{\epsilon}, \boldsymbol{\theta}}),$$

since $g(\boldsymbol{x}; \Delta) - \boldsymbol{x} \in \mathcal{S}$.

Therefore, we have

$$\mathcal{R}_{bd}(h_{\boldsymbol{\theta}}) \leq \mathcal{R}_{adv}(h_{\boldsymbol{\theta}}).$$

$\square$

## A.2 Proof of Proposition 2

*Proof.* By the definition of backdoor attack, a poisoned sample $g(\boldsymbol{x}; \Delta)$ is said to attack $h_{\boldsymbol{\theta}}$ successfully if and only if $h_{\boldsymbol{\theta}}(g(\boldsymbol{x}; \Delta)) = \hat{y}$.

As $h_{\boldsymbol{\theta}_{bd}}(g(\boldsymbol{x}; \Delta)) = \hat{y}$ for all $\boldsymbol{x} \in \mathcal{D}_s$ (definition of $\mathcal{D}_s$), we have $h_{\boldsymbol{\theta}}(g(\boldsymbol{x}; \Delta)) = h_{\boldsymbol{\theta}_{bd}}(g(\boldsymbol{x}; \Delta)) \neq y$ if and only if $g(\boldsymbol{x}; \Delta)$ can attack $h_{\boldsymbol{\theta}}$ successfully, for all $\boldsymbol{x} \in \mathcal{D}_s$.

Furthermore, under Assumption 1, $g(\boldsymbol{x}; \Delta)$ is an shared adversarial example of $h_{\boldsymbol{\theta}}$ and $h_{\boldsymbol{\theta}_{bd}}$ if $h_{\boldsymbol{\theta}}(g(\boldsymbol{x}; \Delta)) = h_{\boldsymbol{\theta}_{bd}}(g(\boldsymbol{x}; \Delta)) \neq y$ by the Definition 1.

$\square$

## A.3 Proof of Proposition 3

*Proof.* By Proposition 1 and the decomposition of $\mathcal{R}_{adv}$ in Section 3.2, we have

$$\mathcal{R}_{bd}(h_{\boldsymbol{\theta}}) \leq \mathcal{R}_{adv}(h_{\boldsymbol{\theta}}),$$
$$\mathcal{R}_{sub}(h_{\boldsymbol{\theta}}) \leq \mathcal{R}_{adv}(h_{\boldsymbol{\theta}}).$$

Therefore, we discuss the relation between $\mathcal{R}_{bd}(h_{\boldsymbol{\theta}})$ and $\mathcal{R}_{sub}(h_{\boldsymbol{\theta}})$ now.

Recall that

$$\mathcal{R}_{sub}(h_{\boldsymbol{\theta}}) = \frac{1}{|\mathcal{D}_{-\hat{y}}|} \sum_{i=1}^{N} \max_{\boldsymbol{\epsilon}_i \in \mathcal{S}} \Big\{ \mathbb{I}(\boldsymbol{x}_i \in \mathcal{D}_s) \mathbb{I}(h_{\boldsymbol{\theta}}(\tilde{\boldsymbol{x}}_{i, \boldsymbol{\epsilon}_i}) = h_{\boldsymbol{\theta}_{bd}}(\tilde{\boldsymbol{x}}_{i, \boldsymbol{\epsilon}_i}), \tilde{\boldsymbol{x}}_{i, \boldsymbol{\epsilon}_i} \in \mathcal{A}_{\boldsymbol{\epsilon}_i, \boldsymbol{\theta}}) \tag{15}$$
$$+ \mathbb{I}(\boldsymbol{x}_i \in \mathcal{D}_{-\hat{y}} \setminus \mathcal{D}_s, \tilde{\boldsymbol{x}}_{i, \boldsymbol{\epsilon}_i} \in \mathcal{A}_{\boldsymbol{\epsilon}_i, \boldsymbol{\theta}}) \Big\}.$$

Then, under Assumption 1, we have

- For $\boldsymbol{x} \in \mathcal{D}_s$, Proposition 2 yields that

$$\mathbb{I}(h_{\boldsymbol{\theta}}(g(\boldsymbol{x}; \Delta)) = \hat{y}) = \mathbb{I}(h_{\boldsymbol{\theta}}(g(\boldsymbol{x}; \Delta)) = h_{\boldsymbol{\theta}_{bd}}(g(\boldsymbol{x}; \Delta)))$$
$$\leq \max_{\boldsymbol{\epsilon} \in \mathcal{S}} \mathbb{I}(h_{\boldsymbol{\theta}}(\tilde{\boldsymbol{x}}_{\boldsymbol{\epsilon}}) = h_{\boldsymbol{\theta}_{bd}}(\tilde{\boldsymbol{x}}_{\boldsymbol{\epsilon}}), \tilde{\boldsymbol{x}}_{\boldsymbol{\epsilon}} \in \mathcal{A}_{\boldsymbol{\epsilon}, \boldsymbol{\theta}}). \tag{16}$$

- For $\boldsymbol{x} \in \mathcal{D}_{-\hat{y}} \setminus \mathcal{D}_s$, similar as Proposition 1, we have

$$\mathbb{I}(h_{\boldsymbol{\theta}}(g(\boldsymbol{x}; \Delta)) = \hat{y}) \leq \mathbb{I}(g(\boldsymbol{x}; \Delta) \in \mathcal{A}_{g(\boldsymbol{x}; \Delta) - \boldsymbol{x}, \boldsymbol{\theta}}) \leq \max_{\boldsymbol{\epsilon} \in \mathcal{S}} \mathbb{I}(\tilde{\boldsymbol{x}}_{\boldsymbol{\epsilon}} \in \mathcal{A}_{\boldsymbol{\epsilon}, \boldsymbol{\theta}}).$$

Therefore, we have

$$\mathcal{R}_{bd}(h_{\boldsymbol{\theta}}) \leq \mathcal{R}_{sub}(h_{\boldsymbol{\theta}}).$$

So, under Assumption 1, the following inequalities holds:

$$\mathcal{R}_{bd}(h_{\boldsymbol{\theta}}) \leq \mathcal{R}_{sub}(h_{\boldsymbol{\theta}}) \leq \mathcal{R}_{adv}(h_{\boldsymbol{\theta}}).$$

$\square$

## A.4 Extension to Universal Adversarial Perturbation

Universal Adversarial Perturbation (UAP) has achieved remarkable performance in previous studies [59, 4], especially for defending against backdoor attacks with fixed additive triggers. Now, we extend $\mathcal{R}_{sub}$ to UAP and show that shared adversarial examples also benefit adversarial training with UAP.

We denote the Universal Adversarial Risk by

$$\mathcal{R}_{uadv}(h_{\boldsymbol{\theta}}) = \frac{\max_{\boldsymbol{\epsilon} \in \mathcal{S}} \sum_{i=1}^{N} \mathbb{I}(\tilde{\boldsymbol{x}}_{i,\boldsymbol{\epsilon}} \in \mathcal{A}_{\boldsymbol{\epsilon},\boldsymbol{\theta}}, \boldsymbol{x}_i \in \mathcal{D}_{-\hat{y}})}{|\mathcal{D}_{-\hat{y}}|}.$$

To bridge UAP and backdoor trigger, we first provide the following assumption:

**Assumption 2.** *Assume that the* $g(\boldsymbol{x}, \Delta) = \boldsymbol{x} + \Delta$ *for all* $\boldsymbol{x} \in \mathcal{D}_{cl}$.

Assumption 2 says that the poisoned samples are planted with a fixed additive trigger.

Then, under Assumption 2 and Assumption 1, we have

$$\mathcal{R}_{bd}(h_{\boldsymbol{\theta}}) \leq \mathcal{R}_{uadv} \leq \mathcal{R}_{adv}.$$

by the fact that $\Delta \in \mathcal{S}$.

For clarity, we replace $\tilde{\boldsymbol{x}}_{\boldsymbol{\epsilon}} \in \mathcal{A}_{\boldsymbol{\epsilon},\boldsymbol{\theta}}$ by $h_{\boldsymbol{\theta}}(\boldsymbol{x} + \boldsymbol{\epsilon}) \neq y$ and $\tilde{\boldsymbol{x}}_{\boldsymbol{\epsilon}}$ by $\boldsymbol{x} + \boldsymbol{\epsilon}$ which makes the rest derivation easier to follow.

Then, similar as $\mathcal{R}_{sub}$, we define the universal sub-adversarial risk $\mathcal{R}_{usub}$ as below:

$$\mathcal{R}_{usub}(h_{\boldsymbol{\theta}}) = \frac{1}{|\mathcal{D}_{-\hat{y}}|} \max_{\boldsymbol{\epsilon} \in \mathcal{S}} \sum_{i=1}^{N} \Big\{ \mathbb{I}(h_{\boldsymbol{\theta}}(\boldsymbol{x} + \boldsymbol{\epsilon}) = h_{\boldsymbol{\theta}_{bd}}(\boldsymbol{x} + \boldsymbol{\epsilon}) \neq y, \boldsymbol{x} \in \mathcal{D}_s) +$$
$$\mathbb{I}(h_{\boldsymbol{\theta}}(\boldsymbol{x} + \boldsymbol{\epsilon}) \neq y, \boldsymbol{x} \in \mathcal{D}_{-\hat{y}} \setminus \mathcal{D}_s) \Big\},$$

which considers shared universal adversarial risk on $\mathcal{D}_s$ and vanilla universal adversarial risk on $\mathcal{D}_{-\hat{y}} \setminus \mathcal{D}_s$.

By the definition of $\mathcal{R}_{usub}$, we can easily find $\mathcal{R}_{usub} \leq \mathcal{R}_{uadv}$.

Furthermore, we have

$$\mathcal{R}_{bd}(h_{\boldsymbol{\theta}}) = \frac{\sum_{i=1}^{N} \mathbb{I}(h_{\boldsymbol{\theta}}(\boldsymbol{x} + \Delta) = \hat{y}, \boldsymbol{x} \in \mathcal{D}_{-\hat{y}})}{|\mathcal{D}_{-\hat{y}}|}$$
$$= \frac{\sum_{i=1}^{N} \{ \mathbb{I}(h_{\boldsymbol{\theta}}(\boldsymbol{x} + \Delta) = h_{\boldsymbol{\theta}_{bd}}(\boldsymbol{x} + \Delta), \boldsymbol{x} \in \mathcal{D}_s) + \mathbb{I}(h_{\boldsymbol{\theta}}(\boldsymbol{x} + \Delta) = \hat{y}, \boldsymbol{x} \in \mathcal{D}_{-\hat{y}} \setminus \mathcal{D}_s) \}}{|\mathcal{D}_{-\hat{y}}|}$$
$$\leq \frac{\max_{\boldsymbol{\epsilon} \in \mathcal{S}} \sum_{i=1}^{N} \{ \mathbb{I}(h_{\boldsymbol{\theta}}(\boldsymbol{x} + \boldsymbol{\epsilon}) = h_{\boldsymbol{\theta}_{bd}}(\boldsymbol{x} + \boldsymbol{\epsilon}), \boldsymbol{x} \in \mathcal{D}_s) + \mathbb{I}(h_{\boldsymbol{\theta}}(\boldsymbol{x} + \boldsymbol{\epsilon}) \neq y, \boldsymbol{x} \in \mathcal{D}_{-\hat{y}} \setminus \mathcal{D}_s) \}}{|\mathcal{D}_{-\hat{y}}|}$$
$$= \mathcal{R}_{usub}(h_{\boldsymbol{\theta}}).$$

Therefore, we reach the following proposition

**Proposition 4.** *Under Assumption 2 and Assumption 1, for a classifier* $h_{\boldsymbol{\theta}}$, *the following inequalities hold*

$$\mathcal{R}_{bd}(h_{\boldsymbol{\theta}}) \leq \mathcal{R}_{usub}(h_{\boldsymbol{\theta}}) \leq \mathcal{R}_{uadv}(h_{\boldsymbol{\theta}}) \leq \mathcal{R}_{adv}(h_{\boldsymbol{\theta}}).$$

Therefore, the shared adversarial examples also benefit adversarial training with UAP.

## A.5 Extension to Targeted Adversarial Training

In the previous sections, we assume that the defender has no access to either the trigger or the target label. Since there are already some works for backdoor trigger detection Guo et al. [16], Dong et al. [10], we discuss the case where the defender can estimate the target label $\hat{y}$ and show that shared adversarial examples can also be beneficial for adversarial training with target label.

Given a target label $\hat{y}$, we define the Targeted Adversarial Risk as

$$\mathcal{R}_{tadv}(h_{\boldsymbol{\theta}}) = \frac{\sum_{i=1}^{N} \max_{\boldsymbol{\epsilon}_i \in \mathcal{S}} \mathbb{I}(h_{\boldsymbol{\theta}}(\tilde{\boldsymbol{x}}_{i,\boldsymbol{\epsilon}_i}) = \hat{y}, \boldsymbol{x}_i \in \mathcal{D}_{-\hat{y}})}{|\mathcal{D}_{-\hat{y}}|}. \tag{17}$$

By the definition of $\hat{y}$, we have $\mathcal{R}_{bd}(h_{\boldsymbol{\theta}}) \leq \mathcal{R}_{tadv}(h_{\boldsymbol{\theta}})$ under Assumption 1.

Then, we have

$$
\begin{aligned}
\mathcal{R}_{tadv}(h_{\boldsymbol{\theta}}) = \frac{1}{|\mathcal{D}_{-\hat{y}}|} \sum_{i=1}^{N} \max_{\boldsymbol{\epsilon}_i \in \mathcal{S}} &\Big\{ \mathbb{I}(\boldsymbol{x}_i \in \mathcal{D}_s)\mathbb{I}(h_{\boldsymbol{\theta}}(\tilde{\boldsymbol{x}}_{i,\boldsymbol{\epsilon}_i}) = h_{\boldsymbol{\theta}_{bd}}(\tilde{\boldsymbol{x}}_{i,\boldsymbol{\epsilon}_i}) = \hat{y}) \\
&+ \mathbb{I}(\boldsymbol{x}_i \in \mathcal{D}_s)\mathbb{I}(h_{\boldsymbol{\theta}}(\tilde{\boldsymbol{x}}_{i,\boldsymbol{\epsilon}_i}) = \hat{y}, h_{\boldsymbol{\theta}_{bd}}(\tilde{\boldsymbol{x}}_{i,\boldsymbol{\epsilon}_i}) \neq \hat{y}) + \mathbb{I}(\boldsymbol{x}_i \in \mathcal{D}_{-\hat{y}} \setminus \mathcal{D}_s, h_{\boldsymbol{\theta}}(\tilde{\boldsymbol{x}}_{i,\boldsymbol{\epsilon}_i}) = \hat{y}) \Big\}.
\end{aligned}
\tag{18}
$$

Then, similar as $\mathcal{R}_{sub}$, we define the targeted sub-adversarial risk $\mathcal{R}_{tsub}$ by removing the second component in $\mathcal{R}_{tadv}(h_{\boldsymbol{\theta}})$:

$$
\begin{aligned}
\mathcal{R}_{tsub}(h_{\boldsymbol{\theta}}) = \frac{1}{|\mathcal{D}_{-\hat{y}}|} \sum_{i=1}^{N} \max_{\boldsymbol{\epsilon}_i \in \mathcal{S}} &\Big\{ \mathbb{I}(\boldsymbol{x}_i \in \mathcal{D}_s)\mathbb{I}(h_{\boldsymbol{\theta}}(\tilde{\boldsymbol{x}}_{i,\boldsymbol{\epsilon}_i}) = h_{\boldsymbol{\theta}_{bd}}(\tilde{\boldsymbol{x}}_{i,\boldsymbol{\epsilon}_i}) = \hat{y}) \\
&+ \mathbb{I}(\boldsymbol{x}_i \in \mathcal{D}_{-\hat{y}} \setminus \mathcal{D}_s, h_{\boldsymbol{\theta}}(\tilde{\boldsymbol{x}}_{i,\boldsymbol{\epsilon}_i}) = \hat{y}) \Big\}.
\end{aligned}
$$

which considers shared targeted adversarial risk on $\mathcal{D}_s$ and vanilla targeted adversarial risk on $\mathcal{D}_{-\hat{y}} \setminus \mathcal{D}_s$.

Since the removed component is irrelevant to backdoor risk, we have $\mathcal{R}_{bd} \leq \mathcal{R}_{tsub}$ under Assumption 1.

Therefore, we reach the following proposition

*Proof.* Under Assumption 2 and Assumption 1, for a classifier $h_{\boldsymbol{\theta}}$, the following inequalities hold

$$\mathcal{R}_{bd}(h_{\boldsymbol{\theta}}) \leq \mathcal{R}_{tsub}(h_{\boldsymbol{\theta}}) \leq \mathcal{R}_{tadv}(h_{\boldsymbol{\theta}}) \leq \mathcal{R}_{adv}(h_{\boldsymbol{\theta}}).$$

$\square$

Therefore, the shared adversarial examples also benefit adversarial training with target label.

Moreover, since $\mathcal{R}_{tsub}(h_{\boldsymbol{\theta}})$ only consider shared adversarial example with target label $\hat{y}$, the following proposition holds,

*Proof.* Under Assumption 2 and Assumption 1, for a classifier $h_{\boldsymbol{\theta}}$, the following inequalities hold

$$\mathcal{R}_{bd}(h_{\boldsymbol{\theta}}) \leq \mathcal{R}_{tsub}(h_{\boldsymbol{\theta}}) \leq \mathcal{R}_{sub}(h_{\boldsymbol{\theta}}) \leq \mathcal{R}_{adv}(h_{\boldsymbol{\theta}}).$$

$\square$

The above proposition shows that $\mathcal{R}_{tsub}(h_{\boldsymbol{\theta}})$ is a tighter bound of $\mathcal{R}_{bd}(h_{\boldsymbol{\theta}})$ than $\mathcal{R}_{sub}(h_{\boldsymbol{\theta}})$.

### A.6 Extension to Multi-target/Multi-trigger

In previous sections, we assume that there is only one trigger and one target for all samples. Now, we consider to a more general case where a set of triggers $\boldsymbol{V}$ and a set of target labels $\hat{\boldsymbol{Y}}$ are given. This is a more challenging case since there may be multiple triggers and/or multiple targets for one sample.

We define $T$ to be a mapping from the sample to the corresponding set of feasible trigger-target pairs. Therefore, given a sample $\boldsymbol{x}$, the set of all feasible trigger-target pairs is given by $T(\boldsymbol{x})$. Then, Assumption 1 is extended to all triggers accordingly, *i.e.*, $g(\boldsymbol{x}, \Delta) - \boldsymbol{x} \in \mathcal{S}, \forall (\Delta, \hat{y}) \in T(\boldsymbol{x})$.

In this setting, we define that a classifier $h_{\boldsymbol{\theta}}$ is said to be attacked by a poisoned sample generated by $\boldsymbol{x}$ if there exists $(\Delta, \hat{y}) \in T(\boldsymbol{x})$ such that $h_{\boldsymbol{\theta}}(g(\boldsymbol{x}, \Delta)) = \hat{y} \neq y$.

Then, the (maximum) backdoor risk on $\mathcal{D}_{cl}$ is defined to be

$$\mathcal{R}_{bd}(h_{\boldsymbol{\theta}}) = \frac{\sum_{i=1}^{N} \max_{(\Delta_i, \hat{y}_i) \in T(\boldsymbol{x}_i)} \mathbb{I}(h_{\boldsymbol{\theta}}(g(\boldsymbol{x}_i, \Delta_i)) = \hat{y}_i \neq y)}{|\mathcal{D}_{cl}|}. \tag{19}$$

Similarly, the adversarial risk on $\mathcal{D}_{cl}$ is defined as

$$\mathcal{R}_{adv} = \frac{1}{|\mathcal{D}_{cl}|} \sum_{i=1}^{N} \max_{\boldsymbol{\epsilon}_i \in \mathcal{S}} \{\mathbb{I}(h_{\boldsymbol{\theta}}(g(\boldsymbol{x}_i, \boldsymbol{\epsilon}_i)) \neq y)\}.$$

We define $\mathcal{D}_s = \{\boldsymbol{x} \in \mathcal{D}_{cl} : h_{\boldsymbol{\theta}_{bd}}(g(\boldsymbol{x}, \Delta)) = \hat{y} \neq y, \forall (\Delta, \hat{y}) \in T(\boldsymbol{x})\}$ to be the set of samples on which planting any feasible trigger can activate the corresponding backdoor. Then, the corresponding sub-adversarial risk is defined as

$$\mathcal{R}_{sub} = \frac{1}{|\mathcal{D}_{cl}|} \sum_{i=1}^{N} \max_{\boldsymbol{\epsilon}_i \in \mathcal{S}} \{\mathbb{I}(h_{\boldsymbol{\theta}}(g(\boldsymbol{x}_i, \boldsymbol{\epsilon}_i)) = h_{\boldsymbol{\theta}_{bd}}(g(\boldsymbol{x}_i, \boldsymbol{\epsilon}_i)) \neq y, \boldsymbol{x}_i \in \mathcal{D}_s)$$
$$+ \mathbb{I}(h_{\boldsymbol{\theta}}(g(\boldsymbol{x}_i, \boldsymbol{\epsilon}_i)) \neq y, \boldsymbol{x}_i \in \mathcal{D}_{cl} \setminus \mathcal{D}_s)\}.$$

which considers shared adversarial risk on $\mathcal{D}_s$ and vanilla targeted adversarial risk on $\mathcal{D}_{cl} \setminus \mathcal{D}_s$.

By definition, we have $\mathcal{R}_{sub} \leq \mathcal{R}_{adv}$

Moreover, we have

$$\mathcal{R}_{bd}(h_{\boldsymbol{\theta}}) = \frac{\sum_{i=1}^{N} \max_{(\Delta_i, \hat{y}_i) \in T(\boldsymbol{x}_i)} \mathbb{I}(h_{\boldsymbol{\theta}}(g(\boldsymbol{x}_i, \Delta_i)) = \hat{y}_i \neq y)}{|\mathcal{D}_{cl}|}$$

$$= \frac{1}{|\mathcal{D}_{cl}|} \sum_{i=1}^{N} \max_{(\Delta_i, \hat{y}_i) \in T(\boldsymbol{x}_i)} \{\mathbb{I}(h_{\boldsymbol{\theta}}(g(\boldsymbol{x}_i, \Delta_i)) = h_{\boldsymbol{\theta}_{bd}}(g(\boldsymbol{x}_i, \Delta_i)), h_{\boldsymbol{\theta}_{bd}}(g(\boldsymbol{x}_i, \Delta_i)) = \hat{y}_i \neq y)$$
$$+ \mathbb{I}(h_{\boldsymbol{\theta}}(g(\boldsymbol{x}_i, \Delta_i)) \neq y, h_{\boldsymbol{\theta}_{bd}}(g(\boldsymbol{x}_i, \Delta_i)) \neq \hat{y}_i)\}$$

$$\leq \frac{1}{|\mathcal{D}_{cl}|} \sum_{i=1}^{N} \max_{(\Delta_i, \hat{y}_i) \in T(\boldsymbol{x}_i)} \{\mathbb{I}(h_{\boldsymbol{\theta}}(g(\boldsymbol{x}_i, \Delta_i)) = h_{\boldsymbol{\theta}_{bd}}(g(\boldsymbol{x}_i, \Delta_i)) \neq y, \boldsymbol{x}_i \in \mathcal{D}_s)$$
$$+ \mathbb{I}(h_{\boldsymbol{\theta}}(g(\boldsymbol{x}_i, \Delta_i)) \neq y, \boldsymbol{x}_i \in \mathcal{D}_{cl} \setminus \mathcal{D}_s)\}$$

$$\leq \frac{1}{|\mathcal{D}_{cl}|} \sum_{i=1}^{N} \max_{\boldsymbol{\epsilon}_i \in \mathcal{S}} \{\mathbb{I}(h_{\boldsymbol{\theta}}(g(\boldsymbol{x}_i, \boldsymbol{\epsilon}_i)) = h_{\boldsymbol{\theta}_{bd}}(g(\boldsymbol{x}_i, \boldsymbol{\epsilon}_i)) \neq y, \boldsymbol{x}_i \in \mathcal{D}_s)$$
$$+ \mathbb{I}(h_{\boldsymbol{\theta}}(g(\boldsymbol{x}_i, \boldsymbol{\epsilon}_i)) \neq y, \boldsymbol{x}_i \in \mathcal{D}_{cl} \setminus \mathcal{D}_s)\}$$
$$= \mathcal{R}_{sub}.$$

Therefore, for a backdoor attack with multiple targets and/or multiple triggers, the proposed sub-adversarial risk $\mathcal{R}_{sub}$ is still a tighter upper bound of $\mathcal{R}_{bd}$ compared to $\mathcal{R}_{adv}$.

## A.7 Relaxation gap

To build a practical objective for defending against a backdoor attack, two relaxations are applied to $\mathcal{R}_{sub}$. Here, we discuss the relaxation gap in the relaxing process:

- **Replace $\mathcal{D}_s$ by $\mathcal{D}_{-\hat{y}}$.**

  Since $\mathcal{D}_s$ is not accessible, the first relaxation is to replace $\mathcal{D}_s$ by $\mathcal{D}_{-\hat{y}}$ and consider shared adversarial examples on all samples in $\mathcal{D}_{-\hat{y}}$. Denote $\mathcal{R}_1(h_{\boldsymbol{\theta}})$ to be the relaxed risk after replacing

$\mathcal{D}_s$ by $\mathcal{D}_{-\hat{y}}$. Since $\mathcal{R}_{sub}$ can be decomposed as follows:

$$
\begin{aligned}
\mathcal{R}_{sub}(h_{\boldsymbol{\theta}}) =& \frac{1}{|\mathcal{D}_{-\hat{y}}|} \sum_{i=1}^{N} \max_{\boldsymbol{\epsilon}_i \in \mathcal{S}} \Big\{ \mathbb{I}(\boldsymbol{x}_i \in \mathcal{D}_s) \mathbb{I}(h_{\boldsymbol{\theta}}(\tilde{\boldsymbol{x}}_{i,\boldsymbol{\epsilon}_i}) = h_{\boldsymbol{\theta}_{bd}}(\tilde{\boldsymbol{x}}_{i,\boldsymbol{\epsilon}_i}), \tilde{\boldsymbol{x}}_{i,\boldsymbol{\epsilon}_i} \in \mathcal{A}_{\boldsymbol{\epsilon}_i,\boldsymbol{\theta}}) \\
& + \mathbb{I}(\boldsymbol{x}_i \in \mathcal{D}_{-\hat{y}} \setminus \mathcal{D}_s, \tilde{\boldsymbol{x}}_{i,\boldsymbol{\epsilon}_i} \in \mathcal{A}_{\boldsymbol{\epsilon}_i,\boldsymbol{\theta}}) \Big\} \\
=& \frac{1}{|\mathcal{D}_{-\hat{y}}|} \sum_{i=1}^{N} \max_{\boldsymbol{\epsilon}_i \in \mathcal{S}} \Big\{ \mathbb{I}(\boldsymbol{x}_i \in \mathcal{D}_{-\hat{y}}) \mathbb{I}(h_{\boldsymbol{\theta}}(\tilde{\boldsymbol{x}}_{i,\boldsymbol{\epsilon}_i}) = h_{\boldsymbol{\theta}_{bd}}(\tilde{\boldsymbol{x}}_{i,\boldsymbol{\epsilon}_i}), \tilde{\boldsymbol{x}}_{i,\boldsymbol{\epsilon}_i} \in \mathcal{A}_{\boldsymbol{\epsilon}_i,\boldsymbol{\theta}}) \\
& + \mathbb{I}(\boldsymbol{x}_i \in \mathcal{D}_{-\hat{y}} \setminus \mathcal{D}_s) \mathbb{I}(h_{\boldsymbol{\theta}}(\tilde{\boldsymbol{x}}_{i,\boldsymbol{\epsilon}_i}) \neq h_{\boldsymbol{\theta}_{bd}}(\tilde{\boldsymbol{x}}_{i,\boldsymbol{\epsilon}_i}), \tilde{\boldsymbol{x}}_{i,\boldsymbol{\epsilon}_i} \in \mathcal{A}_{\boldsymbol{\epsilon}_i,\boldsymbol{\theta}}) \Big\}.
\end{aligned}
$$

We can find that the relaxation gap is

$$
\begin{aligned}
\mathcal{R}_{sub} - \mathcal{R}_1 \leq& \frac{1}{|\mathcal{D}_{-\hat{y}}|} \sum_{i=1}^{N} \max_{\boldsymbol{\epsilon}_i \in \mathcal{S}} \mathbb{I}(\boldsymbol{x}_i \in \mathcal{D}_{-\hat{y}} \setminus \mathcal{D}_s) \mathbb{I}(h_{\boldsymbol{\theta}}(\tilde{\boldsymbol{x}}_{i,\boldsymbol{\epsilon}_i}) \neq h_{\boldsymbol{\theta}_{bd}}(\tilde{\boldsymbol{x}}_{i,\boldsymbol{\epsilon}_i}), \tilde{\boldsymbol{x}}_{i,\boldsymbol{\epsilon}_i} \in \mathcal{A}_{\boldsymbol{\epsilon}_i,\boldsymbol{\theta}}) \\
\leq& \frac{1}{|\mathcal{D}_{-\hat{y}}|} \sum_{i=1}^{N} \mathbb{I}(\boldsymbol{x}_i \in \mathcal{D}_{-\hat{y}} \setminus \mathcal{D}_s) \\
=& 1 - \mathcal{R}_{bd}(h_{\boldsymbol{\theta}_{bd}}).
\end{aligned}
$$

And the relaxation gap $\text{GAP}_1$ is bounded by $1 - \mathcal{R}_{bd}(h_{\boldsymbol{\theta}_{bd}})$. Since a poisoned model $h_{\boldsymbol{\theta}_{bd}}$ usually has a high ASR for the backdoor attack, the gap $\text{GAP}_1$ is usually negligible.

**Remark:** We remark that the experiment results show that our method also works well for the poisoned models with low ASR. See Section D for more details.

- **Replace $\mathcal{D}_{-\hat{y}}$ by $\mathcal{D}_{cl}$.**

  The second relaxation is to replace $\mathcal{D}_{-\hat{y}}$ by $\mathcal{D}_{cl}$ when $\hat{y}$ is not accessible. Let $\mathcal{D}_{\hat{y}} = \mathcal{D}_{cl} \setminus \mathcal{D}_{-\hat{y}}$. Then, the risk after the second relaxation is:

$$
\begin{aligned}
\mathcal{R}_{share}(h_{\boldsymbol{\theta}}) =& \frac{1}{N} \sum_{i=1}^{N} \max_{\boldsymbol{\epsilon}_i \in \mathcal{S}} \Big\{ \mathbb{I}(h_{\boldsymbol{\theta}}(\tilde{\boldsymbol{x}}_{i,\boldsymbol{\epsilon}_i}) = h_{\boldsymbol{\theta}_{bd}}(\tilde{\boldsymbol{x}}_{i,\boldsymbol{\epsilon}_i}), \tilde{\boldsymbol{x}}_{i,\boldsymbol{\epsilon}_i} \in \mathcal{A}_{\boldsymbol{\epsilon}_i,\boldsymbol{\theta}}) \Big\} \\
=& \frac{|\mathcal{D}_{-\hat{y}}|}{N} \frac{1}{|\mathcal{D}_{-\hat{y}}|} \sum_{i=1}^{N} \max_{\boldsymbol{\epsilon}_i \in \mathcal{S}} \Big\{ \mathbb{I}(\boldsymbol{x}_i \in \mathcal{D}_{-\hat{y}}) \mathbb{I}(h_{\boldsymbol{\theta}}(\tilde{\boldsymbol{x}}_{i,\boldsymbol{\epsilon}_i}) = h_{\boldsymbol{\theta}_{bd}}(\tilde{\boldsymbol{x}}_{i,\boldsymbol{\epsilon}_i}), \tilde{\boldsymbol{x}}_{i,\boldsymbol{\epsilon}_i} \in \mathcal{A}_{\boldsymbol{\epsilon}_i,\boldsymbol{\theta}}) \Big\} \\
& + \frac{|\mathcal{D}_{\hat{y}}|}{N} \frac{1}{|\mathcal{D}_{\hat{y}}|} \sum_{i=1}^{N} \max_{\boldsymbol{\epsilon}_i \in \mathcal{S}} \Big\{ \mathbb{I}(\boldsymbol{x}_i \in \mathcal{D}_{\hat{y}}) \mathbb{I}(h_{\boldsymbol{\theta}}(\tilde{\boldsymbol{x}}_{i,\boldsymbol{\epsilon}_i}) = h_{\boldsymbol{\theta}_{bd}}(\tilde{\boldsymbol{x}}_{i,\boldsymbol{\epsilon}_i}), \tilde{\boldsymbol{x}}_{i,\boldsymbol{\epsilon}_i} \in \mathcal{A}_{\boldsymbol{\epsilon}_i,\boldsymbol{\theta}}) \Big\}.
\end{aligned}
$$

where the first component is $\mathcal{R}_1$ scaled by $\frac{|\mathcal{D}_{-\hat{y}}|}{N}$ and the second component is the vanilla adversarial risk on $\mathcal{D}_{\hat{y}}$, scaled by $\frac{|\mathcal{D}_{\hat{y}}|}{N}$. So, the relaxation gap for the second relaxation is

$$
\begin{aligned}
\mathcal{R}_1 - \mathcal{R}_{share} =& \frac{|\mathcal{D}_{\hat{y}}|}{N} \mathcal{R}_1 \\
& - \frac{|\mathcal{D}_{\hat{y}}|}{N} \frac{1}{|\mathcal{D}_{\hat{y}}|} \sum_{i=1}^{N} \max_{\boldsymbol{\epsilon}_i \in \mathcal{S}} \Big\{ \mathbb{I}(\boldsymbol{x}_i \in \mathcal{D}_{\hat{y}}) \mathbb{I}(h_{\boldsymbol{\theta}}(\tilde{\boldsymbol{x}}_{i,\boldsymbol{\epsilon}_i}) = h_{\boldsymbol{\theta}_{bd}}(\tilde{\boldsymbol{x}}_{i,\boldsymbol{\epsilon}_i}), \tilde{\boldsymbol{x}}_{i,\boldsymbol{\epsilon}_i} \in \mathcal{A}_{\boldsymbol{\epsilon}_i,\boldsymbol{\theta}}) \Big\}.
\end{aligned}
$$

We can find that the generalization gap is related to the $\frac{|\mathcal{D}_{\hat{y}}|}{N}$, *i.e.*, the portion of samples whose labels are target labels, and negligible when $\frac{|\mathcal{D}_{\hat{y}}|}{N}$ is small.

**Remark.** For using $\mathcal{R}_{share}(h_{\boldsymbol{\theta}})$ to mitigate the backdoor, the influence of $\frac{|\mathcal{D}_{-\hat{y}}|}{N}$ can be eliminated by altering the trade-off parameter in Problem (8). Also, adding triggers to samples labeled $\hat{y}$ is harmless during the testing/deployment phase. So, the vanilla adversarial risk on $\mathcal{D}_{\hat{y}}$ has a negligible negative impact on defending the backdoor. Therefore, replacing $\mathcal{D}_{-\hat{y}}$ with $\mathcal{D}_{cl}$ has little negative impact on mitigating the backdoor.

# B  Discussion

In this section, we provide a comprehensive discussion of the proposed method, including its efficiency, comparison to other methods, choice of perturbation, generalization ability to attack with excessive-magnitude triggers, performance on the generated dataset, and resistance to defense-aware/adaptive attacks.

## B.1  Efficiency and Computational Cost

Here, we would like to emphasize that SAU is an efficient and effective defense method that can mitigate backdoor attacks with acceptable overhead for the following reasons:

- SAU is a post-processing method that fine-tunes the backdoor model on a clean dataset with a small size. For instance, we can use only 500 samples from CIFAR-10 to fine-tune the model by SAU (see Appendix D.6 for more details), which takes 1.52 seconds/epoch to fine-tune PreAct-ResNet18 with RTX 4090. Therefore, it can be executed efficiently without requiring a large amount of data or computation resources.
- SAU only takes a few epochs to take effect, which further reduces the computational cost of executing it in practice. As shown in Appendix D.4, SAU can achieve a low Attack Success Rate and a high Accuracy in a few epochs.

To further demonstrate the efficiency of SAU, we compare its average runtime with other state-of-the-art defense methods against attacks. We use the same experimental setting as in Section 4, where the poisoning ratio is 10% and the backbone model is PreAct-ResNet18. All experiments are conducted on a server with GPU RTX 4090 and CPU AMD EPYC 7543 32-Core Processor. As different methods adopt different learning paradigms for backdoor mitigation, we measure the average runtime for each defense method to take effect on CIFAR-10 and Tiny ImageNet (ASR $< 5\%$ and ACC $> 85\%$ on CIFAR-10, or ASR $< 5\%$, ACC $> 45\%$ on Tiny ImageNet). Note that if a defend method cannot reach the criteria for "Take effect", the maximum runtime is reported. The results are summarized in the following table.

Table 7: Time to take effect for different methods, averaged over different attacks

| Dataset | ANP [55] | FP [31] | NC [47] | NAD [28] | EP [64] | i-BAU [59] | SAU (Ours) |
|---|---|---|---|---|---|---|---|
| CIFAR-10 | 289.86s | 266.57s | 825.86s | 75.14s | 65.71s | 47.00s | 43.86s |
| Tiny ImageNet | 1086.50s | 318.30s | 27359.70s | 227.01s | 141.67s | 621.33s | 262.60s |

From Table 7, SAU is faster than most of the existing methods, except for NAD and EP, which shows that SAU is an efficient and effective defense method.

## B.2  Comparison with Certified Robustness methods

In backdoor defense, methods can be categorized into two classes: empirical methods which aim to develop effective and efficient methods for backdoor defense, and certified methods which aim to build a provable framework for backdoor robustness. As **our method is an empirical method**, we would like to point out that our method is different from the existing works on certified robustness in several aspects, which makes them incomparable or impractical to apply to our setting. Specifically:

- Efficiency and scalability. Certified methods usually have very high computation costs and are infeasible to apply to large-scale deep neural networks, while our method is efficient and scalable. For example, the representative methods RAB [51] and BagFlip [61] rely on voting over 1000 smoothed models for prediction and are only feasible for some simple neural networks and simple classification problems. To apply these methods to our setting, we would need to train 1000 PreAct-ResNet models on CIFAR-10 from scratch, which would take over 600 hours (25 days) and 620GB of space for a single attack on a server with one RTX 3090.
- Applicability to model architecture. Many certified methods are restricted to some simple or classical models, while our method is general and applicable to any deep neural network

architecture. For example, the method in [20] is designed for only k-Nearest Neighbors (kNN) and radius Nearest Neighbors (rNN) models with some feature extractor and cannot be applied to modern deep neural networks.

- Difference in the threat model. Most certified methods focus on the threat model where the training dataset is accessible to defenders while our method focuses on the threat model where only a poisoned model and a few clean samples are given.

Therefore, we believe that our method is more practical and effective for defending against backdoor attacks in deep neural networks than the existing works on certified robustness from the perspective of efficiency, scalability, and model structures.

## B.3 Comparison with other AT based methods

Here, we compare the proposed method SAU with other representative Adversarial Training based methods from the following aspects:

- Threat Model: whether the training dataset is accessible and whether the extra clean dataset is accessible.
- Types of Perturbation: whether universal adversarial perturbation is supported.
- Types of Adversarial Attack: whether Targeted Adversarial Attack is supported.

The answer "Yes" and "No" are indicated by $\sqrt{}$ and $\times$, respectively. The "Optional" means the method supports but is not limited to a specific answer. We summarize the result in Table 8.

Table 8: Comparison between SAU with other AT based methods

| Method | Training Dataset | Extra Clean Dataset | UAP | Targeted Attack |
|---|---|---|---|---|
| AT from Scratch [52] | $\sqrt{}$ | $\times$ | $\times$ | $\times$ |
| Composite AT [13] | $\sqrt{}$ | $\times$ | $\times$ | $\times$ |
| i-BAU [59] | $\times$ | $\sqrt{}$ | $\sqrt{}$ | $\times$ |
| PBE [35] | $\sqrt{}$ | $\times$ | $\times$ | $\times$ |
| AFT [35] | $\times$ | $\sqrt{}$ | $\times$ | $\times$ |
| SAU (Ours) | $\times$ | $\sqrt{}$ | Optional | Optional |

We remark that two methods, $i.e.$, PBE and AFT are proposed in Mu et al. [35] for different threat models, where PBE is designed to filter a clean dataset from a training dataset, and AFT is to perform AT on the given clean dataset. Therefore, **AFT is equivalent to the Vanilla AT in the Figure 1 and Section 4.3.**

As summarized in Table 8, the current AT-based methods can be roughly categorized into two types based on their threat model: in-training (given training dataset) and post-training (given poisoned model and extra clean dataset), and **the difference in threat model make methods from different categories incomparable.**

**1. Comparison with In-training AT-based methods.** Although SAU is not comparable with the in-training AT-based method due to the underlying assumption of the threat model, we would like to adopt different threat models for different methods to highlight the advantages of SAU compared to the in-training AT-based methods. Therefore, **we compare SAU with the latest SOTA in-processing AT-based method,** $i.e.$**, Composite AT (CAT) [13].**

We first highlight the following differences between CAT and SAU:

- SAU is theoretically motivated by the relationship between adversarial robustness and backdoor risk, while CAT is empirically driven by some experimental observations. Therefore, our method has a theoretical guarantee for the performance against various backdoor attacks, while CAT does not.
- SAU fine-tunes a backdoored model with a small clean dataset, while CAT trains a backdoor-free model from scratch on a poisoned dataset. Therefore, SAU is more efficient than CAT.

Then, we conduct experiments on defending against backdoor attacks and summarize the results in Table 9 and 10.

Table 9: Experiments Results with poisoning ratio 1%

| Method | No Defense | No Defense | No Defense | CAT [13] | CAT [13] | CAT [13] | SAU (**Ours**) | SAU (**Ours**) | SAU (**Ours**) |
|--------|------|------|-------|------|------|-------|------|------|-------|
| Attack | Acc | ASR | R-ACC | Acc | ASR | R-ACC | Acc | ASR | R-ACC |
| BadNets [15] | 93.14 | 74.73 | 24.24 | 75.52 | 2.50 | 74.73 | **91.25** | **0.94** | **91.16** |
| Input-Aware | 91.74 | 79.18 | 19.89 | 75.24 | 72.48 | 23.97 | **92.20** | **3.63** | **84.06** |
| SSBA [30] | 93.43 | 73.44 | 24.89 | 74.94 | 3.07 | 71.40 | **91.34** | **0.79** | **88.46** |
| WaNet [38] | 90.65 | 12.63 | 79.94 | 75.21 | 2.68 | 74.46 | **91.84** | **1.23** | **89.89** |
| Average | 92.24 | 60.00 | 37.24 | 75.23 | 20.18 | 61.14 | **91.66** | **1.65** | **88.39** |

Table 10: Experiments Results with poisoning ratio 10%

| Method | No Defense | No Defense | No Defense | CAT [13] | CAT [13] | CAT [13] | SAU (**Ours**) | SAU (**Ours**) | SAU (**Ours**) |
|--------|------|------|-------|------|------|-------|------|------|-------|
| Attack | Acc | ASR | R-ACC | Acc | ASR | R-ACC | Acc | ASR | R-ACC |
| BadNets [15] | 91.32 | 95.03 | 4.67 | 74.42 | 92.49 | 6.21 | **89.31** | **1.53** | **88.81** |
| Input-Aware | 90.67 | 98.26 | 1.66 | 74.21 | 96.81 | 2.88 | **91.59** | **1.27** | **88.54** |
| SSBA [30] | 92.88 | 97.8 | 1.99 | 74.29 | 28.29 | 57.49 | **90.84** | **1.79** | **85.83** |
| WaNet [38] | 91.25 | 89.73 | 9.76 | 74.62 | 4.87 | 73.07 | **90.41** | **2.51** | **79.69** |
| Average | 91.53 | 95.21 | 4.52 | 74.39 | 55.61 | 34.91 | **92.54** | **1.78** | **85.72** |

The experimental results show that our method is superior to CAT, as evidenced by the following aspects:

- SAU achieves significantly higher accuracy (ACC) and lower attack success rate (ASR) than CAT on both poisoning ratios.

- SAU is more robust to different poisoning ratios than CAT. In contrast, CAT fails on a high poisoning ratio (10%) (also observed in Table XI of Gao et al. [13]).

**2. Comparison with Post-training AT-based methods.** The most related methods to SAU are the post-processing AT-based method including I-BAU and AFT, and we highlight the following differences between them:

- I-BAU is proposed under the assumption that the same trigger are used for all poisoned samples which may not be true for more advanced attacks, while SAU is shown to be effective for attacks with multi-trigger and sample-specific trigger theoretically and empirically.

- AFT, which is to perform AT on clean dataset, is motivated by some empirical observation, while SAU is motivated by a novel analysis on the relation between adversarial examples and poisoned samples.

Besides, the experimental comparison between SAU and I-BAU in Section 4.2 and the experimental between SAU and AFT (Vanilla AT on Clean Dataset) in Section 4.3 show that SAU outperforms I-BAU and AFT by a large margin.

### B.4 Choice of perturbation and generalization ability of SAU

In this section, we discuss the choice of perturbation set and show that the proposed method, $i.e.$, SAU, can generalize to attacks with excessive-magnitude triggers.

**1. Type of adversarial perturbation (AP) and perturbation set.** The theoretical analysis and the proposed method SAU is independent of the AP type and AP set (not only $L_p$ ball). For sample space $\mathcal{X}$, define the set $G = \{g(x, \Delta) - x | x \in \mathcal{X}\}$ and the general perturbation set $S = \{T(x, \epsilon) - x | x \in \mathcal{X}\}$ for adversarial attack $T$ with learnable parameter $\epsilon \in \Omega$ such that an adversarial example can be generated by $\tilde{x}_\epsilon = T(x, \epsilon)$. Then, the theoretical analysis and proposed method can be applied if $G \subseteq S$, where the AP can be any type, such as additive AP, Spatially Transformed AP [56], or Perceptual AP [25], etc.

**2. Effect of perturbation set $S$.**  The key of SAU is to consider Shared Adversarial Example (SAE) as surrogates for poisoned samples. Given models, **the set of SAE is determined by $S$: larger $S$ leads to larger set of SAE, and vice versa** Therefore, the perturbation set $S$ plays an important role in the effectiveness of SAU. Specifically, adopting a larger $S$ can cover unknown triggers but also increases the chance of picking SAE which is irrelevant to poisoned samples, and may weaken the defense performance. Meanwhile, some poisoned samples may be beyond the scope of SAE if a smaller $S$ is adopted.

**3. Generalization to excessive-magnitude trigger.**  In practice, we find that the proposed method SAU is effective for various attacks, even if Assumption 1 does not hold. For example, SAU with $L_\infty$ norm bound 0.2 can still effectively mitigate SSBA attack whose average $L_\infty$ trigger norm is much larger than 0.2, revealing the generalization of SAU to attacks with excessive-magnitude trigger.

Table 11: SAU for SSBA Attack

| Dataset | $L_1$ | $L_2$ | $L_\infty$ | ACC | ASR | R-ACC |
|---|---|---|---|---|---|---|
| CIFAR-10 | 108.33 | 3.29 | 0.33 | 90.99 | 0.58 | 87.04 |
| Tiny | 581.65 | 8.26 | 0.43 | 51.85 | 0.11 | 36.36 |

To investigate the SAU's generalization ability, we conduct experiments on BadNets with various strengths. Specifically, for BadNets with a 3x3 patched trigger, we alter the pixel value of the trigger from 0.2 to 1.0.

Table 12: Defend against BadNets [15] Attack with Various Pixel Value

| Pixel | $L_\infty$ | No Defense ACC | No Defense ASR | ANP [55] ACC | ANP [55] ASR | SAU (**Ours**) ACC | SAU (**Ours**) ASR |
|---|---|---|---|---|---|---|---|
| 0.2 | 0.42 | 91.69 | 95.60 | 91.30 | 2.64 | 90.01 | **1.38** |
| 0.4 | 0.31 | 91.61 | 97.79 | 91.44 | **1.00** | 88.43 | 1.59 |
| 0.6 | 0.35 | 91.94 | 96.43 | 91.72 | 7.81 | 90.49 | **1.30** |
| 0.8 | 0.48 | 91.72 | 93.20 | 91.06 | 2.37 | 89.23 | **1.36** |
| 1.0 | 0.67 | 91.32 | 95.03 | 90.88 | 4.88 | 89.31 | **1.53** |

Table 12 shows the results for defending various BadNets attacks by SAU following the setting in Section 4.2, from which we can find that **SAU can consistently mitigate the backdoor even when the trigger is beyond the perturbation set**. One reasonable explanation is that although the poisoned samples are not included in SAEs, there exists SAE that has a similar effect as poisoned samples as shown in Figure 4, resulting in strong generalization ability of SAU.

**4. Effect of perturbation type.**  To explore the effect of perturbation types, we extend SAU to incorporate two other perturbation types and denote the resulting methods by **Spatial SAU** and **Ensemble SAU**, respectively. Specifically, for spatial SAU, we employ spatial adversarial attack [56] to generate spatial SAEs by spatial transformation, following the same setting as [13]. For Ensemble SAU, the perturbation set $S$ is composed of a collection of subsets, including $L_1$ ball (bound 500), $L_2$ ball (bound 10), $L_\infty$ ball (bound 0.2), and the spatial AP set. Then, for each batch, Ensemble SAU randomly chooses one subset of perturbation to generate the SAEs. We compare **SAU**, **Spatial SAU** and **Ensemble SAU**, following the setting in Section 4.

Table 13: Defend against BadNets [15] Attack with Various Pixel Value

| Dataset | Attack | No Defense ACC | No Defense ASR | SAU ACC | SAU ASR | Spatial SAU ACC | Spatial SAU ASR | Ensemble SAU ACC | Ensemble SAU ASR |
|---|---|---|---|---|---|---|---|---|---|
| CIFAR-10 | BadNets [15] | 91.32 | 95.03 | **89.31** | 1.53 | 88.27 | 0.97 | 88.16 | **0.67** |
| CIFAR-10 | SSBA [30] | 92.88 | 97.86 | **90.84** | 1.79 | 89.65 | 4.43 | 89.92 | **0.81** |
| CIFAR-10 | WaNet [38] | 91.25 | 89.73 | **91.26** | 1.02 | 90.21 | 4.34 | 89.61 | **0.66** |
| Tiny ImageNet | BadNets [15] | 56.23 | 100 | **51.52** | 0.53 | 50.68 | 99.81 | 49.32 | **0.49** |
| Tiny ImageNet | SSBA [30] | 55.22 | 97.71 | **51.85** | 0.11 | 49.79 | 79.72 | 49.73 | **0.05** |
| Tiny ImageNet | WaNet [38] | 56.78 | 54.65 | **54.65** | 85.75 | 50.12 | 0.7 | 51.14 | **0.15** |

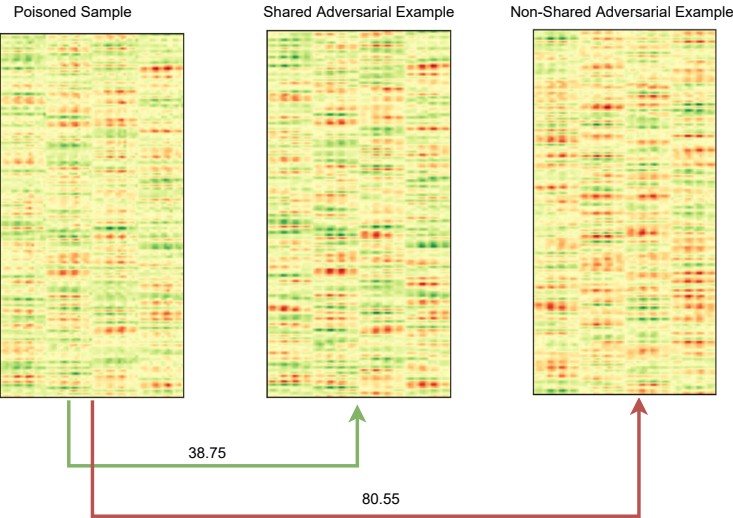

Figure 4: Visualization of Features of Poisoned Sample, Shared Adversarial Example and Non-shared Adversarial Example in latent space for BadNets Attack with PreAct-ResNet18. Both Shared Adversarial Example and Adversarial Example are generated by PGD-5 with $L_\infty$ norm 0.2. The $L_2$ distance between the features of the Poisoned sample and Shared Adversarial Example is 38.75, while the L2 distance between the features of the Poisoned sample and the Non-shared Adversarial Example is 80.55, showing that the Shared Adversarial Example can still be close to the Poisoned Sample in latent space even if the trigger is beyond perturbation set.

From Table 13, we compare SAU, Spatial SAU and Ensemble SAU, following the setting in Section 4. From the table, we find that Spatial SAU works well on CIFAR-10 dataset but fails to defend BadNets and SSBA attacks on Tiny ImageNet. For WaNet, spatial SAU can effectively defend it on both datasets. We remark that **the Ensemble SAU achieves the lowest ASR in all experiments**. However, as Ensemble SAU has larger perturbation set, its accuracy is lower than SAU by 1 2%.

In summary, our theoretical analysis and the proposed method SAU are flexible to the perturbation set, and SAU can be generalized to attack with excessive trigger norms. Moreover, the Ensemble SAU serves as a stronger variant of SAU and can be effective for various attacks.

### B.5   SAU with generated datasets

Like most other post-processing methods [31, 55, 47, 28, 59], SAU assumes that a small clean dataset is available for conducting defense, which can be obtained by ways such as buying from a trustworthy data supplier, using modern generative models, collecting from the internet, or applying some data cleansing techniques.

In this section, we explore a more practical case for defense, $i.e.$, using the generated dataset, to evaluate our method. Specifically, we conduct experiments on synthetic datasets and compare SAU with NAD. We test both methods with a synthetic dataset CIFAR-5m [36] which provides generated CIFAR-10-like images whose distribution is close but not identical to CIFAR-10, and can be regarded as OOD data. We build mixed datasets by randomly picking samples from CIFAR-10 and CIFAR-5m with a **mixed ratio**, and evaluate both methods on the mixed datasets. A larger mixed ratio indicates more synthetic data in the mixed dataset. The experiment results are summarized in the following tables.

From Table 14, we can find that **all methods can work on the mixed dataset but as the mixed ratio increase, the accuracy (ACC) may decrease due to the distribution difference between the two datasets.** It's notable that the model distillation-based method NAD is influenced by the quality of the dataset most, with a reduction of 4.91% in ACC, while SAU is quite robust to the data quality. Another interesting phenomenon is that the ASR of ANP decreases when more synthetic data is used, which shows that the distribution of data may influence the performance of backdoor defense.

Table 14: Results on Generated Dataset

| | ANP [55] | ANP [55] | NAD [28] | NAD [28] | SAU (**Ours**) | SAU (**Ours**) |
|---|---|---|---|---|---|---|
| Mixed Ratio | ACC | ASR | ACC | ASR | ACC | ASR |
| 0.0 | **90.88** | 4.88 | 89.87 | 2.14 | 89.31 | **1.53** |
| 0.2 | **90.68** | 2.10 | 87.96 | 1.88 | 89.83 | **1.63** |
| 0.4 | **90.21** | **1.11** | 87.69 | 1.88 | 89.36 | 1.60 |
| 0.6 | **90.31** | **1.51** | 86.69 | 2.19 | 88.44 | 1.66 |
| 0.8 | **89.37** | **0.56** | 85.90 | 1.86 | 88.13 | 1.79 |
| 1.0 | **88.74** | **0.40** | 84.96 | 2.48 | 88.32 | 1.01 |

All those observations motivate us to conduct research on the relationship between data quality and backdoor defense in the future.

## B.6 Resistance to defense-aware/adaptive Attacks

When the attacker is aware of the defender's adversarial training strategy, the attacker can train the backdoored model using Adversarial Training (AT) on the poisoned dataset, such that the model can be resistant to adversarial attack, hindering the generation of shared adversarial examples.

To evaluate the performance of SAU against such defense-aware backdoor attacks, we conduct experiments on the CIFAR-10 dataset following the settings in Section 4.2. For AT on the training dataset, we use PGD-10 attack with $L_\infty$ norm 8/255, which is commonly used in adversarial training literature.

Table 15: Experiments Results for Defense-aware Attack

| | No Defense | No Defense | SAU (**Ours**) | SAU (**Ours**) |
|---|---|---|---|---|
| Attack | ACC | ASR | ACC | ASR |
| BadNets [15] | 79.89 | 93.46 | 79.83 | 2.39 |
| Input-Aware | 79.87 | 96.60 | 78.64 | 4.84 |
| SIG [2] | 73.50 | 99.50 | 76.41 | 0.10 |
| Average | 77.75 | 96.52 | 78.30 | 2.44 |

From Table 15, we can find that SAU can achieve an average ACC of 78.3% and ASR of 2.44%, showing its resistance to such adaptive attacks.

## C Experiment details

In this section, we provide the experiment details. In our experiments, all baselines and settings are adapted from BackdoorBench [54], and we present the experiment details in this section.

## C.1 Attack details

- BadNets [15] is one of the earliest works for backdoor learning, which inserts a small patch of fixed pattern to replace some pixels in the image. We use the default setting in BackdoorBench.

- Blended backdoor attack (Blended) [7] uses an alpha-blending strategy to fuse images with fixed patterns. We set $\alpha = 0.2$ as the default in BackdoorBench. We remark that **since such a large $\alpha$ produces visual-perceptible change to clean samples, the Blended Attack in this setting is very challenging for all defense methods.**

- Input-aware dynamic backdoor attack (Input-Aware)[37] is a training-controllable attack that learns a trigger generator to produce the sample-specific trigger in the training process of the model. We use the default setting in BackdoorBench.

- Low-frequency attack (LF) [58] uses smoothed trigger by filtering high-frequency artifacts from a UAP. We use the default setting in BackdoorBench.

- Sinusoidal signal backdoor attack (SIG) [2] is a clean-label attack that uses a sinusoidal signal as the trigger to perturb the clean images in the target label. We use the default setting in BackdoorBench.

- Sample-specific backdoor attack (SSBA) [30] uses an auto-encoder to fuse a trigger into clean samples and generate poisoned samples. We use the default setting in BackdoorBench.

- Warping-based poisoned networks (WaNet) [38] is also a training-controllable attack that uses a warping function to perturb the clean samples to construct the poisoned samples. We use the default setting in BackdoorBench.

## C.2 Defense details

The details for each defense used in our experiment/discussion are summarized below:

- ANP [55] is a pruning-based method that prunes neurons sensitive to weight perturbation. Note that in BackdoorBench, ANP can select its pruning threshold by grid searching on the test dataset (default way in BackdoorBench) or a given constant threshold. To produce a fair comparison with other baselines, we use a constant threshold for ANP and set the pruning threshold to $0.4$, which is found to produce better results than the recommended constant threshold of $0.2$ in BackdoorBench. All other settings are the same as the default setting in BackdoorBench.

- FP [31] is a pruning-based method that prunes neurons according to their activations and then fine-tunes the model to keep clean accuracy. We use the default setting in BackdoorBench.

- NAD [28] uses Attention Distillation to mitigate backdoors. We use the default setting in BackdoorBench.

- NC [47] first optimizes a possible trigger to detect whether the model is backdoored. Then, if the model is detected as a backdoored model, it mitigates the backdoor by unlearning the optimized trigger. We use the default setting in BackdoorBench. We remark that **if the model is detected as a clean model, NC just returns the model without any changes.**

- EP [64] uses the distribution of activation to detect the backdoor neurons and prunes them to mitigate the backdoor. We use the default setting in BackdoorBench.

- i-BAU [59] uses adversarial training with UAP and hyper-gradient to mitigate the backdoor. We use the default setting in BackdoorBench.

- SAU (Ours) uses PGD to generate shared adversarial examples and unlearns them to mitigate the backdoor. For all experiments, we set $\lambda_1 = \lambda_2 = \lambda_4 = 1$ and $\lambda_3 = 0.01$ in (13). Then, we run PGD 5 steps with $L_\infty$ norm bound $0.2$ and a large step size $0.2$ to accelerate the inner maximization in (8). For unlearning step (fine-tuning on $\mathcal{D}_{cl}$ and SAEs), we use the same setting as i-BAU and fine-tuning (FT) in BackdoorBench. We run SAU 100 epochs in CIFAR-10 and GTSRB. In Tiny ImageNet, we run SAU 20 epochsNote that the number of epochs used in our experiment largely exceeds the necessary epochs for SAU to take effect and can be further reduced to save computational costs.

- Vanilla Adversarial Training (vanilla AT) [33] use PGD to generate vanilla adversarial examples and unlearn them to mitigate the backdoor. In all experiments, vanilla AT uses the same setting as SAU, including the step size, norm bound, optimizer, and learning rate, except for the experiment in Figure 1, where we train vanilla AT 100 epochs on Blended Attack with PreAct-ResNet18 and Tiny ImageNet to show its long-term performance.

- Adversarial Fine-tuning (AFT) [35] performs adversarial training on clean dataset, which is equivalent to vanilla AT.

- Composite Adversarial Training (Composite AT, CAT) [13] use perturbation from different types to defend against backdoor attacks. We request the official implementation of CAT from the authors and use the recommended setting in the original paper.

- Progressive Backdoor Erasing (PBE) [35] uses the adversarial attack to progressively filter the clean dataset from the training dataset and remove the backdoor. **Since the code haven't been released yet and the details of some critical parameters such as the configuration of adversarial attacks, the data filtering ratio in each step, and the learning rates are not covered in the paper, we failed to produce satisfactory results.** Besides, the threat

model of PBE is different from ours which makes it incomparable to SAU, and therefore, we do not include it in the experiment section.

**Adaptation to batch normalization.** Batch normalization (BN) has been widely used in modern deep neural networks due to improved convergence. However, recent works show that it is difficult for BN to estimate the correct normalization statistics of a mixture of distributions of adversarial examples and clean examples [57, 48]. Such a problem is magnified when we adversarially fine-tune a model with a small set of samples and may destroy the generalization ability of the model due to biased BN statistics. To address this problem, we propose **the fixed BN strategy** in the adversarial unlearning/training process. Note that in all experiments, the fixed BN strategy is applied to i-BAU, SAU, and vanilla AT when BN is used in the model architecture like PreAct-ResNet18.

**Experiments on VGG19.** For VGG19 [41], BN is not used. Therefore, the fixed BN strategy is not applied. Moreover, some methods that depend on BN, such as ANP and EP, are not applicable to VGG19.

# D  Additional experiments

In this section, we provide additional experiments.

## D.1  Main experiments on VGG19

Table 16: Results on CIFAR-10 with VGG19 and poisoning ratio 10%.

| Defense | No Defense | | | | FP [31] | | | | NC [47] | | | |
|---|---|---|---|---|---|---|---|---|---|---|---|---|
| Attack | ACC | ASR | R-ACC | DER | ACC | ASR | R-ACC | DER | ACC | ASR | R-ACC | DER |
| BadNets [15] | 89.36 | 95.93 | 3.81 | N/A | **89.23** | 92.61 | 6.82 | 51.6 | 87.86 | 1.00 | **88.01** | **96.72** |
| Blended [7] | 90.17 | 99.12 | 0.82 | N/A | **90.07** | 99.11 | 0.82 | 49.96 | 85.92 | 1.79 | 74.13 | 96.54 |
| Input-Aware [37] | 77.69 | 94.59 | 4.79 | N/A | **78.62** | 86.77 | 11.79 | 53.91 | 77.67 | 94.58 | 4.79 | 50.00 |
| LF [58] | 88.94 | 93.93 | 5.62 | N/A | **88.98** | 91.8 | 7.46 | 51.07 | 85.35 | 9.99 | 72.79 | 90.18 |
| SIG [2] | 81.69 | 99.8 | 0.12 | N/A | 84.52 | 99.93 | 0.07 | 50.0 | 81.69 | 99.80 | 0.12 | 50.00 |
| SSBA [30] | 89.48 | 91.86 | 7.29 | N/A | 89.40 | 89.66 | 9.22 | 51.06 | **89.48** | 91.86 | 7.29 | 50.00 |
| WaNet [38] | 88.43 | 88.9 | 10.3 | N/A | 89.61 | 73.39 | 24.57 | 57.76 | 88.43 | 88.89 | 10.30 | 50.01 |
| Average | 86.54 | 94.88 | 4.68 | N/A | **87.20** | 90.47 | 8.68 | 52.19 | 85.20 | 55.42 | 36.78 | 69.06 |

| Defense | NAD [28] | | | | i-BAU [59] | | | | SAU (**Ours**) | | | |
|---|---|---|---|---|---|---|---|---|---|---|---|---|
| Attack | ACC | ASR | R-ACC | DER | ACC | ASR | R-ACC | DER | ACC | ASR | R-ACC | DER |
| BadNets [15] | 87.51 | 38.17 | 58.30 | 77.96 | 87.82 | 25.72 | 63.24 | 84.34 | 86.71 | **0.08** | 32.77 | 96.60 |
| Blended [7] | 88.35 | 93.08 | 6.33 | 52.11 | 88.61 | 59.86 | 32.31 | 68.85 | 87.01 | 1.32 | 23.74 | **97.32** |
| Input-Aware [37] | 75.70 | 23.36 | 54.71 | 84.62 | 72.15 | 26.22 | 42.51 | 81.41 | 75.11 | 12.49 | 46.19 | 89.76 |
| LF [58] | 87.85 | 59.38 | 35.01 | 66.73 | 87.97 | 79.10 | 18.04 | 56.93 | 86.65 | 6.43 | 72.51 | 92.60 |
| SIG [2] | 86.01 | 99.18 | 0.77 | 50.31 | 85.06 | 90.89 | 5.27 | 54.46 | 84.87 | 0.11 | 10.82 | **99.84** |
| SSBA [30] | 87.65 | 37.54 | 54.58 | 76.24 | 88.08 | 3.74 | 76.98 | 93.36 | 87.3 | 1.66 | 78.70 | 94.01 |
| WaNet [38] | 90.82 | 44.93 | 51.18 | 71.98 | 90.31 | 25.83 | 67.87 | 81.53 | 87.07 | 5.32 | 83.37 | 91.11 |
| Average | 86.27 | 56.52 | 37.27 | 68.56 | 85.71 | 44.48 | 43.75 | 74.41 | 84.96 | **3.92** | 49.73 | **94.46** |

This section provides additional experiment results on VGG19 with CIFAR-10 (Table 16) and Tiny ImageNet (Table 17) to supplement Section 4. Table 16 shows that SAU outperforms all baselines on CIFAR-10 and VGG19, with a 40.56% decrease of ASR compared to the second lowest ASR. Moreover, SAU achieves the highest DER in 6 of 7 attacks, demonstrating a significantly better tradeoff between accuracy and ASR. It also achieves the top-2 R-ACC in 5 of 7 attacks and the best average R-ACC, indicating its good ability to recover the prediction of poisoned samples. Table 17 shows that SAU achieves the best DER in 5 of 6 attacks and the top-2 lowest ASR in 5 of 6 attacks. Although SAU only achieves the second-best average ASR in Table 17, it has significantly higher accuracy than NC, which achieves the best average ASR.

## D.2  Main experiments on GTSRB

This section provides additional experiment results on GTSRB with PreAct-ResNet18 (Table 18) and VGG19 (Table 19) to supplement Section 4. In both tables, SAU achieves the top-2 lowest ASR in 4 of 6 attacks and the lowest ASR on average. Furthermore, SAU achieves the top-2 highest DER in 4 of 6 attacks for PreAct-ResNet18 (Table 18) and 5 of 6 attacks for VGG19 (Table 19).

Table 17: Results on Tiny ImageNet with VGG19 and poisoning ratio $10\%$.

| Defense | No Defense | | | | FP [31] | | | | NC [47] | | | |
|---|---|---|---|---|---|---|---|---|---|---|---|---|
| Attack | ACC | ASR | R-ACC | DER | ACC | ASR | R-ACC | DER | ACC | ASR | R-ACC | DER |
| BadNets [15] | 42.26 | 99.99 | 0.0 | N/A | **40.70** | 99.63 | 0.20 | 49.4 | 26.47 | 0.53 | **25.93** | 91.83 |
| Blended [7] | 43.5 | 99.32 | 0.32 | N/A | **41.82** | 98.67 | 0.71 | 49.48 | 35.93 | 0.03 | 18.87 | 95.86 |
| Input-Aware [37] | 46.36 | 99.48 | 0.38 | N/A | **45.95** | 99.02 | 0.69 | 50.03 | 38.51 | 0.45 | 36.00 | 95.59 |
| LF [58] | 43.14 | 95.41 | 2.3 | N/A | **41.63** | 85.22 | 5.82 | 54.34 | 10.55 | 0.77 | 5.37 | 81.02 |
| SSBA [30] | 41.67 | 96.57 | 1.71 | N/A | **40.04** | 69.27 | 9.10 | 62.84 | 28.23 | 0.21 | 22.86 | 91.46 |
| WaNet [38] | 43.6 | 99.85 | 0.09 | N/A | **42.85** | 99.31 | 0.46 | 49.9 | 8.14 | 1.17 | 7.13 | 81.61 |
| Average | 43.42 | 98.44 | 0.8 | N/A | **42.16** | 91.85 | 2.83 | 52.28 | 24.64 | 0.53 | 19.36 | 83.91 |

| Defense | NAD [28] | | | | i-BAU [59] | | | | SAU (**Ours**) | | | |
|---|---|---|---|---|---|---|---|---|---|---|---|---|
| Attack | ACC | ASR | R-ACC | DER | ACC | ASR | R-ACC | DER | ACC | ASR | R-ACC | DER |
| BadNets [15] | 37.68 | 96.40 | 2.02 | 49.5 | 40.10 | 100.0 | 0.0 | 48.92 | 38.25 | **0.06** | 14.32 | **97.96** |
| Blended [7] | 38.59 | 95.98 | 1.64 | 49.21 | 40.11 | 98.88 | 0.47 | 48.52 | 38.24 | **0.01** | 6.10 | **97.02** |
| Input-Aware [37] | 36.14 | 2.79 | 31.87 | 93.23 | 44.49 | 58.66 | 18.85 | 69.47 | 42.87 | 6.94 | 35.15 | 94.52 |
| LF [58] | 38.06 | 81.10 | 7.40 | 54.62 | 40.23 | 86.89 | 6.05 | 52.8 | 39.15 | 15.46 | 20.31 | 87.98 |
| SSBA [30] | 36.27 | 84.86 | 5.78 | 53.15 | 38.93 | 87.81 | 4.96 | 53.01 | 36.31 | 0.32 | 8.98 | **95.45** |
| WaNet [38] | 34.89 | 63.91 | 13.98 | 63.61 | 39.82 | 98.69 | 0.62 | 48.69 | 37.04 | 2.52 | 20.02 | **95.38** |
| Average | 36.94 | 70.84 | 10.45 | 59.05 | 40.61 | 88.49 | 5.16 | 53.06 | 38.64 | 4.22 | 17.48 | **88.33** |

Table 18: Results on GTSRB with PreAct-ResNet18 and poisoning ratio $10\%$.

| Defense | No Defense | | | | ANP [55] | | | | FP [31] | | | | NC [47] | | | |
|---|---|---|---|---|---|---|---|---|---|---|---|---|---|---|---|---|
| Attack | ACC | ASR | R-ACC | DER | ACC | ASR | R-ACC | DER | ACC | ASR | R-ACC | DER | ACC | ASR | R-ACC | DER |
| BadNets [15] | 97.24 | 59.25 | 4.42 | N/A | 96.89 | 0.06 | 96.79 | 79.42 | 98.21 | 0.09 | 70.93 | 79.58 | 97.48 | **0.01** | 97.45 | **79.62** |
| Blended [7] | 98.58 | 99.99 | 0.0 | N/A | **98.75** | 99.82 | 0.18 | 50.09 | 98.38 | 100.0 | 0.0 | 49.9 | 97.76 | 8.03 | 59.74 | 95.57 |
| Input-Aware [37] | 97.26 | 92.74 | 7.23 | N/A | **99.14** | **0.00** | 96.09 | 96.37 | 98.08 | 2.32 | 88.91 | 95.21 | 98.55 | 0.01 | 96.71 | 96.36 |
| LF [58] | 97.93 | 99.57 | 0.42 | N/A | 97.80 | 81.38 | 16.45 | 59.03 | 97.59 | 99.7 | 0.27 | 49.83 | 97.97 | 1.34 | 77.37 | 99.11 |
| SSBA [30] | 97.98 | 99.56 | 0.49 | N/A | 97.86 | 98.73 | 1.13 | 50.36 | 97.75 | 99.46 | 0.45 | 49.94 | 97.72 | 0.29 | 87.30 | 99.50 |
| WaNet [38] | 97.74 | 94.25 | 5.59 | N/A | 98.00 | **0.00** | 97.08 | **97.12** | 97.62 | 88.07 | 8.71 | 53.03 | 98.25 | 0.00 | 98.03 | 97.12 |
| Average | 97.79 | 90.89 | 3.02 | N/A | 98.07 | 46.66 | 51.29 | 68.91 | 97.94 | 64.94 | 28.21 | 61.07 | 97.96 | 1.61 | 86.10 | 88.18 |

| Defense | NAD [28] | | | | EP [64] | | | | i-BAU [59] | | | | SAU (**Ours**) | | | |
|---|---|---|---|---|---|---|---|---|---|---|---|---|---|---|---|---|
| Attack | ACC | ASR | R-ACC | DER | ACC | ASR | R-ACC | DER | ACC | ASR | R-ACC | DER | ACC | ASR | R-ACC | DER |
| BadNets [15] | **98.69** | 0.63 | **98.00** | 79.31 | 97.69 | 10.94 | 88.22 | 74.16 | 96.03 | 0.02 | 96.07 | 79.01 | 98.0 | 0.01 | 97.92 | 79.62 |
| Blended [7] | 98.61 | 100.0 | 0.00 | 50.0 | 97.9 | 100.0 | 0.0 | 49.66 | 95.65 | 86.45 | 6.52 | 55.31 | 96.73 | 3.41 | 34.42 | **97.36** |
| Input-Aware [37] | 98.27 | 40.65 | 58.68 | 76.04 | 98.5 | 49.49 | 49.5 | 71.62 | 97.93 | 1.94 | 93.58 | 95.4 | 98.09 | **0.01** | 93.02 | 96.36 |
| LF [58] | **98.14** | 51.83 | 40.02 | 73.87 | 97.59 | 99.2 | 0.72 | 50.02 | 95.59 | 17.3 | 32.16 | 89.97 | 96.08 | **0.37** | 16.89 | 98.68 |
| SSBA [30] | 97.95 | 99.39 | 0.59 | 50.07 | 97.66 | 98.66 | 1.13 | 50.29 | 96.13 | 1.27 | 58.10 | 98.22 | 96.94 | 0.45 | 40.33 | 99.04 |
| WaNet [38] | 98.61 | 0.56 | 97.63 | 96.84 | 96.96 | 8.97 | 82.93 | 92.25 | 96.92 | 0.11 | 96.25 | 96.66 | 97.86 | 0.02 | 97.42 | 97.11 |
| Average | **98.38** | 48.84 | 49.15 | 68.02 | 97.72 | 61.21 | 37.08 | 62.57 | 96.38 | 17.85 | 63.78 | 80.65 | 97.28 | **0.71** | 63.33 | **88.31** |

## D.3 Experiments with 5% poisoning ratio

This section reports the experiment results for comparing SAU with baselines with a poisoning ratio of 5% on CIFAR-10 and Tiny-ImageNet. As with the results for a poisoning ratio of 10%, SAU achieves the top-2 lowest ASR in most of the attacks. Tables 20 and 21 show that SAU achieves the lowest average ASR. In summary, SAU still outperforms other baselines for a poisoning ratio of 5%.

## D.4 Learning curves for Shared Adversarial Risk

This section presents a plot of the learning curves for shared adversarial risk to illustrate the relationship between the backdoor risk (measured by ASR) and the shared adversarial risk. For each plot, we run SAU on CIFAR-10 with PreAct-ResNet18 and a poisoning ratio of 10% for 10 epochs. The SAR is computed on the test dataset by computing the shared adversarial example using the inner maximization step in (8). Figure 5 demonstrates that SAU can effectively reduce the shared adversarial risk, and thus, mitigate the backdoor risk. We note that in order to satisfy Assumption 1, the perturbation set adopted by SAU is much larger than the perturbation budget in the adversarial robustness area, resulting in a high adversarial risk in Figure 5.

## D.5 Experiments on different PGD configurations

In this section, we test SAU using different PGD configurations to generate the shared adversarial examples. Specifically, we consider the following configurations for PGD:

- Config 1: PGD with $L_\infty$ norm bound 0.2, step size 0.2, and 5 steps.

Table 19: Results on GTSRB with VGG19 and poisoning ratio 10%.

| Defense | No Defense | | | | FP [31] | | | | NC [47] | | | |
|---|---|---|---|---|---|---|---|---|---|---|---|---|
| Attack | ACC | ASR | R-ACC | DER | ACC | ASR | R-ACC | DER | ACC | ASR | R-ACC | DER |
| BadNets [15] | 96.25 | 57.11 | 4.88 | N/A | **97.02** | 2.89 | 18.64 | 77.11 | 96.86 | **0.00** | 96.64 | 78.56 |
| Blended [7] | 95.98 | 99.86 | 0.04 | N/A | 96.48 | 99.79 | 0.08 | 50.03 | 95.32 | 20.80 | 38.37 | 89.20 |
| Input-Aware [37] | 96.03 | 76.69 | 22.35 | N/A | 97.08 | 24.09 | 73.09 | 76.3 | 96.71 | 0.02 | 94.22 | 88.33 |
| LF [58] | 95.05 | 98.79 | 0.99 | N/A | 95.28 | 98.13 | 1.48 | 50.33 | 93.25 | 0.60 | 67.61 | 98.19 |
| SSBA [30] | 96.43 | 99.31 | 0.43 | N/A | 96.45 | 99.0 | 0.64 | 50.16 | **96.56** | 1.03 | 79.49 | 99.14 |
| WaNet [38] | 95.27 | 92.09 | 7.35 | N/A | 96.90 | 68.48 | 29.00 | 61.81 | 96.84 | 1.11 | 94.67 | 95.49 |
| Average | 95.84 | 87.31 | 6.01 | N/A | 96.54 | 65.4 | 20.49 | 59.39 | 95.92 | 3.93 | 78.50 | 85.56 |

| Defense | NAD [28] | | | | i-BAU [59] | | | | SAU (**Ours**) | | | |
|---|---|---|---|---|---|---|---|---|---|---|---|---|
| Attack | ACC | ASR | R-ACC | DER | ACC | ASR | R-ACC | DER | ACC | ASR | R-ACC | DER |
| BadNets [15] | 96.78 | 73.52 | 25.73 | 50.0 | 95.81 | 0.01 | 94.70 | 78.33 | 96.21 | 0.02 | 79.69 | 78.53 |
| Blended [7] | **96.60** | 98.27 | 0.55 | 50.79 | 95.11 | **0.79** | 16.55 | 99.10 | 95.45 | 1.79 | 12.61 | 98.77 |
| Input-Aware [37] | 97.77 | 53.03 | 45.94 | 61.83 | 96.92 | 10.06 | 72.08 | 83.31 | 96.48 | 0.01 | 11.34 | **88.34** |
| LF [58] | 94.69 | 97.07 | 2.43 | 50.68 | **95.51** | 0.83 | 22.43 | 98.98 | 95.17 | 0.02 | 15.83 | **99.38** |
| SSBA [30] | 96.38 | 98.38 | 0.95 | 50.44 | 94.57 | 2.55 | 61.79 | 97.45 | 95.92 | 7.44 | 31.49 | 95.68 |
| WaNet [38] | 97.47 | 39.34 | 57.96 | 76.38 | 96.86 | 6.28 | 81.71 | 92.90 | 97.04 | 5.89 | 88.15 | 93.10 |
| Average | **96.62** | 76.6 | 22.26 | 55.73 | 95.80 | 3.42 | 58.21 | 85.72 | 96.04 | **2.53** | 39.85 | **86.26** |

Table 20: Results on CIFAR-10 with PreAct-ResNet18 and poisoning ratio 5%.

| Defense | No Defense | | | | ANP [55] | | | | FP [31] | | | | NC [47] | | | |
|---|---|---|---|---|---|---|---|---|---|---|---|---|---|---|---|---|
| Attack | ACC | ASR | R-ACC | DER | ACC | ASR | R-ACC | DER | ACC | ASR | R-ACC | DER | ACC | ASR | R-ACC | DER |
| BadNets [15] | 92.64 | 88.74 | 10.78 | N/A | 92.38 | 3.10 | 90.13 | 92.69 | **92.47** | 17.47 | 77.9 | 85.55 | 89.89 | 1.17 | 89.70 | 92.41 |
| Blended [7] | 93.67 | 99.61 | 0.39 | N/A | 93.32 | 98.00 | 1.92 | 50.63 | 93.46 | 98.87 | 1.11 | 50.27 | **93.66** | 99.61 | 0.39 | 50.00 |
| Input-Aware [37] | 91.52 | 90.2 | 8.91 | N/A | 91.64 | **1.08** | 87.01 | 94.56 | 91.89 | 52.74 | 44.49 | 68.73 | 91.51 | 90.20 | 8.91 | 50.00 |
| LF [58] | 93.35 | 98.03 | 1.86 | N/A | 92.60 | 96.98 | 2.87 | 50.15 | 93.14 | 98.37 | 1.53 | 49.9 | **93.35** | 98.03 | 1.86 | 50.00 |
| SIG [2] | 93.64 | 97.09 | 2.9 | N/A | 93.09 | 92.38 | 7.39 | 52.08 | 93.27 | 99.79 | 0.2 | 49.81 | **93.65** | 97.09 | 2.90 | 50.00 |
| SSBA [30] | 93.27 | 94.91 | 4.82 | N/A | **93.13** | 52.66 | 43.68 | 71.06 | 93.05 | 87.14 | 12.09 | 53.77 | 91.04 | **0.80** | 87.93 | 95.94 |
| WaNet [38] | 91.76 | 85.5 | 13.49 | N/A | 90.67 | **2.31** | 88.31 | 91.05 | 92.14 | 26.1 | 63.5 | 79.7 | 91.76 | 85.50 | 13.49 | 50.00 |
| Average | 92.84 | 93.44 | 6.16 | N/A | 92.40 | 49.50 | 45.90 | 71.75 | **92.77** | 68.64 | 28.69 | 62.53 | 92.12 | 67.49 | 29.31 | 62.62 |

| Defense | NAD [28] | | | | EP [64] | | | | i-BAU [59] | | | | SAU (**Ours**) | | | |
|---|---|---|---|---|---|---|---|---|---|---|---|---|---|---|---|---|
| Attack | ACC | ASR | R-ACC | DER | ACC | ASR | R-ACC | DER | ACC | ASR | R-ACC | DER | ACC | ASR | R-ACC | DER |
| BadNets [15] | 91.03 | 4.73 | 87.93 | 91.2 | 91.04 | **0.98** | **91.26** | 93.08 | 88.8 | 2.51 | 87.82 | 91.20 | 90.54 | 1.36 | 90.32 | 92.64 |
| Blended [7] | 93.10 | 99.06 | 0.93 | 49.99 | 92.07 | 98.88 | 1.01 | 49.57 | 88.39 | 35.89 | 43.28 | 79.22 | 91.25 | 5.64 | 67.56 | 95.77 |
| Input-Aware [37] | **93.07** | 97.12 | 2.68 | 50.0 | 90.59 | 1.51 | 85.09 | 93.88 | 91.34 | 7.63 | 80.10 | 91.19 | 91.12 | 2.03 | 82.11 | 93.88 |
| LF [58] | 92.98 | 94.23 | 5.3 | 51.72 | 91.65 | 83.71 | 15.22 | 56.31 | 87.25 | 34.61 | 41.31 | 78.66 | 91.31 | 2.59 | 78.93 | 96.70 |
| SIG [2] | 92.49 | 96.98 | 2.93 | 49.48 | 92.4 | 1.86 | **60.29** | 97.00 | 86.44 | 5.73 | 48.86 | 92.08 | 90.87 | **0.97** | 54.98 | 96.68 |
| SSBA [30] | 92.49 | 88.63 | 10.58 | 52.75 | 92.29 | 12.14 | 78.70 | 90.89 | 87.15 | 3.86 | 78.57 | 92.47 | 91.15 | 1.43 | 85.87 | 95.68 |
| WaNet [38] | **93.31** | 50.4 | 46.46 | 67.55 | 91.36 | 31.98 | 62.87 | 76.56 | 89.71 | 4.24 | 83.68 | 89.60 | 91.69 | 3.58 | 88.33 | 90.93 |
| Average | 92.64 | 75.88 | 22.4 | 58.96 | 91.63 | 33.01 | 56.35 | 79.61 | 88.44 | 13.50 | 66.23 | 87.77 | 91.13 | **2.51** | 78.30 | **94.61** |

- Config 2: PGD with $L_\infty$ norm bound 0.1, step size 0.1, and 5 steps.

- Config 3: PGD with $L_\infty$ norm bound 0.2, step size 0.1, and 5 steps.

- Config 4: PGD with $L_2$ norm bound 5, step size 0.2, and 5 steps.

- Config 5: PGD with $L_1$ norm bound 300, step size 0.2, and 5 steps.

The experiments are conducted for CIFAR-10 with PreAct-ResNet18 and poisoning ratio 10%. We summarize the experiment results in Table 22, from which we can find the SAU can effectively mitigate backdoors under various configurations of PGD.

Table 21: Results on Tiny ImageNet with PreAct-ResNet18 and poisoning ratio 5%.

| Defense | No Defense | | | | ANP [55] | | | | FP [31] | | | | NC [47] | | | |
|---|---|---|---|---|---|---|---|---|---|---|---|---|---|---|---|---|
| Attack | ACC | ASR | R-ACC | DER | ACC | ASR | R-ACC | DER | ACC | ASR | R-ACC | DER | ACC | ASR | R-ACC | DER |
| BadNets [15] | 56.36 | 99.86 | 0.13 | N/A | 44.54 | **0.02** | 43.80 | 94.01 | 53.00 | 96.83 | 2.86 | 49.83 | 52.97 | 65.94 | 25.45 | 65.26 |
| Blended [7] | 56.69 | 98.75 | 0.94 | N/A | 45.19 | 92.89 | 3.67 | 47.18 | 52.96 | 94.47 | 3.21 | 50.28 | **53.63** | 96.52 | 2.21 | 49.59 |
| Input-Aware [37] | 57.87 | 98.25 | 1.51 | N/A | 49.38 | 0.17 | 48.84 | 94.80 | 56.27 | 89.87 | 6.52 | 53.39 | 56.48 | 0.49 | **53.17** | 98.18 |
| LF [58] | 56.67 | 95.99 | 2.75 | N/A | 41.26 | **38.48** | 17.21 | 71.05 | 53.11 | 86.8 | 6.47 | 52.81 | 51.49 | 67.51 | 16.57 | 61.65 |
| SSBA [30] | 56.86 | 95.69 | 3.05 | N/A | 44.93 | 50.44 | 20.23 | 66.66 | 53.26 | 79.97 | 10.63 | 56.06 | 54.04 | 0.04 | 38.80 | 96.41 |
| WaNet [38] | 56.83 | 98.36 | 1.21 | N/A | 29.42 | **0.03** | 27.29 | 85.46 | 54.18 | 93.36 | 2.09 | 51.18 | 53.02 | 0.27 | 50.56 | 97.14 |
| Average | 56.88 | 97.82 | 1.6 | N/A | 42.45 | 30.34 | 26.84 | 72.74 | 53.80 | 90.22 | 5.3 | 51.94 | 53.60 | 38.46 | 31.13 | 74.03 |

| Defense | NAD [28] | | | | EP [64] | | | | i-BAU [59] | | | | SAU (Ours) | | | |
|---|---|---|---|---|---|---|---|---|---|---|---|---|---|---|---|---|
| Attack | ACC | ASR | R-ACC | DER | ACC | ASR | R-ACC | DER | ACC | ASR | R-ACC | DER | ACC | ASR | R-ACC | DER |
| BadNets [15] | 45.51 | 34.63 | 34.01 | 77.19 | 51.58 | 0.10 | **51.06** | 97.49 | 53.60 | 96.58 | 3.17 | 50.26 | 51.94 | 0.78 | 50.78 | 97.33 |
| Blended [7] | 46.72 | 88.89 | 4.96 | 49.94 | 53.13 | 82.26 | 8.58 | 56.47 | 52.76 | 92.01 | 4.7 | 51.41 | 52.47 | **3.40** | 24.90 | 95.57 |
| Input-Aware [37] | 48.72 | 1.04 | 45.83 | 94.03 | 57.09 | 0.92 | 51.27 | 98.27 | 56.13 | 36.99 | 40.9 | 79.76 | 53.92 | **0.11** | 50.89 | 97.10 |
| LF [58] | 46.93 | 63.9 | 14.66 | 61.18 | 54.04 | 83.80 | 10.06 | 54.78 | 52.45 | 90.81 | 5.77 | 50.48 | 52.57 | 44.27 | 23.78 | 73.81 |
| SSBA [30] | 46.65 | 42.45 | 24.65 | 71.51 | 53.87 | 52.07 | 23.44 | 70.31 | 52.65 | 90.72 | 5.95 | 50.38 | 53.57 | **0.02** | 39.77 | 96.19 |
| WaNet [38] | 46.81 | 2.44 | 39.54 | 92.95 | 53.98 | 0.12 | **52.62** | 97.70 | 52.99 | 93.43 | 4.93 | 50.55 | 50.15 | 81.75 | 12.37 | 54.97 |
| Average | 46.89 | 38.89 | 27.27 | 70.97 | **53.95** | 36.54 | 32.84 | 75.00 | 53.43 | 83.42 | 10.9 | 54.69 | 52.44 | 21.72 | 33.75 | 80.71 |

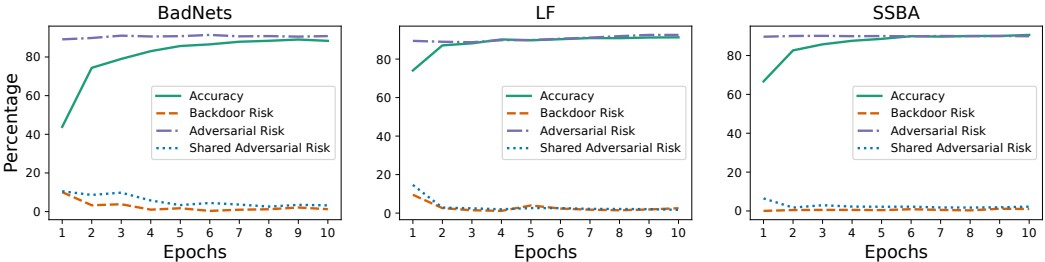

Figure 5: Learn curves for Accuracy, Backdoor Risk, Adversarial Risk, and Shared Adversarial Risk.

Table 22: Results for SAU with different PGD configurations

| Defense | No Defense | | | | Config 1 (**Default**) | | | | Config 2 | | | |
|---|---|---|---|---|---|---|---|---|---|---|---|---|
| Attack | ACC | ASR | R-ACC | DER | ACC | ASR | R-ACC | DER | ACC | ASR | R-ACC | DER |
| BadNets [15] | 91.32 | 95.03 | 4.67 | N/A | 89.31 | 1.53 | 88.81 | 95.74 | 89.56 | **0.80** | 89.60 | 96.24 |
| Blended [7] | 93.47 | 99.92 | 0.08 | N/A | 90.96 | 6.14 | 64.89 | 95.63 | 91.05 | 6.06 | 61.11 | 95.72 |
| Input-Aware | 90.67 | 98.26 | 1.66 | N/A | **91.59** | 1.27 | **88.54** | 98.49 | 90.93 | 1.56 | 84.76 | 98.35 |
| LF [58] | 93.19 | 99.28 | 0.71 | N/A | 90.32 | 4.18 | 81.54 | 96.12 | 90.83 | 3.76 | 77.93 | 96.58 |
| SIG [2] | 84.48 | 98.27 | 1.72 | N/A | 88.56 | 1.67 | 57.96 | 98.30 | 88.29 | 0.94 | 49.41 | 98.66 |
| SSBA [30] | 92.88 | 97.86 | 1.99 | N/A | 90.84 | 1.79 | 85.83 | 97.01 | 89.81 | 1.63 | 80.29 | 96.58 |
| WaNet [38] | 91.25 | 89.73 | 9.76 | N/A | 91.26 | **1.02** | 90.28 | 94.36 | 86.22 | 3.12 | 86.89 | 90.79 |
| Average | 91.04 | 96.91 | 2.94 | N/A | 90.41 | 2.51 | 79.69 | 96.52 | 89.53 | 2.55 | 75.71 | 96.13 |

| Defense | Config 3 | | | | Config 4 | | | | Config 5 | | | |
|---|---|---|---|---|---|---|---|---|---|---|---|---|
| Attack | ACC | ASR | R-ACC | DER | ACC | ASR | R-ACC | DER | ACC | ASR | R-ACC | DER |
| BadNets [15] | 89.69 | 2.21 | 89.19 | 95.60 | **89.78** | 1.11 | 89.38 | 96.19 | 89.04 | 1.29 | 88.63 | 95.73 |
| Blended [7] | 90.98 | 6.12 | 61.64 | 95.65 | **91.58** | 8.31 | 59.09 | 94.86 | 91.49 | **5.83** | 63.04 | **96.05** |
| Input-Aware | 90.75 | 1.86 | 86.21 | 98.20 | 91.30 | **0.94** | 84.24 | **98.66** | 91.21 | 1.20 | 86.60 | 98.53 |
| LF [58] | 91.05 | 3.00 | 81.23 | **97.07** | 87.06 | **1.94** | 73.72 | 95.60 | **91.29** | 5.12 | 80.26 | 96.13 |
| SIG [2] | 87.18 | 1.32 | 55.12 | 98.47 | 88.20 | **0.78** | 49.09 | **98.74** | 86.69 | 2.08 | 55.64 | 98.09 |
| SSBA [30] | 90.39 | 2.68 | 82.04 | 96.34 | 90.65 | 2.59 | 84.40 | 96.52 | 90.48 | **1.62** | 84.19 | 96.92 |
| WaNet [38] | 91.14 | 1.60 | 87.96 | 94.01 | 90.79 | 1.13 | 89.70 | 94.07 | 90.55 | 1.82 | 89.11 | 93.61 |
| Average | 90.17 | 2.68 | 77.63 | 96.48 | 89.91 | **2.40** | 75.66 | 96.38 | 90.11 | 2.71 | 78.21 | 96.44 |

## D.6 Experiments on different numbers of clean samples

This section examines the influence of clean sample size for SAU by evaluating SAU with different numbers of clean samples. The experiment is conducted on CIFAR-10 with PreAct-ResNet18 and the results are summarized in Table 23. Table 23 shows that SAU can consistently mitigate backdoors with sample sizes ranging from 2500 (5%) to 500 (1%) with high accuracy. When the sample size decreases to 50 (0.1%, 5 samples per class) or 10 (0.02%, 1 sample per class), SAU can still reduce

the backdoor to a low ASR. However, clean accuracy is difficult to guarantee with such limited clean samples.

Table 23: Results for SAU with different numbers of clean samples

| Defense | No Defense | | SAU -2500 | | SAU -1000 | | SAU -500 | | SAU -50 | | SAU -10 | |
|---|---|---|---|---|---|---|---|---|---|---|---|---|
| Attack | ACC | ASR | ACC | ASR | ACC | ASR | ACC | ASR | ACC | ASR | ACC | ASR |
| BadNets [15] | 91.32 | 95.03 | 89.31 | 1.53 | 89.24 | 1.31 | 83.04 | 2.17 | 67.86 | 1.41 | 51.43 | 14.38 |
| Blended [7] | 93.47 | 99.92 | 90.96 | 6.14 | 88.82 | 2.53 | 87.71 | 5.41 | 73.95 | 11.62 | 43.54 | 0.00 |
| LF [58] | 93.19 | 99.28 | 90.32 | 4.18 | 88.00 | 6.79 | 88.69 | 2.84 | 66.69 | 0.30 | 54.95 | 12.62 |

## D.7 Experiments on Multi-trigger Attacks

In this section, we evaluate our method for the multi-trigger/multi-target cases. Specifically, we use two different attacks (denoted by Attack-1 and Attack-2) with different target labels. Following Section 4.2, we test our method (SAU) against a strong baseline ANP on the CIFAR-10 dataset with a poisoning ratio of 10% and backbone PreAct-ResNet18. The experiment results are summarized in the following table, where we use ASR-1 and ASR-2 to represent the Attack Success Rate for Attack-1 and Attack-2, respectively. From Table 24, we can find that SAU achieves much higher accuracy (5.64% higher) and a much lower ASR for Attack-1 (26.46% lower) compared to ANP, although the average ASR for Attack-2 of ANP is slightly lower than that of SAU (only 0.13% lower). These results show that SAU outperforms the baseline in most cases, which demonstrates the effectiveness and robustness of our method in this challenging scenario.

Table 24: Results for defending against Multi-trigger Attacks

| Attack-1 | Attack-2 | No Defense | No Defense | No Defense | ANP [55] | ANP [55] | ANP [55] | SAU (**Ours**) | SAU (**Ours**) | SAU (**Ours**) |
|---|---|---|---|---|---|---|---|---|---|---|
| | | ACC | ASR-1 | ASR-2 | ACC | ASR-1 | ASR-2 | ACC | ASR-1 | ASR-2 |
| Blended [7] | BadNets [15] | 90.22 | 99.27 | 95.03 | 85.55 | 37.07 | **0.01** | **88.93** | **2.21** | 0.54 |
| InputAware | BadNets [15] | 89.71 | 78.62 | 94.77 | 82.21 | 4.48 | **0.01** | **88.88** | **4.14** | 0.70 |
| LF [58] | BadNets [15] | 90.16 | 98.08 | 95.15 | 84.21 | 79.11 | 2.30 | **88.45** | **2.93** | **0.34** |
| SIG [2] | BadNets [15] | 82.12 | 98.48 | 95.03 | 76.43 | 26.92 | 0.14 | 85.82 | **0.60** | 0.67 |
| SSBA [30] | BadNets [15] | 89.69 | 95.13 | 95.11 | 83.25 | 20.03 | 0.14 | 87.60 | 1.38 | 0.52 |
| WaNet [38] | BadNets [15] | 89.44 | 90.21 | 95.40 | 82.21 | 5.53 | **0.01** | **88.02** | **3.12** | 0.67 |
| Average | | 88.56 | 93.30 | 95.08 | 82.31 | 28.86 | **0.44** | 87.95 | 2.40 | 0.57 |

## D.8 Experiments on ALL to ALL attack

In this section, we compare SAU with other baselines on ALL to ALL attacks on CIFAR-10 with PreAct-ResNet18 and poisoning ratio 10%. Specifically, the target labels for the sample with original labels $y$ are set to $y_t = (y + 1) \mod K$ where $\mod$ is short for "modulus". The experiment results are summarized in Table 25. From Table 25, we can find that SAU achieves the best defending performance in 5 of 7 attacks and the lowest average ASR. At the same time, SAU also achieves the best average R-ACC and average DER, which further demonstrates its effectiveness in defending against backdoor attacks with multiple targets.

Note that the ASR for the poisoned model is significantly lower than the ASR in the single-target case. Thus, Table 25 also shows that SAU can still effectively mitigate backdoor even the ASR of the poisoned model is much lower than 100%.

## D.9 Comparison to Reconstructive Neuron Pruning (RNP)

In this section, we compare SAU with RNP [29] to further improve our submission. We adopted the official implementation of RNP (https://github.com/bboylyg/RNP) into the framework of Backdoor-Bench for fair comparison and the specific experimental settings are:

- **Dataset and model architecture**: we compare our SAU with RNP on CIFAR-10 dataset with ResNet18 and PreAct-ResNet18.
- **Clean data size and poisoning ratio**: We adopt clean datasets: 1% and 5%, and poisoning ratios: 0.5%, and 10% to compare their effectiveness for large and small poisoning ratios.

Table 25: Results for defending against ALL to ALL attacks

| Defense | No Defense | | | | ANP [55] | | | | FP [31] | | | | NC [47] | | | |
|---|---|---|---|---|---|---|---|---|---|---|---|---|---|---|---|---|
| Attack | ACC | ASR | R-ACC | DER | ACC | ASR | R-ACC | DER | ACC | ASR | R-ACC | DER | ACC | ASR | R-ACC | DER |
| BadNets [15] | 91.89 | 74.42 | 18.66 | N/A | **92.33** | 2.56 | 90.12 | 85.93 | 83.91 | 1.72 | 83.71 | 82.36 | 91.88 | 74.41 | 18.67 | 50.0 |
| Blended [7] | 93.67 | 86.69 | 2.0 | N/A | 93.25 | 51.35 | 36.39 | 67.46 | 84.43 | 3.53 | 69.05 | 86.96 | **93.67** | 86.69 | 2.0 | 50.0 |
| Input-Aware | 91.92 | 83.69 | 7.23 | N/A | 91.04 | **0.85** | 88.34 | **90.98** | 85.69 | 2.01 | 81.09 | 87.72 | 91.20 | 81.49 | 8.75 | 50.74 |
| LF [58] | 93.84 | 89.96 | 2.44 | N/A | 93.10 | 2.58 | **90.31** | 93.32 | 84.54 | 1.77 | 80.8 | 89.44 | **93.84** | 89.96 | 2.44 | 50.0 |
| SIG [2] | 93.4 | 90.61 | 1.09 | N/A | 92.83 | 86.56 | 2.69 | 51.74 | 85.01 | 6.09 | 59.7 | 88.06 | **93.40** | 90.61 | 1.09 | 50.0 |
| SSBA [30] | 93.46 | 87.84 | 3.7 | N/A | 93.29 | 7.10 | 84.36 | 90.29 | 82.97 | 2.48 | 80.26 | 87.44 | 93.46 | 87.84 | 3.7 | 50.0 |
| WaNet [38] | 89.91 | 78.58 | 10.62 | N/A | 90.27 | **1.00** | 88.91 | **88.79** | 86.62 | 1.65 | 84.55 | 86.82 | 89.91 | 78.58 | 10.61 | 50.0 |
| Average | 92.58 | 84.54 | 6.53 | N/A | 92.30 | 21.71 | 68.73 | 81.22 | 84.74 | 2.75 | 77.02 | 86.97 | 92.48 | 84.23 | 6.75 | 50.11 |

| Defense | NAD [28] | | | | EP [64] | | | | i-BAU [59] | | | | SAU (**Ours**) | | | |
|---|---|---|---|---|---|---|---|---|---|---|---|---|---|---|---|---|
| Attack | ACC | ASR | R-ACC | DER | ACC | ASR | R-ACC | DER | ACC | ASR | R-ACC | DER | ACC | ASR | R-ACC | DER |
| BadNets [15] | 80.49 | 2.51 | 80.54 | 80.25 | 88.72 | 3.00 | 87.94 | 84.12 | 89.39 | 1.29 | 89.90 | 85.32 | 90.69 | **1.02** | 90.61 | **86.10** |
| Blended [7] | 76.85 | 5.63 | 58.85 | 82.12 | 91.69 | 82.62 | 3.66 | 51.05 | 90.46 | 8.83 | 72.20 | 87.32 | 91.47 | **2.17** | 82.08 | **91.16** |
| Input-Aware | 82.54 | 1.75 | 76.86 | 86.28 | 90.2 | 1.13 | 87.69 | 90.42 | 89.40 | 6.03 | 81.10 | 87.57 | **91.74** | 1.87 | 89.12 | 90.82 |
| LF [58] | 73.58 | 2.83 | 74.55 | 83.43 | 91.88 | 84.64 | 5.97 | 51.68 | 89.38 | 7.88 | 64.68 | 88.81 | 91.67 | **1.11** | 85.84 | 93.34 |
| SIG [2] | 79.99 | 5.78 | 55.48 | 85.71 | 90.91 | 86.10 | 2.44 | 51.01 | 90.74 | 1.57 | 86.31 | 93.19 | 91.13 | **1.48** | 85.14 | 93.43 |
| SSBA [30] | 74.12 | 3.58 | 69.61 | 82.46 | 90.33 | 41.83 | 46.59 | 71.44 | 88.56 | 6.49 | 78.49 | 88.22 | 91.35 | **1.84** | 85.53 | 91.94 |
| WaNet [38] | 85.26 | 1.99 | 82.59 | 85.97 | 86.66 | 74.74 | 11.13 | 50.3 | 91.71 | 1.63 | 90.98 | 88.48 | 91.73 | 1.43 | 89.34 | 88.58 |
| Average | 78.98 | 3.44 | 71.21 | 83.75 | 90.06 | 53.44 | 35.06 | 64.29 | 89.95 | 4.82 | 80.52 | 88.42 | 91.40 | **1.56** | 86.81 | 90.77 |

- **Hyper-parameters of RNP**: We notice that the hyper-parameters of RNP are sensitive to model architecture. After carefully inspecting the pruning process of RNP, we set the pruning threshold to 0.75 for ResNet18 and 0.05 for PreAct-ResNet18 which can achieve a good overall performance across various attacks.

The experimental results are shown in Table 26,27,28. According to the results and methodologies, we briefly summarize a few important differences between them:

- **Sensitivity to the size of the clean dataset.** SAU is effective across various sizes of clean data, while RNP achieves good performance for clean data size 0.5% 1%, as shown in Table 26 and discussed in Section 4 of [29]. Therefore, SAU can be applied to various sizes of clean datasets, especially when the defenders do not know the exact size of the training dataset.

- **Sensitivity to poisoning ratio.** SAU is effective across various poisoning ratios, while RNP faces challenges when defending against low poisoning ratio attacks, as shown in Table 28 and discussed in Section 5 of [29].

- **Sensitivity to model architecture.** SAU is effective and stable across various model architectures, while the parameters of RNP are sensitive to the model structure, as we observe that the pruning threshold for RNP varies significantly for various models.

- **Motivation.** SAU is theoretically motivated by the relationship between adversarial robustness and backdoor risk, while RNP is empirically driven by some experimental observations.

Table 26: Result on ResNet-18 with Poisoning Ratio 10%

| Attack | Clean Ratio | No Defense | No Defense | RNP [29] | RNP [29] | SAU (**Ours**) | SAU (**Ours**) |
|---|---|---|---|---|---|---|---|
| | | ACC | ASR | ACC | ASR | ACC | ASR |
| BadNets [15] | 1% | 91.46 | 95.31 | 88.94 | 2.14 | 83.62 | 2.04 |
| Input-Aware | 1% | 90.31 | 98.86 | 89.47 | 6.60 | 87.16 | 4.79 |
| SIG [2] | 1% | 84.64 | 99.69 | 80.88 | 0.03 | 84.08 | 0.37 |
| BadNets [15] | 5% | 91.46 | 95.31 | 10.65 | 0.00 | 89.25 | 1.33 |
| Input-Aware | 5% | 90.31 | 98.86 | 10.09 | 0.00 | 88.20 | 0.60 |
| SIG [2] | 5% | 84.64 | 99.69 | 10.00 | 0.00 | 89.07 | 3.99 |

Table 27: Result on PreAct-ResNet-18 with Poisoning Ratio 10%

| Attack | Clean Ratio | No Defense | No Defense | RNP [29] | RNP [29] | SAU (**Ours**) | SAU (**Ours**) |
|---|---|---|---|---|---|---|---|
| | | ACC | ASR | ACC | ASR | ACC | ASR |
| BadNets [15] | 1% | 91.32 | 95.03 | 66.86 | 0.00 | 85.12 | 2.52 |
| Input-Aware | 1% | 90.67 | 98.26 | 79.88 | 0.00 | 88.56 | 2.79 |
| SIG [2] | 1% | 84.48 | 98.27 | 74.47 | 0.04 | 84.64 | 0.64 |
| BadNets [15] | 5% | 91.32 | 95.03 | 10.00 | 0.00 | 89.31 | 1.53 |
| Input-Aware | 5% | 90.67 | 98.26 | 10.00 | 0.00 | 91.59 | 1.27 |
| SIG [2] | 5% | 84.48 | 98.27 | 11.56 | 0.00 | 88.56 | 1.67 |

Table 28: Result on ResNet-18 with Poisoning Ratio 0.5%

| Attack | No Defense | No Defense | RNP [29] | RNP [29] | SAU (**Ours**) | SAU (**Ours**) |
|---|---|---|---|---|---|---|
| | ACC | ASR | ACC | ASR | ACC | ASR |
| BadNets [15] | 93.73 | 67.06 | 91.46 | 11.54 | 82.63 | 2.46 |
| Input-Aware | 90.78 | 97.68 | 87.28 | 67.74 | 87.77 | 4.60 |
| SIG [2] | 93.68 | 85.02 | 79.91 | 14.76 | 87.71 | 0.14 |

