# OpenReview forum: "Shared Adversarial Unlearning: Backdoor Mitigation by Unlearning Shared Adversarial Examples"
_NeurIPS.cc/2023/Conference — NeurIPS 2023 poster_

### Official Review · Reviewer_1V93 · 2023-07-05

**Soundness:** 3 good
**Presentation:** 3 good
**Contribution:** 2 fair
**Rating:** 6
**Confidence:** 5

**Summary:**

This paper proposes a Shared Adversarial Unlearning (SAU) method for mitigating backdoor attacks. The motivation behind this method is the recognition that not all adversarial samples are effective for backdoor defense. Therefore, it is important to identify the adversarial samples that truly contribute to backdoor defense during adversarial training. The authors derive an upper bound on the backdoor risk from benign model and backdoor model predictions, and optimize the generation of backdoor samples based on this bound. Experimental results demonstrate the effectiveness of the method in purifying the backdoor model.

**Strengths:**

- Clear research motivation, well-structured writing, and provides ample theoretical support.
- Comprehensive comparison results with mainstream backdoor defense methods on multiple datasets, confirm its effectiveness.


**Weaknesses:**

- Adversarial training often involves complex min-max optimization problems, making it difficult to apply in practical training. However, existing defense work, such as ABL, achieves efficient backdoor removal by detecting suspicious backdoor samples and performing unlearning (gradient ascent) training. ABL's unlearning paradigm is evidently a simpler and more direct backdoor removal strategy. Therefore, it would be interesting to explore the combination of the adversarial sample generation strategy proposed in this paper with ABL's unlearning paradigm, simplifying the optimization process of the proposed adversarial training. It would be helpful if the authors could provide some theoretical or empirical results.

- From Table 3 in the experiments, it is apparent that the proposed SAU method fails to effectively defend against the WaNet attack with full-image triggers on the TinyImageNet dataset (attack success rate after defense is as high as 85.75%). In other words, when the image scale or trigger norm increases, the SAU method may face defense failures. The authors need to provide a reasonable explanation and corresponding countermeasures to demonstrate the generalization performance of the SAU method.

- If the attacker is aware of the defender's adversarial training strategy, can they generate corresponding backdoor trigger samples based on the adversarial training and launch attacks? In such a scenario, is the SAU defense effective?

- To my knowledge, adversarial defense is sensitive to the scale of clean data. In the experiments, the authors state that the defense method requires 5% clean samples. How much will the accuracy and defense performance of SAU decline when the scale of clean samples drops below 1%?

- There is a lack of comparison with more advanced defense benchmarks, such as the recently proposed RNP [1].

Overall, the authors of this paper highlight the differential impact of different types of adversarial samples on backdoor defense during adversarial training. While there are optimization uncertainties and application limitations, such as the difficulty in scaling to large models, associated with mitigating backdoor attacks through adversarial training, the authors' findings provide new insights into the relationship between adversarial samples and backdoor unlearning. Therefore, if the authors can address the aforementioned concerns and related questions, I would be happy to increase the score.

[1] Yige Li, Xixiang Lyu, Xingjun Ma, et al, Reconstructive Neuron Pruning for Backdoor Defense, ICML, 2023.


**Questions:**

Refer to the weaknesses mentioned above.

**Limitations:**

Refer to the weaknesses mentioned above.

---

> ### Author Rebuttal · Authors · 2023-08-09
>
> **Q1. For the combination of ABL**
>
> **A1:** Thanks. We would like to point out that ABL is an in-processing method that can access the whole training dataset for poisoned sample detection and unlearns the suspicious backdoor samples, while our method SAU is a post-processing method and can only use a small clean dataset. Therefore, ABL's unlearning paradigm cannot be directly applied to SAU. We also emphasize that simply replacing the unlearning step of SAU by gradient ascent still leads to a bi-level optimization problem and may not simplify the optimization process.
>
> **Q2. For the concern of generalization ability of the proposed methods and magnitude of trigger norm**
>
> **R2:** Thanks. We refer you to the **Common Response** where we discuss the generalization ability of SAU and test SAU with other perturbations such as the ensemble adversarial perturbation (Ensemble SAU). Note that Ensemble SAU serves as a strong variant of SAU and can achieve better defense performance, e.g., **ASR 0.15% for WaNet on Tiny ImageNet**.
>
> **Q3. For the concern of adaptive attacks**
>
> **R3:** When the attacker is aware of the defender's adversarial training strategy, the attacker can train the backdoored model using Adversarial Training (AT) on the poisoned dataset, such that the model can be resistant to adversarial attack, hindering the generation of shared adversarial examples.
>
> To evaluate the performance of SAU against such defense-aware backdoor attacks, we conduct experiments on CIFAR-10 dataset following the settings in Section 4. For AT, we use PGD-10 attack with $L_{\infty}$ norm 8/255. **From Table 1, we can find that SAU can achieve an average ACC of 78.3% and ASR of 2.44%, showing its resistance to such adaptive attacks.**
>
> **Table 1: Experiment Results for Adaptive Attack**
> ||No Defense|No Defense|SAU|SAU|
> :-:|:-:|:-:|:-:|:-:
> **Attack**|ACC|ASR|ACC|ASR
> BadNets|79.89|93.46|79.83|2.39
> Input-Aware|79.87|96.60|78.64|4.84
> SIG|73.50|99.50|76.41|0.10
> Average|77.75|96.52|78.30|2.44
>
> **Q4. For the concern about dataset size**
>
> **R4:** Thank. We would like to refer you to **Supplementary Material D.6** where we test SAU with various numbers of clean samples (from 5% to 0.02%) and show that SAU can be effective for various sizes of datasets.
>
> **Q5. For the suggestion of comparison with RNP [1].**
>
> **R5:** Since the earliest publication date of [1] is very close to the submission deadline of NeurIPS 2023, we didn't find it and compare with it. We would like to mention that **our submission is not expected to compare to work that appeared only a month or two before the deadline, according to the latest policy of NeurIPS 2023 (see NeurIPS-FAQ)**.
>
> However, we are still very glad to add the comparison with RNP [1] to further improve our submission. We adopted the official implementation of RNP into the framework of BackdoorBench for fair comparison and the specific experimental settings are:
>
> * **Dataset and model architecture**: we compare our SAU with RNP on CIFAR-10 dataset with ResNet18 and PreAct-ResNet18.
> * **Clean data size and poisoning ratio**: We adopt clean datasets: 1% and 5%, and poisoning ratios: 0.5%, and 10% to compare their effectiveness for large and small poisoning ratios.
> * **Hyper-parameters of RNP**: We notice that the hyper-parameters of RNP are sensitive to model architecture. After carefully inspecting the pruning process of RNP, we set the pruning threshold to 0.75 for ResNet18 and 0.05 for PreAct-ResNet18 which can achieve a good overall performance across various attacks.
>
> **Experimental results and analysis**
> The experimental results are shown in Table 2,3,4. According to the results and methodologies, we briefly summarize a few important differences between them:
>
> * **Sensitivity to the size of the clean dataset.** SAU is effective across various sizes of clean data, while RNP achieves good performance for clean data size 0.5%~1%, as shown in Table 2 and discussed in Section 4 of [1]. Therefore, SAU can be applied to various sizes of clean datasets, especially when the defenders do not know the exact size of the training dataset.
>
> * **Sensitivity to poisoning ratio.** SAU is effective across various poisoning ratios, while RNP faces challenges when defending against low poisoning ratio attacks, as shown in Table 4 and discussed in Section 5 of [1].
>
> * **Sensitivity to model architecture.** SAU is effective and stable across various model architectures, while the parameters of RNP are sensitive to the model structure, as we observe that the pruning threshold for RNP varies significantly for various models.
>
> * **Motivation.** SAU is theoretically motivated by the relationship between adversarial robustness and backdoor risk, while RNP is empirically driven by some experimental observations.
>
> **Table 2: Result on ResNet-18 with Poisoning Ratio 10%**
> |||No Defense|No Defense|RNP|RNP|SAU|SAU|
> :-:|:-:|:-:|:-:|:-:|:-:|:-:|:-:
> **Attack**|**Clean Ratio**|ACC|ASR|ACC|ASR|ACC|ASR
> BadNets|1%|91.46|95.31|88.94|2.14|83.62|2.04
> Input-Aware|1%|90.31|98.86|89.47|6.60|87.16|4.79
> SIG|1%|84.64|99.69|80.88|0.03|84.08|0.37
> BadNets|5%|91.46|95.31|10.65|0.00|89.25|1.33
> Input-Aware|5%|90.31|98.86|10.09|0.00|88.20|0.60
> SIG|5%|84.64|99.69|10.00|0.00|89.07|3.99
>
> **Table 3: Result on PreAct-ResNet-18 with Poisoning Ratio 10%**
> |||No Defense|No Defense|RNP|RNP|SAU|SAU|
> :-:|:-:|:-:|:-:|:-:|:-:|:-:|:-:
> **Attack**|**Clean Ratio**|ACC|ASR|ACC|ASR|ACC|ASR
> BadNets|1%|91.32|95.03|66.86|0.00|85.12|2.52
> Input-Aware|1%|90.67|98.26|79.88|0.00|88.56|2.79
> SIG|1%|84.48|98.27|74.47|0.04|84.64|0.64
> BadNets|5%|91.32|95.03|10.00|0.00|89.31|1.53
> Input-Aware|5%|90.67|98.26|10.00|0.00|91.59|1.27
> SIG|5%|84.48|98.27|11.56|0.00|88.56|1.67
>
> **Table 4: Result on ResNet-18 with Poisoning Ratio 0.5%**
> ||No Defense|No Defense|RNP|RNP|SAU|SAU|
> :-:|:-:|:-:|:-:|:-:|:-:|:-:
> **Attack**|ACC|ASR|ACC|ASR|ACC|ASR
> BadNets|93.73|67.06|91.46|11.54|82.63|2.46
> Input-Aware|90.78|97.68|87.28|67.74|87.77|4.60
> SIG|93.68|85.02|79.91|14.76|87.71|0.14

---

> ### Author Response · Authors · 2023-08-16
>
> Dear Reviewer  1V93:
>
> We would like to express our sincere gratitude for your valuable insights and suggestions on our work.
>
> We have tried our best to address the concerns and queries you raised during the rebuttal process. However, we would greatly appreciate knowing whether our response has effectively resolved your doubts. Your feedback will be instrumental in improving the quality of our work. **As the end of the discussion period is approaching, we eagerly await your reply before the end.**
>
> Sincerely,
>
> Authors

---

> > ### Comment · Reviewer_1V93 · 2023-08-17
> > **Thanks for the response.**
> >
> > I greatly appreciate the author's efforts in the rebuttal stage, providing more detailed empirical results that effectively support the effectiveness and generalizability of the proposed SAU method. The authors also compared it with the currently most advanced defense methods, such as RNP. Despite some limitations in terms of innovation, the SAU method shows the ability to effectively defend against a wide range of attack types. Its good generalization across different model architectures and data scales further demonstrates its superiority. Considering these advantages, I have raised my score to 6.

---

> > > ### Author Response · Authors · 2023-08-17
> > > **Thanks for your feedback**
> > >
> > > Dear Reviewer 1V93,
> > >
> > > We sincerely appreciate your thoughtful response and the time you've dedicated to reviewing our paper.
> > >
> > > **We are strongly encouraged by your recognition of our efforts, including our motivation, writing, theoretical analysis, comprehensive experiments, and novel insights.** We will incorporate your suggestions and insights into the revised manuscript. Thank you once again for your thorough review and your positive evaluation. Your support and input are greatly appreciated.
> > >
> > > Sincerely,
> > >
> > > Authors

---

### Official Review · Reviewer_mKLw · 2023-07-06

**Soundness:** 3 good
**Presentation:** 3 good
**Contribution:** 3 good
**Rating:** 4
**Confidence:** 4

**Summary:**

This paper proposes a method to defend against backdoor attacks in deep neural networks through adversarial training techniques. By exploring the connection between adversarial examples and poisoned samples, the authors propose an upper bound for backdoor risk and a bi-level formulation for mitigating backdoor attacks in poisoned models. The proposed approach can identify shared adversarial examples and unlearns them to break the connection between the poisoned sample and the target label.

**Strengths:**

- This paper explores the connection between adversarial examples and backdoor attacks, which is interesting.

- Extensive experiments are conducted to evaluate the performance of the proposed method.

- The paper is well written and easy to read.

**Weaknesses:**

- Some important related works are missing. There are some existing works that study provable/certified robustness against backdoor attacks or data poisoning [1,2,3]. However, the authors do not mention those works in the paper. It would be interesting if the authors could compare their method with those works.

- This paper assumes that the trigger magnitude is bounded by a norm, which may degrade the practicality of the proposed method. In practice, it is entirely possible for the attacker to adopt triggers with very large magnitudes, and the proposed method may not work in this case.

- The proposed method relies on a clean dataset to achieve the defense goal, and the quality of the dataset may affect the performance of the proposed method. However, it is not clear how to obtain a clean dataset with high quality in practice.

     [1] RAB: Provable Robustness Against Backdoor Attacks. IEEE S&P 2022.

     [2] Certified Robustness of Nearest Neighbors against Data Poisoning and Backdoor Attacks. AAAI 2022.

     [3] BagFlip: A Certified Defense against Data Poisoning. NeurIPS 2022.

**Questions:**

- Compare the proposed method with existing works that study provable/certified robustness against backdoor attacks or data poisoning.

- How to achieve good performance when the trigger magnitude is larger than the assumed norm?

- Does the quality of the clean dataset affect the performance of the proposed method? How to obtain the dataset with high quality in practice?


**Limitations:**

The authors have discussed the limitations.

---

> ### Author Rebuttal · Authors · 2023-08-09
>
> **Q1.Suggestion of comparison to provable/certified backdoor defense methods [1,2,3].**
>
> **R1:** We acknowledge the importance of certified robustness against backdoor attacks. In backdoor defense, methods can be categorized into two classes: **empirical methods which aim to develop effective and efficient methods for backdoor defense, and certified methods which aim to build a provable framework for backdoor robustness**. As our method is an empirical method, we would like to point out that our method is different from the existing works on certified robustness in several aspects, which makes them incomparable or impractical to apply to our setting. Specifically:
>
> 1. **Efficiency and scalability.** Certified methods usually have very high computation costs and are infeasible to apply to large-scale deep neural networks, while our method is efficient and scalable. For example, the recommended papers [1] and [3] rely on voting over **1000 smoothed models** for prediction and are only feasible for some simple neural networks and simple classification problems. To apply these methods to our setting, we would need to train 1000 PreAct-ResNet models on CIFAR-10 from scratch, which would take over **600 hours (25 days) and 620GB of space for a single attack on a server with one RTX 3090**. That's why we cannot compare our method with [1,2].
>
> 2. **Applicability to model architecture.** Many certified methods are restricted to some simple or classical models, while our method is general and applicable to any deep neural network architecture. For example, the recommended paper [2] is designed for only k-Nearest Neighbors (kNN) and radius Nearest Neighbors (rNN) models with some feature extractor and cannot be applied to modern deep neural networks.
>
> Therefore, we believe that our method is more practical and effective for defending against backdoor attacks in deep neural networks than the existing works on certified robustness. We will add more discussion on this point in the revised version of the paper.
>
> **Q2. For the concern about trigger magnitude**
>
> **R2:** Thanks. We would like to emphasize that our theoretical analysis and the proposed method are independent of the choice of perturbation set $S$. Therefore, various perturbation types/sets (not only $L_p$ ball) can be adapted to SAU. We refer you to the **Common Responses** for a more comprehensive analysis of different types of adversarial perturbations and the generalization ability for SAU, including defending against BadNets with various strengths.
>
> We remark that the **Ensemble SAU** in the **Common Responses** which generates Shared Adversarial Examples by a set of perturbation types, achieves the lowest ASR in all experiments, demonstrating its effectiveness for backdoor mitigation against various attacks.
>
> We hope the above discussion and the analysis in the **Common Response** could address your concern.
>
> **Q3. For the concern of how to collect clean data and the influence of data quality**
>
> **R3:** We would like to emphasize that accessing a small clean dataset is a common and reasonable assumption for most post-processing methods for backdoor defense, such as [5,6,7,8,9]. There are some possible ways to obtain a clean dataset in practice, such as buying from a trustworthy data supplier, using modern generative models, collecting from the internet, or applying some data cleansing techniques [10].
>
> We acknowledge that the influence of data quality on post-processing methods is interesting. To investigate the influence of data quality for post-processing backdoor defense methods, we conduct experiments on synthetic datasets and compare SAU with two SOTA methods, i.e., **ANP** and **NAD**. Specifically, we test the three methods with a synthetic dataset CIFAR-5m [11] which provides generated CIFAR-10-like images whose distribution is close but not identical to CIFAR-10, and can be regarded as OOD data. We build mixed datasets by randomly picking samples from CIFAR-10 and CIFAR-5m with a **mixed ratio**, and evaluate the three methods on the mixed datasets. A larger mixed ratio indicates more synthetic data in the mixed dataset. The experiment results are summarized in the following tables.
>
> **Table 1: Experiment for Backdoor Defense on Mixed Datasets**
> ||ANP|ANP|NAD|NAD|SAU|SAU|
> :-:|:-:|:-:|:-:|:-:|:-:|:-:
> **Mixed Ratio**|ACC|ASR|ACC|ASR|ACC|ASR
> 0.0|90.88|4.88|89.87|2.14|89.31|1.53
> 0.2|90.68|2.10|87.96|1.88|89.83|1.63
> 0.4|90.21|1.11|87.69|1.88|89.36|1.60
> 0.6|90.31|1.51|86.69|2.19|88.44|1.66
> 0.8|89.37|0.56|85.90|1.86|88.13|1.79
> 1.0|88.74|0.40|84.96|2.48|88.32|1.01
>
> We can find that **all methods can work on the mixed dataset but as the mixed ratio increase, the accuracy (ACC) may decrease due to the distribution difference between the two datasets**. It's notable that the model distillation-based method NAD is influenced by the quality of the dataset most, with a reduction of 4.91% in ACC, while **SAU is quite robust to the data quality**. Another interesting phenomenon is that the ASR of ANP decreases when more synthetic data is used, which shows that the distribution of data may influence the performance of backdoor defense. All those observations motivate us to conduct research on the relationship between data quality and backdoor defense in the future.
>
> Reference
>
> [5] Adversarial neuron pruning purifies backdoored deep models. NeurIPS 2021.
>
> [6] One-shot neural backdoor erasing via adversarial weight masking. NeurIPS 2022.
>
> [7] Adversarial unlearning of backdoors via implicit hypergradient. ICLR 2021.
>
> [8] Neural cleanse: Identifying and mitigating backdoor attacks in neural networks. SP 2019
>
> [9] Reconstructive Neuron Pruning for Backdoor Defense, ICML 2023.
>
> [10] Effective Backdoor Defense by Exploiting Sensitivity of Poisoned Samples.
> NeurIPS 2022.
>
> [11] The Deep Bootstrap Framework: Good Online Learners are Good Offline
> Generalizers. ICLR 2020.

---

> ### Author Response · Authors · 2023-08-16
>
> Dear Reviewer mKLw:
>
> We would like to express our sincere gratitude for your valuable insights and suggestions on our work.
>
> We have tried our best to address the concerns and queries you raised during the rebuttal process. However, we would greatly appreciate knowing whether our response has effectively resolved your doubts. Your feedback will be instrumental in improving the quality of our work. **As the end of the discussion period is approaching, we eagerly await your reply before the end.**
>
> Sincerely,
>
> Authors

---

### Official Review · Reviewer_Do9c · 2023-07-07

**Soundness:** 2 fair
**Presentation:** 3 good
**Contribution:** 2 fair
**Rating:** 3
**Confidence:** 4

**Summary:**

This paper analysed the relationship between adversarial examples and poisoned examples.
Then, this paper proposed a fine-tuning strategy to purify the poisoned model.

**Strengths:**

1 This paper is easy to follow.

2 This paper provides some experiments that support the proposed method.

3 This paper has some experimental analyses that can lead to the proposed method.

**Weaknesses:**

1 It seems this paper used many  ill-established notions such as backdoor risk, shared adversarial risk, vanilla adversarial risk, etc..  To the best of my knowledge, I haven't heard to backdoor risk in any machine learning/AI books.

2 Adversarial training has the issue of high computational overhead, which could limit the practicality of the proposed method in real-world applications.

3 The key reference is missing. This paper also need to compared with the paper [1].
Previous work [1] has proposed to tackle with backdoor attacks with multiple adversarial perturbations (L_p adversarial attack and spatial adversarial attack). I encourage the authors to discuss the differences between both works and explore more types of adversarial attack such as spatial adversarial attacks [2] or perceptual adversarial attacks [3].  Besides, some popular backdoor attacks do not satisfy Assumption 1 such as BadNets which has a large L_p norm.

[1] On the Effectiveness of Adversarial Training against Backdoor Attacks. TNNLS 2023.

[2] Spatially Transformed Adversarial Examples. ICLR 2018.

[3] Perceptual Adversarial Robustness: Defense Against Unseen Threat Models. ICLR 2021.

**Questions:**

Can the proposed method mitigate the backdoor issues of the contrastive learning [4], which is a more practical setting.

[4] Poisoning and backdooring contrastive learning, ICLR2022

**Limitations:**

After the part of conclusion, the authors discussed the limitation of this work.

---

> ### Author Rebuttal · Authors · 2023-08-09
>
> **Q1. For the concern about terminology**
>
> **R1.** We appreciate your interest in our work and the terminology we use. We would like to clarify the following points:
>
> * As backdoor learning is an emerging area, we follow the typical paradigm of scientific research like [4,5] to first formulate the question by risk and then replace the risk with surrogate loss. In this way, we first defined the backdoor risk to investigate the relationship between adversarial example and poisoned sample. Therefore, **backdoor risk is not covered by current AI/ML books since it's proposed by our paper and one of contributions**.
>
> * The detailed explanation for some terminology is provided below:
>      1. **Backdoor Risk** is defined as the empirical probability that a backdoored model predicts poisoned sample with trigger to target label. Since backdoored model predicts the poisoned samples to the target class, a lower backdoor risk indicates the model is less likely to be affected by triggers.
>
>      2. **Adversarial Risk** is widely used in the adversarial learning community to measure the performance of adversarial attack and defense[4,5]. It is defined as the empirical probability that a model misclassifies an adversarially perturbed sample. Furthermore, the term 'vanilla' means standard version of something. Therefore, vanilla adversarial risk is simply adversarial risk and we will clarify it in the revised version.
>
>      3. **Shared Adversarial Risk** is a novel concept proposed in this work and one of our contributions, which provides a tighter upper bound for backdoor risk and suggests a new perspective for backdoor mitigation. It is defined as the empirical probability that two models predict an adversarial example to the same wrong label.
>
> The formal mathematical definition of Backdoor Risk, Adversarial Risk, and Shared Adversarial Risk are presented in Section 3.
>
> **Q2. For the concern about computation cost**
>
> **R2:** Thanks. We would like to refer you to **Q5 of Reviewer QS3I** for a comprehensive analysis of the computation cost, which shows that **SAU is an efficient and effective defense method for backdoor mitigation**.
>
> **Q3. Suggestion for comparison to [1].**
>
> **R3:** Thank you for your constructive suggestion. Since the earliest publication date of [1] (early access on 14 June 2023) is after the submission deadline of NeurIPS 2023, we didn't find it and compare with it. We would like to mention that **our submission is not expected to compare to work that appeared only a month or two before the deadline, according to the latest policy of NeurIPS 2023 (see NeurIPS-FAQ).**
>
> However, we are still glad to highlight the following differences between our method and Composite Adversarial Training (CAT) proposed in [1]:
>
> 1. SAU is theoretically motivated by the relationship between adversarial robustness and backdoor risk, while CAT is empirically driven by some experimental observations. Therefore, our method has a theoretical guarantee for the performance against various backdoor attacks, while CAT does not.
> 2. SAU fine-tunes a backdoored model with a small clean dataset, while CAT trains a backdoor-free model from scratch on a poisoned dataset. Therefore, SAU is more efficient than CAT.
>
> We have also compared the performance of SAU and CAT on CIFAR-10 dataset following the settings in Section 4. The hyper-parameters for CAT are the same as [1]. The results are shown in the following tables:
>
> **Table 1: Experiments Results with poisoning ratio 1%**
>
> |Method|No Defense|No Defense|No Defense|CAT|CAT|CAT|SAU (Ours)|SAU (Ours)|SAU (Ours)|
> :-:|:-:|:-:|:-:|:-:|:-:|:-:|:-:|:-:|:-:
> Attack|Acc|ASR|R-ACC|Acc|ASR|R-ACC|Acc|ASR|R-ACC
> BadNets|93.14|74.73|24.24|75.52|2.50|74.73|**91.25**|**0.94**|**91.16**
> Input-Aware|91.74|79.18|19.89|75.24|72.48|23.97|**92.20**|**3.63**|**84.06**
> SSBA|93.43|73.44|24.89|74.94|3.07|71.40|**91.34**|**0.79**|**88.46**
> WaNet|90.65|12.63|79.94|75.21|2.68|74.46|**91.84**|**1.23**|**89.89**
> Average|92.24|60.00|37.24|75.23|20.18|61.14|**91.66**|**1.65**|**88.39**
>
> **Table 2: Experiments Results with poisoning ratio 10%**
>
> |Method|No Defense|No Defense|No Defense|CAT|CAT|CAT|SAU (Ours)|SAU (Ours)|SAU (Ours)|
> :-:|:-:|:-:|:-:|:-:|:-:|:-:|:-:|:-:|:-:
> Attack|Acc|ASR|R-ACC|Acc|ASR|R-ACC|Acc|ASR|R-ACC
> BadNets|91.32|95.03|4.67|74.42|92.49|6.21|**89.31**|**1.53**|**88.81**
> Input-Aware|90.67|98.26|1.66|74.21|96.81|2.88|**91.59**|**1.27**|**88.54**
> SSBA|92.88|97.8|1.99|74.29|28.29|57.49|**90.84**|**1.79**|**85.83**
> WaNet|91.25|89.73|9.76|74.62|4.87|73.07|**90.41**|**2.51**|**79.69**
> Average|91.53|95.21|4.52|74.39|55.61|34.91|**92.54**|**1.78**|**85.72**
>
> The experimental results show that our method is superior to CAT, as evidenced by the following aspects:
>
> 1. SAU achieves significantly higher accuracy (ACC) and lower attack success rate (ASR) than CAT on both poisoning ratios.
> 2. SAU is more robust to different poisoning ratios than CAT. In contrast, CAT fails on a high poisoning ratio (10%) (also observed in Table XI of [1]).
>
> **Q4. For exploring more types of adversarial perturbations and the concern about Assumption 1.**
>
> **R4.** Thanks. We would like to refer you to the **Common Response** for more comprehensive analysis and experiments for SAU with different adversarial perturbations such as the suggested **Spatial Adversarial Perturbation (Spatial SAU)** and **Ensemble Adversarial Perturbation (Ensemble SAU)**, and the generalization ability of SAU.
>
> **Q5. Suggestion for contrastive learning**
>
> **A5:** We appreciate your insightful suggestion for applying SAU to contrastive learning. However, the current theoretical analysis and method are designed for classification problems and cannot be directly applied to contrastive learning where the label of samples are not accessible.
>
> References
>
> [4] Improving adversarial robustness requires revisiting misclassified examples, ICLR 2019
>
> [5] Theoretically principled trade-off between robustness and accuracy, ICML 2019

---

> > ### Comment · Reviewer_Do9c · 2023-08-15
> > **Post-rebuttal comments.**
> >
> > Thanks for authors providing feedbacks.
> >
> > + I am satisfied with experimental comparisons with the existing literature.
> > Authors are encouraged to make comprehensive comparisons in revision and compare with existing studies in terms of different settings.
> >
> > - I am not satisfied with the newly defined notations of backdoor risk and shared adversarial risk.
> >
> > I would like to keep my score unchanged.

---

> > > ### Author Response · Authors · 2023-08-15
> > > **Comment on the Dissatisfaction of Notations**
> > >
> > > Reviewer Do9c,
> > >
> > > Thanks for your further feedback. We appreciate your interest in our paper and your constructive suggestions. We are eager to improve our paper and address your concerns. Hence, we would like to request more details about your criticism by asking the following questions:
> > >
> > > 1. **Would you please specify that what aspects of the notations are unsatisfying? Are they unclear, inconsistent, or inaccurate?** That will be important for us to further improve these notations.
> > > 2. In terms of rating 3, its official criteria is "Reject: For instance, a paper with technical flaws, weak evaluation, inadequate reproducibility and incompletely addressed ethical considerations". **Would you please specify which aspect our submission satisfies?**
> > >
> > > Your help is greatly appreciated. Thanks again for your valuable time and attention.
> > >
> > > Sincerely,
> > > Authors

---

### Official Review · Reviewer_qs3i · 2023-07-08

**Soundness:** 3 good
**Presentation:** 2 fair
**Contribution:** 2 fair
**Rating:** 5
**Confidence:** 3

**Summary:**

summary: The paper proposes a mitigation approach for backdoor attacks in ML (attacks where a model predicts target classes for poisoned samples when triggered by the adversary). The authors analyze the relationship between backdoor risk and adversarial risk [adversarial examples and poisoned samples] to create an upper bound for the backdoor risk, identify --Shared Adversarial Examples-- between the poisoned model and the fine-tuned model (which can successfully activate the backdoor) and then formulate an optimization or a purification problem for mitigating backdoor attacks (Shared Adversarial Unlearning) to break the connection between poisoned samples and the target label, such that they are either classified correctly by the fine-tuned model or differently by the two models. The authors further thoroughly analyze their approach, comparing it with six other SOTA defense methods on seven types of backdoor attacks with different model structures and datasets.



**Strengths:**

The paper targets an important problem. The approach is novel and the thorough analysis of several key metrics indicates the efficacy of the approach.


**Weaknesses:**


* Many  important part are in the supplemental materials, including some of the approaches such as the SAU implementation using SGD to unlearn the SAEs, the all to all case, and multi-trigger in threat model (section 3.1) is unclear.



**Questions:**

*  Type I adv. examples where hθbd and hθ can be misled to the same class represents an upper bound for back door examples. How do you account for false positives? Or is it assumed to be always 100%?

* How does ASR vary in comparison to other unlearning techniques for backdoor mitigation such as I-BAU.

* What are the tradeoffs associated with this approach in terms of implementation - specifically around the cost of identifying the SAEs and SAU?



**Limitations:**

* Writing flow can be improved. Adding a toy example would definitely improve the overall understanding of the paper.

---

> ### Author Rebuttal · Authors · 2023-08-09
>
> **Q1. Suggestion for paper layout**
>
> **R1:** Thanks for your constructive suggestion. We will update the layout in the revised manuscript by moving the suggested contents and other important contents from the supplementary material to the main manuscript, to make it more self-contained and legible. Thanks again for your helpful suggestion.
>
>
> **Q2. Multi-trigger in threat model (section 3.1) is unclear.**
>
> **R2:** We appreciate your interest in the threat model of multi-trigger cases. We address the **multi-trigger and multi-target setting** of the threat model in **Supplemental Materials A.6**. There, we formally define the Multi-target/Multi-trigger threat model and extend our theoretical analysis (Proposition 3) to this case.
>
> We have also evaluated our method for the multi-trigger/multi-target cases. Specifically, we use two different attacks (denoted by Attack-1 and Attack-2) with different target labels. Following Section 4, we test our method (SAU) against a strong baseline ANP on the CIFAR-10 dataset with a poisoning ratio of 10% and backbone PreAct-ResNet18. The experiment results are summarized in the following table, where we use ASR-1 and ASR-2 to represent the Attack Success Rate for Attack-1 and Attack-2, respectively. These results show that SAU outperforms the baseline in most cases, which demonstrates the effectiveness and robustness of our method in this challenging scenario.
>
> **Table 1: Experiment Results for Multi-trigger and Multi-target Case**
>
> |||NO Defense|NO Defense|NO Defense|ANP|ANP|ANP|SAU|SAU|SAU|
> :-:|:-:|:-:|:-:|:-:|:-:|:-:|:-:|:-:|:-:|:-:
> Attack-1|Attack-2|ACC|ASR-1|ASR-2|ACC|ASR-1|ASR-2|ACC|ASR-1|ASR-2
> Blended|BadNet|90.22|99.27|95.03|85.55|37.07|**0.01**|**88.93**|**2.21**|0.54
> InputAware|BadNet|89.71|78.62|94.77|82.21|4.48|**0.01**|**88.88**|**4.14**|0.70
> LF|BadNet|90.16|98.08|95.15|84.21|79.11|2.30|**88.45**|**2.93**|**0.34**
> SIG|BadNet|82.12|98.48|95.03|76.43|26.92|**0.14**|**85.82**|**0.60**|0.67
> SSBA|BadNet|89.69|95.13|95.11|83.25|20.03|**0.14**|**87.60**|**1.38**|0.52
> WaNet|BadNet|89.44|90.21|95.40|82.21|5.53|**0.01**|**88.02**|**3.12**|0.67
> Average||88.56|93.30|95.08|82.31|28.86|**0.44**|**87.95**|**2.40**|0.57
>
> **Q3. Type I adv. examples where $h_{\theta}^{bd}$ and $h_{\theta}$ can be misled to the same class represents an upper bound for back door examples. How do you account for false positives? Or is it assumed to be always 100%?**
>
> **R3:** Thank you for your interest in shared (Type I) adversarial examples (SAEs) and the proposed upper bound for backdoor examples. We would like to clarify that **not all shared adversarial examples are poisoned samples, and there is a possibility of false positives.** Note that SAEs is a subset of standard adversarial examples. Therefore, **the false positive rate of SAEs is significantly lower than that of standard adversarial examples**, which leads to remarkable improvement in reducing ASR and maintaining clean accuracy. However, there may still be some non-poisoned samples among the shared adversarial examples, which motivates further research on more precise identification of backdoor samples among adversarial examples in the future.
>
> **Q4. How does ASR vary in comparison to other unlearning techniques for backdoor mitigation such as i-BAU.**
>
> **R4:** We appreciate your interest in the comparison between SAU and i-BAU. Here we summarize comparison from Table 2 and Table 3 in our paper:
> * On CIFAR-10, SAU outperforms i-BAU in 5 of 7 attacks with an average ASR reduction of 12.81%.
> * On Tiny ImageNet, SAU outperforms i-BAU in all 6 attacks, with an average ASR reduction of 69.56%.
>
> These results demonstrate that our method is superior to i-BAU in mitigating backdoor effect.
>
> **Q5. What are the tradeoffs associated with this approach in terms of implementation, specifically around the cost of identifying the SAEs and SAU?**
>
> **R5:** Thank you for your concern about the computation cost of SAU. We would like to emphasize that **SAU is an efficient and effective defense method** which can mitigate backdoor attacks with acceptable overhead for the following reasons:
> 1. SAU is a post-processing method that fine-tunes the backdoor model on a clean dataset with a small size. For instance, we can use only 500 samples from CIFAR-10 to fine-tune the model by SAU (see **Supplemental Material D.6** for more details), which takes 1.52 seconds/epoch to fine-tune PreAct-ResNet18 with RTX 4090. Therefore, it can be executed efficiently without requiring a large amount of data or computation resources.
> 2. SAU only takes a few epochs to take effect, which further reduces the computational cost of executing it in practice. As shown in **Supplemental Material D.4**, SAU can achieve a low Attack Success Rate and a high Accuracy in a few epochs.
>
> To further demonstrate the efficiency of SAU, we compare its average runtime with other state-of-the-art defense methods against attacks. We use the same experimental setting as in Section 4, where the poisoning ratio is 10% and the backbone model is PreAct-ResNet18. All experiments are conducted on a server with GPU RTX 4090 and CPU AMD EPYC 7543 32-Core Processor. Table 2 shows the average runtime for each defense method to take effect on CIFAR-10 and Tiny ImageNet (ASR<5% and ACC>85% on CIFAR-10, or ASR<5%, ACC>45% on Tiny ImageNet). Note that if a defend method cannot reach the criteria for "Take effect", the maximum runtime is reported. As we can see, **SAU is faster than most of the existing methods, except for NAD and EP**.
>
> **Table 2: Computation Time**
> |Dataset|ANP|FP|NC|NAD|EP|i-BAU|SAU|
> :-:|:-:|:-:|:-:|:-:|:-:|:-:|:-:
> CIFAR-10|289.86s|266.57s|825.86s|75.14s|65.71s|47.00s|43.86s
> Tiny ImageNet|1086.50s|318.30s|27359.70s|227.01s|141.67s|621.33s|262.60s
>
> **Q6. Suggestion for a Toy Example.**
>
> **R6:** We appreciate your constructive suggestion for a toy example and we will add a toy example in the revised version for demonstration. Thank you again for your helpful suggestion.

---

> ### Author Response · Authors · 2023-08-16
>
> Dear Reviewer qs3i:
>
> We would like to express our sincere gratitude for your valuable insights and suggestions on our work.
>
> We have tried our best to address the concerns and queries you raised during the rebuttal process. However, we would greatly appreciate knowing whether our response has effectively resolved your doubts. Your feedback will be instrumental in improving the quality of our work. **As the end of the discussion period is approaching, we eagerly await your reply before the end.**
>
> Sincerely,
>
> Authors

---

### Author Rebuttal · Authors · 2023-08-10

# Common Response

**Q1. Concerns about the perturbation set $\mathcal{S}$ in Assumption 1 that restricts the trigger magnitude, including its practicality, and its generalization to backdoor attack with excessive-magnitude trigger.**

**A1:** Thanks for the insightful comments. The concerns are raised from Assumption 1, which says that $g(x;\Delta) - x \in \mathcal{S}$ for a sample $x$ and its corresponding poisoned sample $g(x,\Delta)$, which ensures that there exists $\epsilon \in S$ such that $x+ \epsilon = g(x;\Delta)$. We would like to clarify the following points:
1. **Type of adversarial perturbation (AP) and perturbation set.** The theoretical analysis and the proposed method SAU is independent of the AP type and AP set **(not only $L_p$ ball)**. For sample space $\mathcal{X}$, define the set $G=\\{g(x,\Delta)-x|x\in \mathcal{X}\\}$ and the general perturbation set $S=\\{T(x,\epsilon)-x|x\in \mathcal{X}\\}$ for adversarial attack $T$ with learnable parameter $\epsilon\in \Omega$ such that an adversarial example can be generated by $\tilde x_\epsilon=T(x,\epsilon)$. Then, **the theoretical analysis and proposed method can be applied if $G\subseteq S$, where the AP can be any type, such as additive AP, Spatially Transformed AP [1], or Perceptual AP [2], etc.**
2. **Effect of perturbation set $S$.** The key of SAU is to consider Shared Adversarial Example (SAE) as surrogates for poisoned samples. Given models, **the set of SAE is determined by S: larger S leads to larger set of SAE, and vice versa** Therefore, the perturbation set $S$ plays an important role in the effectiveness of SAU. Specifically, adopting a larger $S$ can cover unknown trigger but also increases the chance of picking SAE this is irrelevant to poisoned samples, and may weaken the defense performance. Meanwhile, some poisoned samples may be beyond the scope of SAE if a smaller $S$ is adopted.
3. **Generalization to excessive-magnitude trigger.** In practice, we find that SAU is effective for various attacks, even if Assumption 1 does not hold. For example, SAU with $L_\infty$ norm bound 0.2 can still effectively mitigate SSBA attack whose average $L_\infty$ trigger norm is much larger than 0.2, revealing the generalization of SAU to attacks with excessive-magnitude trigger.

	**Table 1: SAU for SSBA Attack**
	|Dataset|$L_1$|$L_2$|$L_\infty$|ACC|ASR|R-ACC|
	:-:|:-:|:-:|:-:|:-:|:-:|:-:
	CIFAR-10|108.33|3.29|0.33|90.99|0.58|87.04
	Tiny|581.65|8.26|0.43|51.85|0.11|36.36

	To investigate the SAU's generalization ability, we conduct experiments on BadNets with various strengths. Specifically, for BadNets with 3x3 patched trigger, we alter the trigger's pixel value from 0.2 to 1.0. Table 2 shows the results for defending various BadNets attacks by SAU following the setting in Section 4, from which we can find that **SAU can consistently mitigate the backdoor even when the trigger is beyond the perturbation set**. One reasonable explanation is that although the poisoned samples are not included in SAEs, there exists SAE that has similar effect as poisoned samples as shown in Fig. 1 in the below **Supplementary PDF**, resulting in strong generalization ability of SAU.

	**Table 2: Defend against BadNets Attack with Various Pixel Value**
	|||No Defense|No Defense|ANP|ANP|SAU|SAU|
	:-:|:-:|:-:|:-:|:-:|:-:|:-:|:-:
	Pixel|L_inf|ACC|ASR|ACC|ASR|ACC|ASR
	0.2|0.42|91.69|95.60|91.30|2.64|90.01|1.38
	0.4|0.31|91.61|97.79|91.44|1.00|88.43|1.59
	0.6|0.35|91.94|96.43|91.72|7.81|90.49|1.30
	0.8|0.48|91.72|93.20|91.06|2.37|89.23|1.36
	1.0|0.67|91.32|95.03|90.88|4.88|89.31|1.53

4. **Effect of perturbation type:** As pointed by Reviewer 1V93, SAU fails to defend WaNet attack on Tiny ImageNet, which generates poisoned samples by spatial transform and has much larger $L_\infty$ trigger norm. Following the constructive suggestion from Reviewer Do9c, we extend SAU to incorporate two other perturbation types and denote the resulting methods by **Spatial SAU** and **Ensemble SAU**, respectively. Specifically, for spatial SAU, we employ spatial adversarial attack to generate spatial SAEs by spatial transformation. For Ensemble SAU, the perturbation set $S$ is composed of a collection of subsets, including $L_1$ ball (bound 500), $L_2$ ball (bound 10), $L_\infty$ ball (bound 0.2), and the spatial AP set. Then, for each batch, Ensemble SAU randomly chooses one subset of perturbation to generate the SAEs.

	In **Table 3**, we compare SAU, **Spatial SAU** and **Ensemble SAU**, following the setting in Section 4. From the table, we find that Spatial SAU works well on CIFAR-10 dataset but fails to defend BadNets and SSBA attacks on Tiny ImageNet. For WaNet, spatial SAU can effectively defend it on both datasets. We remark that the **Ensemble SAU achieves the lowest ASR in all experiments**. However, as Ensemble SAU has larger perturbation set, its accuracy is lower than SAU by 1~2%.

	**Table 3: Comparison between SAU, Spatial SAU and Ensemble SAU**
	|||No Defense|No Defense|SAU|SAU|Spatial SAU|Spatial SAU|Ensemble SAU|Ensemble SAU|
	:-:|:-:|:-:|:-:|:-:|:-:|:-:|:-:|:-:|:-:
	Dataset|Attack|ACC|ASR|ACC|ASR|ACC|ASR|ACC|ASR
	CIFAR-10|BadNets|91.32|95.03|**89.31**|1.53|88.27|0.97|88.16|**0.67**
	CIFAR-10|SSBA|92.88|97.86|**90.84**|1.79|89.65|4.43|89.92|**0.81**
	CIFAR-10|WaNet|91.25|89.73|**91.26**|1.02|90.21|4.34|89.61|**0.66**
	Tiny ImageNet|BadNets|56.23|100|**51.52**|0.53|50.68|99.81|49.32|**0.49**
	Tiny ImageNet|SSBA|55.22|97.71|**51.85**|0.11|49.79|79.72|49.73|**0.05**
	Tiny ImageNet|WaNet|56.78|54.65|**54.65**|85.75|50.12|0.7|51.14|**0.15**

**In summary**, our theoretical analysis and the proposed method SAU are flexible to the perturbation set, and SAU can be generalized to attack with excessive trigger norms. Moreover, the Ensemble SAU serves as a stronger variant of SAU and can be effective for various attacks.

[1] Spatially Transformed Adversarial Examples.

[2] Perceptual Adversarial Robustness: Defense Against Unseen Threat Models.

---

### Decision · Program_Chairs · 2023-09-21

**Decision:**

Accept (poster)

**Comment:**

Two reviewers tend to reject this paper. One reviewer's rejection reason is that "I am not satisfied with the newly defined notations of backdoor risk and shared adversarial risk". AC feels that this is not a strong reason to reject this paper. AC also read the rebuttal to reviewer mKLw since this reviewer did not respond. AC feel that rebuttal has addressed the concern. Given that, AC feels this paper has the value to the community and tends to accept this paper. AC hopes the authors can add the valuable reviewers' comments into the camera-ready version.